# Revisiting the Last-Iterate Convergence of Stochastic Gradient Methods*

**Zijian Liu,**[†] **Zhengyuan Zhou**
Stern School of Business, New York University
{zl3067,zzhou}@stern.nyu.edu

## Abstract

In the past several years, the last-iterate convergence of the Stochastic Gradient Descent (SGD) algorithm has triggered people's interest due to its good performance in practice but lack of theoretical understanding. For Lipschitz convex functions, different works have established the optimal $O(\log(1/\delta) \log T/\sqrt{T})$ or $O(\sqrt{\log(1/\delta)/T})$ high-probability convergence rates for the final iterate, where $T$ is the time horizon and $\delta$ is the failure probability. However, to prove these bounds, all the existing works are either limited to compact domains or require almost surely bounded noises. It is natural to ask whether the last iterate of SGD can still guarantee the optimal convergence rate but without these two restrictive assumptions. Besides this important question, there are still lots of theoretical problems lacking an answer. For example, compared with the last-iterate convergence of SGD for non-smooth problems, only few results for smooth optimization have yet been developed. Additionally, the existing results are all limited to a non-composite objective and the standard Euclidean norm. It still remains unclear whether the last-iterate convergence can be provably extended to wider composite optimization and non-Euclidean norms. In this work, to address the issues mentioned above, we revisit the last-iterate convergence of stochastic gradient methods and provide the first unified way to prove the convergence rates both in expectation and in high probability to accommodate general domains, composite objectives, non-Euclidean norms, Lipschitz conditions, smoothness, and (strong) convexity simultaneously.

## 1 Introduction

In this paper, we consider the constrained composite optimization problem $\min_{x \in \mathcal{X}} F(x) := f(x) + h(x)$ where both $f(x)$ and $h(x)$ are convex (but possibly satisfying additional conditions such as strong convexity, smoothness, etc.) and $\mathcal{X} \subseteq \mathbb{R}^d$ is a nonempty closed convex set. Since a true gradient is computationally prohibitive to obtain (e.g., large-scale machine learning tasks) or even infeasible to access (e.g., streaming data), the classic Stochastic Gradient Descent (SGD) (Robbins & Monro, 1951) algorithm has emerged to be the gold standard for a light-weight yet effective computational procedure commonly adopted in production for the majority of machine learning tasks: SGD only requires a stochastic first-order oracle $\widehat{\partial} f(x)$ satisfying $\mathbb{E}[\widehat{\partial} f(x) \mid x] \in \partial f(x)$ where $\partial f(x)$ denotes the set of subgradients at $x$ and guarantees provable convergence under certain conditions (e.g., Lipschitz condition for $f(x)$ and finite variance on the stochastic oracle).

A particularly important problem in this area is to understand the last-iterate convergence of SGD, which has been motivated by experimental studies suggesting that returning the final iterate of SGD (or sometimes the average of the last few iterates) – rather than a running average – often yields a solution that works well in practice (e.g., Shalev-Shwartz et al. (2007)). As such, a fruitful line of literature (Rakhlin et al., 2011; Shamir & Zhang, 2013; Harvey et al., 2019a; Orabona, 2020; Jain et al., 2021) developed an extensive theoretical understanding of the non-asymptotic last-iterate convergence rate. Loosely speaking, two optimal upper bounds, $\widetilde{O}(1/\sqrt{T})$ for Lipschitz convex functions

---

*An extended version including more results is available at https://arxiv.org/abs/2312.08531.
†Corresponding author.

and $\widetilde{O}(1/T)$ for Lipschitz strongly convex functions, have been established for both expected and high-probability convergence when $h(x) = 0$ (see Subsection 1.2 for a detailed discussion). However, to prove the high-probability rates, the existing works rely on restrictive assumptions: compact domains or almost surely bounded noises (or both), which can simplify the analysis but are unrealistic in lots of problems. Until today, whether these two assumptions can be relaxed simultaneously or not still remains unclear. Naturally, we want to ask the following question:

*Q1: Is it possible to prove the high-probability last-iterate convergence of SGD for Lipschitz (strongly) convex functions without the compact domain assumption and beyond bounded noises?*

Compared with the fast development of non-smooth problems, the understanding of the last-iterate convergence of SGD for smooth problems (i.e., the gradients of $f(x)$ are Lipschitz) is much slower. The best expected bound for smooth convex optimization under $\mathcal{X} = \mathbb{R}^d$ until now is still $O(1/\sqrt[3]{T})$ due to Moulines & Bach (2011), which is far from the optimal rate $O(1/\sqrt{T})$ of the averaging output (Theorem 4.2 in Lan (2020)). However, temporarily suppose the domain is compact, one can immediately improve the rate from $O(1/\sqrt[3]{T})$ to $\widetilde{O}(1/\sqrt{T})$ by noticing that we can reduce the smooth problem to the Lipschitz problem[1] and use the known bounds from non-smooth convex optimization. Hence, one may expect the last-iterate convergence rate of SGD for smooth convex optimization should still be $O(1/\sqrt{T})$ for any kind of domain. If one further considers smooth and strongly convex problems, as far as we know, no formal result has been established for the final iterate of SGD in a general domain except for the expected $O(1/T)$ rate when $\mathcal{X} = \mathbb{R}^d$ under the PL-condition (which is known as a relaxation for strong convexity) (Gower et al., 2021; Khaled & Richtárik, 2023). The above discussion thereby leads us to the second main question:

*Q2: Does the last iterate of SGD provably converge in the rate of $O(1/\sqrt{T})$ for smooth and convex functions and $O(1/T)$ for smooth and strongly convex functions in a general domain?*

Besides the two aforementioned questions, there are still several important missing parts. First, recalling that our original goal is to optimize the composite objective $F(x) = f(x) + h(x)$, it is still unclear whether – and if so, how – the last-iterate convergence of this harder problem can be proved. Moreover, the previous works are limited to the standard Euclidean norm. Whereas, in lots of specialized tasks, it may be beneficial to employ a general norm instead of the $\ell_2$ norm to capture the non-Euclidean structure. However, whether this extension can be done remains open. Additionally, the proof techniques in the existing works vary in different settings, which builds a barrier for researchers to better understand the convergence of the last iterate of SGD. Motivated by these challenges, we would like to ask the final question:

*Q3: Is there a unified way to analyze the last-iterate convergence of stochastic gradient methods both in expectation and in high probability to accommodate general domains, composite objectives, non-Euclidean norms, Lipschitz conditions, smoothness, and (strong) convexity at once?*

## 1.1 OUR CONTRIBUTIONS

We provide affirmative answers to the above three questions and establish several new results by revisiting a simple algorithm, Composite Stochastic Mirror Descent (CSMD) (Duchi et al., 2010), which is based on the famous Mirror Descent (MD) algorithm (Nemirovski & Yudin, 1983; Beck & Teboulle, 2003) and includes SGD as a special case. Specifically, our contributions are as follows.

- We establish the first high-probability convergence result for the last iterate of CSMD in general domains under sub-Gaussian noises to answer *Q1* affirmatively.

- We prove the last iterate of CSMD can converge in the rate of $\widetilde{O}(1/\sqrt{T})$ for smooth convex optimization and $\widetilde{O}(1/T)$ for smooth strongly convex problems both in expectation and in high probability for any general domain $\mathcal{X}$, hence resolving *Q2*.

- We present a simple unified analysis that differs from the prior works and can be directly applied to various scenarios simultaneously, thus leading to a positive answer to *Q3*.

---

[1]To see why gradients are bounded in this case, we first fix a point $x_0$ in the domain. Then by smoothness, there is $\|\nabla f(x) - \nabla f(x_0)\|_2 = O(\|x - x_0\|_2)$ for any other point $x$, which immediately implies $\|\nabla f(x)\|_2 = O(\|x - x_0\|_2 + \|\nabla f(x_0)\|_2) = O(D + \|\nabla f(x_0)\|_2)$ where $D$ is the domain diameter.

## 1.2 RELATED WORK

We review the literature related to the last-iterate convergence of plain stochastic gradient methods[2] measured by the function value gap (see Subsection 2.1 for why we use this criterion) for both Lipschitz and smooth (strongly) convex optimization. We only focus on the algorithms without momentum or averaging since it is already known that, without further special assumptions, both operations cannot help to improve the lower order term $O(1/\sqrt{T})$ for general convex functions and $O(1/T)$ for strongly convex functions. For the last iterate of accelerated or averaging based stochastic gradient methods, we refer the reader to Nesterov & Shikhman (2015); Lan (2020); Orabona & Pál (2021) for in-expectation rates and Davis & Drusvyatskiy (2020); Gorbunov et al. (2020); Liu et al. (2023); Sadiev et al. (2023) for high-probability bounds. As for the last iterate of stochastic gradient methods for structured problems (e.g., linear regression), the reader can refer to Lei & Zhou (2017); Ge et al. (2019); Varre et al. (2021); Pan et al. (2022); Wu et al. (2022) for recent progress.

**Last iterate for Lipschitz (strongly) convex functions:** Rakhlin et al. (2011) is the first to show an expected $O(1/T)$ convergence for strongly convex functions. But such a bound is obtained under the additional assumption, smoothness with respect to optimum[3], meaning their result does not hold in general. Later on, Shamir & Zhang (2013) proves the first expected last-iterate rates $O(\log T/\sqrt{T})$ and $O(\log T/T)$ for convex and strongly convex objectives, respectively. The high-probability bounds turn out to be much harder than the expected rates. After several years, Harvey et al. (2019a) is the first to establish a high-probability bound in the rate of $O(\log(1/\delta)\log T/\sqrt{T})$ and $O(\log(1/\delta)\log T/T)$ for convex and strongly convex problems where $\delta$ is the probability of failure. Afterward, Jain et al. (2021) improves the previous two rates to $O(\sqrt{\log(1/\delta)/T})$ and $O(\log(1/\delta)/T)$ but with a non-standard step size schedule. They also prove the expected rates $O(1/\sqrt{T})$ and $O(1/T)$ under the new step size.

However, a main drawback for the general convex case in all the above papers is requiring a compact domain. To our best knowledge, Orabona (2020) is the first and the only work showing how to shave off this restriction, and thereby obtains an expected $O(\log T/\sqrt{T})$ rate for general domains yet it is unclear whether his proof can be extended to the high-probability case or not. Until recently, Zamani & Glineur (2023) exhibits a new proof on how to obtain the convergence rate for the last iterate but only for the deterministic case. Lastly, we would like to mention that all of these prior results are built for a non-composite objective $f(x)$ with the standard Euclidean norm.

**Last iterate for smooth (strongly) convex functions:** Compared with Lipschitz problems, much less work is done for smooth optimization. As far as we know, the only result showing a non-asymptotic rate for smooth convex functions dates back to Moulines & Bach (2011), in which the authors prove that the last iterate of SGD on $\mathbb{R}^d$ enjoys the expected rate $O(1/\sqrt[3]{T})$ under additional restrictive assumptions (e.g., mean squared smoothness). As for the strongly convex case, the expected rate $O(1/T)$ under the PL-condition (which is known as a relaxation for strong convexity) has been established but only for non-composite optimization under the Euclidean norm on the domain $\mathcal{X} = \mathbb{R}^d$ (Gower et al., 2021; Khaled & Richtárik, 2023).

**Lower bounds for last iterate:** Under the requirement $d = T$ where $d$ is the dimension of the problem, Harvey et al. (2019a) is the first to provide lower bounds $\Omega(\log T/\sqrt{T})$ under the step size $\Theta(1/\sqrt{t})$ for non-smooth convex functions and $\Omega(\log T/T)$ under the step size $\Theta(1/t)$ when strong convexity is additionally assumed. Note that these two rates are both proved for deterministic optimization meaning that they can be also applied to the expected lower bounds. Subsequently, when $d < T$ holds, Liu & Lu (2021) extends the above two lower bounds to $\Omega(\log d/\sqrt{T})$ (this bound is also true for the step size $\Theta(1/\sqrt{T})$) and $\Omega(\log d/T)$ under the same step size in Harvey et al. (2019a). As a consequence, lower bounds $\Omega(\log(d \wedge T)/T)$ and $\Omega(\log(d \wedge T)/\sqrt{T})$ have been established for both convex and strongly convex problems under the Lipschitz condition. For the high-probability bounds, Harvey et al. (2019a) shows their two deterministic bounds will incur an extra multiplicative factor $\Omega(\log(1/\delta))$, namely, $\Omega(\log(1/\delta)\log T/\sqrt{T})$ and $\Omega(\log(1/\delta)\log T/T)$. However, under more sophisticated designed step sizes, better upper bounds without the $\Omega(\log T)$ factor are possible, for example, see Jain et al. (2021) as mentioned above.

---

[2]To clarify, we mean the algorithm does not contain momentum or averaging operations.

[3]This means $\exists L > 0$ such that $f(x) - f(x^*) \le \frac{L}{2}\|x - x^*\|^2, \forall x \in \mathcal{X}$ where $x^* \in \operatorname{argmin}_{x \in \mathcal{X}} f(x)$.

Another highly related work is Liu et al. (2023), which presents a generic approach to establish the high-probability convergence of the *average iterate* under sub-Gaussian noises. We will show that their idea can be further used to prove the high-probability convergence for the *last iterate*.

## 2 PRELIMINARIES

**Notations:** $\mathbb{N}$ is the set of natural numbers (excluding 0). $[d] := \{1, 2, \cdots, d\}$ for any $d \in \mathbb{N}$. $a \vee b$ and $a \wedge b$ are defined as $\max\{a, b\}$ and $\min\{a, b\}$, respectively. $\langle \cdot, \cdot \rangle$ is the standard Euclidean inner product on $\mathbb{R}^d$. $\|\cdot\|$ represents a general norm on $\mathbb{R}^d$ and $\|\cdot\|_*$ is its dual norm. Given a set $A \subseteq \mathbb{R}^d$, $\mathrm{int}(A)$ stands for its interior points. For a function $f$, $\partial f(x)$ denotes the set of subgradients at $x$.

We focus on the following optimization problem in this work

$$\min_{x \in \mathcal{X}} F(x) := f(x) + h(x),$$

where $f$ and $h$ are both convex. $\mathcal{X} \subseteq \mathrm{int}(\mathrm{dom}(f)) \subseteq \mathbb{R}^d$ is a closed convex set. The requirement of $\mathcal{X} \subseteq \mathrm{int}(\mathrm{dom}(f))$ is only to guarantee the existence of $\partial f(x)$ for every point $x$ in $\mathcal{X}$ with no special reason. We emphasize that there is no compactness requirement on $\mathcal{X}$. Additionally, given $\psi$ being a differentiable and 1-strongly convex function with respect to $\|\cdot\|$ on $\mathcal{X}$ (i.e., $\psi(x) \geq \psi(y) + \langle \nabla\psi(y), x - y \rangle + \frac{1}{2}\|x - y\|^2, \forall x, y \in \mathcal{X}^4$), the Bregman divergence with respect to $\psi$ is defined as $D_\psi(x, y) := \psi(x) - \psi(y) - \langle \nabla\psi(y), x - y \rangle$. Throughout this paper, we assume that $\mathrm{argmin}_{x \in \mathcal{X}} h(x) + \langle g, x - y \rangle + \frac{D_\psi(x, y)}{\eta}$ can be solved efficiently for any $g \in \mathbb{R}^d, y \in \mathcal{X}, \eta > 0$.

Next, we list the assumptions used in our analysis:

**1. Existence of a local minimizer:** $\exists x^* \in \arg\min_{x \in \mathcal{X}} F(x)$ satisfying $F(x^*) > -\infty$.

**2. $(\mu_f, \mu_h)$-strongly convex:** For $k = f$ and $k = h$, $\exists \mu_k \geq 0$ such that $\mu_k D_\psi(x, y) \leq k(x) - k(y) - \langle g, x - y \rangle, \forall x, y \in \mathcal{X}, g \in \partial k(y)$. Moreover, we assume at least one of $(\mu_f, \mu_h)$ is zero.

**3. General $(L, M)$-smooth:** $\exists L \geq 0, M \geq 0$ such that $f(x) - f(y) - \langle g, x - y \rangle \leq \frac{L}{2}\|x - y\|^2 + M\|x - y\|, \forall x, y \in \mathcal{X}, g \in \partial f(y)$.

**4. Unbiased gradient estimator:** For a given $x^t \in \mathcal{X}$ in the $t$-th iterate, we can access an unbiased gradient estimator $\widehat{g}^t$, i.e., $\mathbb{E}\left[\widehat{g}^t \mid \mathcal{F}^{t-1}\right] \in \partial f(x^t)$, where $\mathcal{F}^t := \sigma(\widehat{g}^s, s \in [t])$ is the $\sigma$-algebra.

**5A. Finite variance:** $\exists \sigma \geq 0$ such that $\mathbb{E}\left[\|\xi^t\|_*^2 \mid \mathcal{F}^{t-1}\right] \leq \sigma^2$ where $\xi^t := \widehat{g}^t - \mathbb{E}\left[\widehat{g}^t \mid \mathcal{F}^{t-1}\right]$.

**5B. Sub-Gaussian noises:** $\exists \sigma \geq 0$ such that $\mathbb{E}\left[\exp(\lambda\|\xi^t\|_*^2) \mid \mathcal{F}^{t-1}\right] \leq \exp(\lambda\sigma^2), \forall \lambda \in \left[0, \sigma^{-2}\right]$.

We briefly discuss the assumptions here. Assumptions 1, 4, and 5A are standard in the stochastic optimization literature. Assumption 2 is known as relative strong convexity appeared in previous works (Hazan & Kale, 2014; Lu et al., 2018). We use it here since the last-iterate convergence rate will be derived for the CSMD algorithm, which employs Bregman divergence to exploit the non-Euclidean geometry. In particular, when $\|\cdot\|$ is the standard $\ell_2$ norm, we can take $\psi(x) = \frac{1}{2}\|x\|^2$ to recover the common definition of strong convexity. Assumption 3 is borrowed from Section 4.2 in Lan (2020). Note that both $L$-smooth functions (by taking $M = 0$) and $G$-Lipschitz functions (by taking $L = 0$ and $M = 2G$) are subclasses of Assumption 3. Additionally, we remark that Assumption 3 can be further relaxed to the following inequality

$$f(x) - f(y) - \langle g, x - y \rangle \leq LD_\psi(x, y) + M\sqrt{2D_\psi(x, y)}, \forall x, y \in \mathcal{X}, g \in \partial f(y),$$

but without changing the convergence results proved in this paper (see (1) and (5) in the proof of Lemma 4.1). Lastly, Assumption 5B is used for the high-probability convergence bound.

Our proofs for the high-probability convergence rely on the following simple fact for the centered sub-Gaussian random vector. Similar results have been proved in prior works (Vershynin, 2018; Liu et al., 2023). For completeness, we include the proof in Appendix A.

**Lemma 2.1.** *Given a $\sigma$-algebra $\mathcal{F}$ and a random vector $Z \in \mathbb{R}^d$ that is $\mathcal{F}$-measurable, if $\xi \in \mathbb{R}^d$ is a random vector satisfying $\mathbb{E}[\xi \mid \mathcal{F}] = 0$ and $\mathbb{E}\left[\exp(\lambda\|\xi\|_*^2) \mid \mathcal{F}\right] \leq \exp(\lambda\sigma^2), \forall \lambda \in \left[0, \sigma^{-2}\right]$, then*

$$\mathbb{E}\left[\exp\left(\langle \xi, Z \rangle\right) \mid \mathcal{F}\right] \leq \exp\left(\sigma^2\|Z\|^2\right).$$

---

[4] Rigorously speaking, $y$ should be in $\mathrm{int}(\mathcal{X})$. But one can think $\mathcal{X} \subseteq \mathrm{int}(\mathrm{dom}(\psi))$ to avoid this issue.

## 2.1 CONVERGENCE CRITERION

We always measure the convergence via the function value gap, i.e., $F(x) - F(x^*)$. There are several reasons to stick to this criterion. First, for the general convex case, the function value gap is the standard metric. Next, for strongly convex functions, the function value gap is always a stronger measurement than the squared distance to the optimal solution since $\|x - x^*\|^2 = O(F(x) - F(x^*))$ holds by strong convexity. Even if $F(x)$ is additionally assumed to be $(L, 0)$-smooth (e.g., $f(x)$ is $(L, 0)$-smooth and $h(x) = 0$), the bound on $\|x - x^*\|^2$ cannot be converted to the bound on $F(x) - F(x^*)$ since $F(x) - F(x^*) \leq \langle \nabla F(x^*), x - x^* \rangle + \frac{L}{2}\|x - x^*\|^2 = O(\|\nabla F(x^*)\|_* \|x - x^*\| + \|x - x^*\|^2)$, which is probably worse than $O(\|x - x^*\|^2)$ as $x^*$ is only a local minimizer meaning $\|\nabla F(x^*)\|_*$ possibly to be non-zero. Moreover, the function value gap is important in both the theoretical and practical sides of modern machine learning (e.g., the generalization error).

## 3 LAST-ITERATE CONVERGENCE OF STOCHASTIC GRADIENT METHODS

---

**Algorithm 1** Composite Stochastic Mirror Descent (CSMD)

---

**Input:** $x^1 \in \mathcal{X}, \eta_t > 0, \forall t \in [T]$.
**for** $t = 1$ **to** $T$ **do**
    $x^{t+1} = \operatorname{argmin}_{x \in \mathcal{X}} h(x) + \langle \widehat{g}^t, x - x^t \rangle + \frac{D_\psi(x, x^t)}{\eta_t}$
**Return** $x^{T+1}$

---

The algorithm, Composite Stochastic Mirror Descent, is presented in Algorithm 1. When $h(x) = 0$, Algorithm 1 degenerates to the standard Stochastic Mirror Descent algorithm. If we further consider the case $\|\cdot\| = \|\cdot\|_2$, Algorithm 1 can recover the standard projected SGD by taking $\psi(x) = \frac{1}{2}\|x\|_2^2$. We assume $T \geq 2$ throughout the following paper to avoid some algebraic issues in the proof. The full version of every following theorem with its proof is deferred into the appendix.

### 3.1 GENERAL CONVEX FUNCTIONS

In this section, we focus on the last-iterate convergence of Algorithm 1 for general convex functions (i.e., $\mu_f = \mu_h = 0$). First, the in-expectation convergence rates are shown in Theorem 3.1.

**Theorem 3.1.** *Under Assumptions 1-4 and 5A with $\mu_f = \mu_h = 0$:*

*If $T$ is unknown, by taking $\eta_t = \frac{1}{2L} \wedge \frac{\eta}{\sqrt{t}}, \forall t \in [T]$ with $\eta = \Theta\left(\sqrt{\frac{D_\psi(x^*, x^1)}{M^2 + \sigma^2}}\right)$, there is*

$$\mathbb{E}\left[F(x^{T+1}) - F(x^*)\right] \leq O\left(\frac{L D_\psi(x^*, x^1)}{T} + \frac{(M + \sigma)\sqrt{D_\psi(x^*, x^1)}\log T}{\sqrt{T}}\right).$$

*If $T$ is known, by taking $\eta_t = \frac{1}{2L} \wedge \frac{\eta}{\sqrt{T}}, \forall t \in [T]$ with $\eta = \Theta\left(\sqrt{\frac{D_\psi(x^*, x^1)}{(M^2 + \sigma^2)\log T}}\right)$, there is*

$$\mathbb{E}\left[F(x^{T+1}) - F(x^*)\right] \leq O\left(\frac{L D_\psi(x^*, x^1)}{T} + \frac{(M + \sigma)\sqrt{D_\psi(x^*, x^1)\log T}}{\sqrt{T}}\right).$$

Before moving on to the high-probability bounds, we would like to talk more about these in-expectation convergence results. First, the constant $\eta$ here is optimized to obtain the best dependence on the parameters $M, \sigma$ and $D_\psi(x^*, x^1)$. Indeed, the last iterate provably converges for arbitrary $\eta > 0$ but with a worse dependence on $M, \sigma$ and $D_\psi(x^*, x^1)$. We refer the reader to Theorem C.1 in the appendix for a full version of Theorem 3.1 with any $\eta > 0$.

Next, by taking $L = 0$, we immediately get the (nearly) optimal $\widetilde{O}(1/\sqrt{T})$ convergence rate of the last iterate for non-smooth functions. Note that our bounds are better than Shamir & Zhang (2013) since it only works for bounded domains and non-composite optimization. Besides, when considering smooth problems (taking $M = 0$), to our best knowledge, our $\widetilde{O}(L/T + \sigma/\sqrt{T})$ bound

is the first improvement since the $O(1/\sqrt[3]{T})$ rate by Moulines & Bach (2011). Moreover, compared to Moulines & Bach (2011), Theorem 3.1 does not rely on some restrictive assumptions like bounded stochastic gradients or $x^*$ being a global optimal point but is able to be used for the more general composite problems. Additionally, it is worth remarking that the $\widetilde{O}(L/T + \sigma/\sqrt{T})$ rate matches the optimal $O(L/T + \sigma/\sqrt{T})$ rate for the averaged output $x_{avg}^{T+1} = (\sum_{t=2}^{T+1} x^t)/T$ (Lan, 2020) up to an extra logarithmic factor. Notably, our bounds are also adaptive to the noise $\sigma$ in this case. In other words, we can recover the well-known $O(L/T)$ rate for the last iterate of the GD algorithm in the noiseless case. Last but most importantly, our proof is unified and thus can be applied to different settings (e.g., general domains, $(L, M)$-smoothness, non-Euclidean norms, etc.) simultaneously.

*Remark* 3.2. Orabona (2020) exhibited a circuitous method based on comparing the last iterate with the averaged output to show the expected last-iterate convergence for non-composite non-smooth convex optimization in general domains. However, it did not explicitly generalize to the broader problems considered in this paper. Moreover, our method is done in a direct manner (see Section 4).

**Theorem 3.3.** *Under Assumptions 1-4 and 5B with $\mu_f = \mu_h = 0$ and let $\delta \in (0, 1)$:*

*If $T$ is unknown, by taking $\eta_t = \frac{1}{2L} \wedge \frac{\eta}{\sqrt{t}}, \forall t \in [T]$ with $\eta = \Theta\left(\sqrt{\frac{D_\psi(x^*,x^1)}{M^2+\sigma^2 \log \frac{1}{\delta}}}\right)$, then with probability at least $1 - \delta$, there is*

$$F(x^{T+1}) - F(x^*) \leq O\left(\frac{LD_\psi(x^*,x^1)}{T} + \frac{(M + \sigma\sqrt{\log \frac{1}{\delta}})\sqrt{D_\psi(x^*,x^1)} \log T}{\sqrt{T}}\right).$$

*If $T$ is known, by taking $\eta_t = \frac{1}{2L} \wedge \frac{\eta}{\sqrt{T}}, \forall t \in [T]$ with $\eta = \Theta\left(\sqrt{\frac{D_\psi(x^*,x^1)}{(M^2+\sigma^2 \log \frac{1}{\delta}) \log T}}\right)$, then with probability at least $1 - \delta$, there is*

$$F(x^{T+1}) - F(x^*) \leq O\left(\frac{LD_\psi(x^*,x^1)}{T} + \frac{(M + \sigma\sqrt{\log \frac{1}{\delta}})\sqrt{D_\psi(x^*,x^1) \log T}}{\sqrt{T}}\right).$$

In Theorem 3.3, we present the high-probability bounds for $(L, M)$-smooth functions. Again, the constant $\eta$ is picked to get the best dependence on the parameters $M, \sigma, D_\psi(x^*, x^1)$ and $\log(1/\delta)$. The full version of Theorem 3.3 with arbitrary $\eta$, Theorem C.2, is deferred into the appendix. Compared with Theorem 3.1, the high-probability rates only incur an extra $O(\sqrt{\log(1/\delta)})$ factor (or $O(\log(1/\delta))$ for arbitrary $\eta$, which is known to be optimal for $L = 0$ (Harvey et al., 2019a)).

In contrast to the previous bounds (Harvey et al., 2019a; Jain et al., 2021) that only work for Lipschitz functions in a compact domain, our results are the first to describe the high-probability behavior of Algorithm 1 for the wider $(L, M)$-smooth function class in a general domain even with sub-Gaussian noises, not to mention composite objectives and non-Euclidean norms. Even in the special smooth case (setting $M = 0$), as far as we know, this is also the first last-iterate high-probability bound being adaptive to the noise $\sigma$ at the same time for plain stochastic gradient methods. Unlike the previous proofs employing some new probability tools (e.g., the generalized Freedman's inequality in Harvey et al. (2019a)), our high-probability argument is simple and only based on the basic property of sub-Gaussian random vectors (see Lemma 2.1). Therefore, we believe our work can bring some new insights to researchers to gain a better understanding of the convergence for the last iterate of stochastic gradient methods.

## 3.2 STRONGLY CONVEX FUNCTIONS

Now we turn our attention to strongly convex functions. Due to the space limitation, we only provide the results for the case of $\mu_f > 0$ and $\mu_h = 0$. The other case, $\mu_f = 0$ and $\mu_h > 0$, will be delivered in Appendix D.2.

**Theorem 3.4.** *Under Assumptions 1-4 and 5A with $\mu_f > 0$ and $\mu_h = 0$, let $\kappa_f := \frac{L}{\mu_f} \geq 0$:*

*If $T$ is unknown, by taking either $\eta_t = \frac{1}{\mu_f(t+2\kappa_f)}, \forall t \in [T]$ or $\eta_t = \frac{2}{\mu_f(t+1+4\kappa_f)}, \forall t \in [T]$, there is*

$$\mathbb{E}\left[F(x^{T+1}) - F(x^*)\right] \leq \begin{cases} O\left(\frac{LD_\psi(x^*,x^1)}{T} + \frac{(M^2+\sigma^2)\log T}{\mu_f(T+\kappa_f)}\right) & \eta_t = \frac{1}{\mu_f(t+2\kappa_f)}, \forall t \in [T] \\ O\left(\frac{L(1+\kappa_f)D_\psi(x^*,x^1)}{T(T+\kappa_f)} + \frac{(M^2+\sigma^2)\log T}{\mu_f(T+\kappa_f)}\right) & \eta_t = \frac{2}{\mu_f(t+1+4\kappa_f)}, \forall t \in [T] \end{cases}.$$

*If $T$ is known, by taking $\eta_t = \begin{cases} \frac{1}{\mu_f(1+2\kappa_f)} & t = 1 \\ \frac{1}{\mu_f(\eta+2\kappa_f)} & 2 \leq t \leq \tau \\ \frac{2}{\mu_f(t-\tau+2+4\kappa_f)} & t \geq \tau + 1 \end{cases}, \forall t \in [T]$ with $\eta := 1.5$ and $\tau := \lceil \frac{T}{2} \rceil$,*

*there is*

$$\mathbb{E}\left[F(x^{T+1}) - F(x^*)\right] \leq O\left(\frac{LD_\psi(x^*, x^1)}{\exp\left(\frac{T}{3+4\kappa_f}\right)} + \frac{(M^2+\sigma^2)\log T}{\mu_f(T+\kappa_f)}\right).$$

The in-expectation rates are stated in Theorem 3.4 where the constant $\eta = 1.5$ is chosen without any special reason. Generally speaking, it can be any non-negative number satisfying $\eta + \kappa_f > 1$. The interested reader could refer to Theorem D.1 in the appendix for a completed version of Theorem 3.4. We would like to remind that $\kappa_f \geq 1$ is not necessary as we are considering the general $(L, M)$-smooth functions. Hence, it can be zero.

As before, we first take $L = 0$ to consider the special Lipschitz case. Due to $\kappa_f = 0$ now, all bounds will degenerate to $O(\log T/T)$, which is known to be optimal for the step size $1/\mu_f t$ (Harvey et al., 2019a) and only incurs an extra $O(\log T)$ factor compared with the best $O(1/T)$ bound when $T$ is known (Jain et al., 2021). We would also like to mention that Theorem 3.4 is the first to give the in-expectation last-iterate bound for the step size $2/\mu_f(t+1)$. Interestingly, the extra $O(\log T)$ factor appears again compared to the known $O(1/T)$ bound on the function value gap for the non-uniform averaging strategy under this step size (Lacoste-Julien et al., 2012). Besides, Lacoste-Julien et al. (2012) also shows $\mathbb{E}\left[\|x^{T+1} - x^*\|_2^2\right] = O(1/T)$. Whereas, it is currently unknown whether our $\mathbb{E}\left[F(x^{T+1}) - F(x^*)\right] = O(\log T/T)$ bound can be improved to match the $O(1/T)$ rate or not.

For the general $(L, M)$-smooth case (even for $(L, 0)$-smoothness), our bounds are the first convergence results for the last iterate of stochastic gradient methods with respect to the function value gap[5]. Remarkably, all of these rates do not require prior knowledge of $M$ or $\sigma$ to set the step size. In particular, the bound for known $T$ is adaptive to $\sigma$ when $M = 0$, i.e., it can recover the well-known linear convergence rate $O(\exp(-T/\kappa_f))$ when $\sigma = 0$.

**Theorem 3.5.** *Under Assumptions 1-4 and 5B with $\mu_f > 0$ and $\mu_h = 0$, let $\kappa_f := \frac{L}{\mu_f} \geq 0$ and $\delta \in (0, 1)$:*

*If $T$ is unknown, by taking either $\eta_t = \frac{1}{\mu_f(t+2\kappa_f)}, \forall t \in [T]$ or $\eta_t = \frac{2}{\mu_f(t+1+4\kappa_f)}, \forall t \in [T]$, then with probability at least $1 - \delta$, there is*

$$F(x^{T+1}) - F(x^*) \leq \begin{cases} O\left(\frac{\mu_f(1+\kappa_f)D_\psi(x^*,x^1)}{T} + \frac{(M^2+\sigma^2\log\frac{1}{\delta})\log T}{\mu_f(T+\kappa_f)}\right) & \eta_t = \frac{1}{\mu_f(t+2\kappa_f)}, \forall t \in [T] \\ O\left(\frac{\mu_f(1+\kappa_f)^2 D_\psi(x^*,x^1)}{T(T+\kappa_f)} + \frac{(M^2+\sigma^2\log\frac{1}{\delta})\log T}{\mu_f(T+\kappa_f)}\right) & \eta_t = \frac{2}{\mu_f(t+1+4\kappa_f)}, \forall t \in [T] \end{cases}.$$

*If $T$ is known, by taking $\eta_t = \begin{cases} \frac{1}{\mu_f(1+2\kappa_f)} & t = 1 \\ \frac{1}{\mu_f(\eta+2\kappa_f)} & 2 \leq t \leq \tau \\ \frac{2}{\mu_f(t-\tau+2+4\kappa_f)} & t \geq \tau + 1 \end{cases}, \forall t \in [T]$ with $\eta := 1.5$ and $\tau := \lceil \frac{T}{2} \rceil$,*

*then with probability at least $1 - \delta$, there is*

$$F(x^{T+1}) - F(x^*) \leq O\left(\frac{\mu_f(1+\kappa_f)D_\psi(x^*, x^1)}{\exp\left(\frac{T}{3+4\kappa_f}\right)} + \frac{(M^2+\sigma^2\log\frac{1}{\delta})\log T}{\mu_f(T+\kappa_f)}\right).$$

To finish this section, we provide the high-probability convergence results in Theorem 3.5. Again, the constant $\eta = 1.5$ is set without any particular reason. The full statement with general $\eta$, Theorem D.2, can be found in the appendix. Besides, $\kappa_f$ is possible to be zero as mentioned above. Compared with Theorem 3.4, only an additional $O(\log(1/\delta))$ factor appears. Such extra loss is known to be inevitable for $L = 0$ due to Harvey et al. (2019a).

---

[5]Note that the rates under the PL-condition (e.g., Gower et al. (2021); Khaled & Richtárik (2023)) are incompatible with our settings since they can be only applied to non-constrained, non-composite and $(L, 0)$-smooth optimization problems with the Euclidean norm.

For the Lipschitz case (i.e., $L = \kappa_f = 0$), by noticing $D_\psi(x^*, x^1) = O(M^2/\mu_f^2)^6$, all of these bounds will degenerate to $O(\log(1/\delta)\log T/T)$ matching the best-known last-iterate bound proved by Harvey et al. (2019a) for the step size $1/\mu_f t$. For the step size $2/\mu_f(t+1)$, Harvey et al. (2019b) has proved the high-probability bound $O(\log(1/\delta)/T)$ for the non-uniform averaging output instead of the last iterate. Hence, as far as we know, our high-probability rate for the step size $2/\mu_f(t+1)$ is new. However, we would like to mention that our bound for known $T$ is worse by a logarithmic factor than Jain et al. (2021), though, which assumes bounded noises.

Finally, let us go back to the general $(L, M)$-smooth case. To our best knowledge, our results are first to prove the last iterate of plain stochastic gradient methods enjoying the provable high-probability convergence even for the smooth case ($M = 0$). Hence, we believe our work closes the gap between the lack of theoretical understanding and good performance of the last iterate of SGD for smooth and strongly convex functions. Lastly, the same as the in-expectation bound for known $T$ in Theorem 3.4, our high-probability bound is also adaptive to $\sigma$ when $M = 0$.

## 4 Unified Theoretical Analysis

In this section, we introduce the ideas in our analysis and present three important lemmas, all the missing proofs of which are deferred into Appendix B.

The key insight in our proofs is to utilize the convexity of $F(x)$, which is highly inspired by the recent work (Zamani & Glineur, 2023). To be more precise, using the classic convergence analysis for non-composite Lipschitz convex problems as an example, people always consider to upper bound the function value gap $f(x^t) - f(x^*)$ (probably with some weight before it) then sum them over time to obtain the ergodic rate. Whereas, in such an argument, convexity is not necessary in fact (except if one wants to bound the average iterate in the last step). Hence, if the convexity of $f$ can be utilized somewhere, it is reasonable to expect a last-iterate convergence guarantee. Actually, this thought is possible as shown by Zamani & Glineur (2023), in which the authors upper bound the quantity $f(x^t) - f(z^t)$ where $z^t$ is a carefully chosen convex combination of other points and finally obtain the last-iterate rate by lower bounding $-f(z^t)$ via convexity. More precisely, suppose $z^t := \alpha_0^t x^* + \sum_{s=1}^t \alpha_s^t x^t$ where $\alpha_s^t \geq 0, \forall s \in \{0\} \cup [t], \forall t \in [T]$ satisfy $\sum_{s=0}^t \alpha_s^t = 1, \forall t \in [T]$, then there is $-f(z^t) \geq -\alpha_0^t f(x^*) - \sum_{s=1}^t \alpha_s^t f(x^t)$ by the convexity of $f$. By properly picking $\alpha_s^t$, one can finally bound $f(x^T) - f(x^*)$ as proved by Zamani & Glineur (2023).

Though Zamani & Glineur (2023) only shows how to prove the last-iterate convergence for deterministic non-composite Lipschitz convex optimization under the Euclidean norm, we can catch the most important message conveyed by their paper and apply it to our settings. Formally speaking, we will upper bound the term $F(x^{t+1}) - F(z^t)$ for a well-designed $z^t$ rather than directly bound the function value gap $F(x^{t+1}) - F(x^*)$. This idea can finally help us construct a unified proof and obtain several novel results without prior restrictive assumptions. By careful calculations, the new analysis leads us to the following most important and unified result, Lemma 4.1.

**Lemma 4.1.** *Under Assumptions 1-3, suppose* $\eta_t \leq \frac{1}{2L \vee \mu_f}, \forall t \in [T]$ *and let* $\gamma_t := \eta_t \prod_{s=2}^t \frac{1 + \mu_h \eta_{s-1}}{1 - \mu_f \eta_s}, \forall t \in [T]$, *if* $w_t \geq 0, \forall t \in [T]$ *is a non-increasing sequence and* $v_t > 0$ *is defined as* $v_t := \frac{w_T \gamma_T}{\sum_{s=t}^T w_s \gamma_s}, \forall t \in [T]$ *and* $v_0 := v_1$, *then we have*

$$w_T \gamma_T v_T \left( F(x^{T+1}) - F(x^*) \right)$$

$$\leq w_1(1 - \mu_f \eta_1)v_0 D_\psi(x^*, x^1) + \sum_{t=1}^T 2w_t \gamma_t \eta_t v_t(M^2 + \|\xi^t\|_*^2)$$

$$+ \sum_{t=1}^T w_t \gamma_t v_{t-1} \langle \xi^t, z^{t-1} - x^t \rangle + \sum_{t=2}^T (w_t - w_{t-1})\gamma_t(\eta_t^{-1} - \mu_f)v_{t-1} D_\psi(z^{t-1}, x^t),$$

*where* $\xi^t := \widehat{g}^t - \mathbb{E}\left[\widehat{g}^t \mid \mathcal{F}^{t-1}\right], \forall t \in [T]$ *and* $z^t := \frac{v_0}{v_t}x^* + \sum_{s=1}^t \frac{v_s - v_{s-1}}{v_t}x^s, \forall t \in \{0\} \cup [T]$.

Let us discuss Lemma 4.1 more here. The requirement of the step size $\eta_t$ having an upper bound $1/2L \vee \mu_f$ is common in the optimization literature. $\gamma_t$ is used to ensure we can telescope sum some

---

[6] This holds now due to $\mu_f \|x^* - x^1\|^2/2 \leq \mu_f D_\psi(x^*, x^1) \leq M\|x^* - x^1\|$.

terms. For the special case $\mu_f = \mu_h = 0$, it degenerates to $\eta_t$. $\xi^t$ naturally shows up as we are considering stochastic optimization. The most important sequences are $w_t, v_t$ and $z^t$. As mentioned above, the appearance of $z^t$ is to make sure to get the last-iterate convergence. For how to find such a sequence, we refer the reader to our proofs in Appendix B for details.

We would like to say more about the sequence $w_t$ before moving on. Suppose we are in the deterministic case temporarily, i.e., $\xi^t = 0$, then a natural choice is to set $w_t = 1, \forall t \in [T]$ to remove the last residual summation. It turns out this is the correct choice even for the following in-expectation bound in Lemma 4.2. So why do we still need this redundant $w_t$? The reason is that setting $w_t$ to be one is not enough for the high-probability bound. More precisely, if we still choose $w_t = 1, \forall t \in [T]$, then there will be some extra positive terms after the concentration argument in the R.H.S. of the inequality in Lemma 4.1. To deal with this issue, we borrow the idea recently developed by Liu et al. (2023), in which the authors employ an extra sequence $w_t$ to give a clear proof for the high-probability bound for stochastic gradient methods. We refer the reader to Liu et al. (2023) for a detailed explanation of this technique.

**Lemma 4.2.** *Under Assumptions 1-4 and 5A, suppose* $\eta_t \leq \frac{1}{2L \vee \mu_f}, \forall t \in [T]$ *and let* $\gamma_t :=$ $\eta_t \prod_{s=2}^t \frac{1+\mu_h \eta_{s-1}}{1-\mu_f \eta_s}, \forall t \in [T]$, *then we have*

$$\mathbb{E}\left[F(x^{T+1}) - F(x^*)\right] \leq \frac{(1-\mu_f \eta_1)D_\psi(x^*, x^1)}{\sum_{t=1}^T \gamma_t} + 2(M^2 + \sigma^2)\sum_{t=1}^T \frac{\gamma_t \eta_t}{\sum_{s=t}^T \gamma_s}.$$

Suppose Lemma 4.1 holds, Lemma 4.2 is immediately obtained by setting $w_t = 1, \forall t \in [T]$ and using Assumptions 4 and 5A. This unified result for the expected last-iterate convergence can be applied to many different settings like composite optimization and non-Euclidean norms without any restrictive assumptions.

**Lemma 4.3.** *Under Assumptions 1-4 and 5B, suppose* $\eta_t \leq \frac{1}{2L \vee \mu_f}, \forall t \in [T]$ *and let* $\gamma_t :=$ $\eta_t \prod_{s=2}^t \frac{1+\mu_h \eta_{s-1}}{1-\mu_f \eta_s}, \forall t \in [T]$, *then for any* $\delta \in (0,1)$, *with probability at least* $1-\delta$, *we have*

$$F(x^{T+1}) - F(x^*) \leq 2\left(1 + \max_{2 \leq t \leq T} \frac{1}{1-\mu_f \eta_t}\right)$$
$$\times \left[\frac{D_\psi(x^*, x^1)}{\sum_{t=1}^T \gamma_t} + \left(M^2 + \sigma^2\left(1 + 2\log\frac{2}{\delta}\right)\right)\sum_{t=1}^T \frac{\gamma_t \eta_t}{\sum_{s=t}^T \gamma_s}\right].$$

To get Lemma 4.3, we need some extra effort to find the correct $w_t$ and invoke a simple property of sub-Gaussian random vectors (Lemma 2.1). The details can be found in Appendix B. Compared with prior works, this unified high-probability bound can be applied to various scenarios including general domains and sub-Gaussian noises.

Equipped with Lemma 4.2 and Lemma 4.3, we can prove all theorems provided in Section 3 by plugging in different step sizes for different cases.

## 5 CONCLUSION

In this work, we present a unified analysis for the last-iterate convergence of stochastic gradient methods and obtain several new results. More specifically, we establish the (nearly) optimal convergence of the last iterate of the CSMD algorithm both in expectation and in high probability. Our proofs can not only handle different function classes simultaneously but also be applied to composite problems with non-Euclidean norms on general domains. We believe our work develops a deeper understanding of stochastic gradient methods. However, there still remain many directions worth exploring. For example, it could be interesting to see whether our proof can be extended to adaptive gradient methods like AdaGrad (McMahan & Streeter, 2010; Duchi et al., 2011). We leave this important question as future work and expect it to be addressed.

## ACKNOWLEDGMENTS

This work is generously supported by the National Science Foundation grant CCF-2106508. Zhengyuan Zhou would also like to thank the 2024-2025 NYU Center for Global Economy and Business faculty grant and the NYU Research Catalyst Prize. We also thank the anonymous reviewers for their constructive comments and suggestions.

**Ethics Statement:** This is a theory work. Hence, there are no potential ethics concerns.

**Reproducibility Statement:** We include the full proofs of all theorems in the appendix.

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

## A    PROOF OF LEMMA 2.1

Before giving the proof of Lemma 2.1, we need the following property of sub-Gaussian vectors. This result is already known before (see Vershynin (2018)). We provide a proof here to make the paper self-consistent.

**Lemma A.1.** *Given a $\sigma$-algebra $\mathcal{F}$, if $\xi \in \mathbb{R}^d$ is a random vector satisfying $\mathbb{E}\left[\exp(\lambda\|\xi\|_*^2) \mid \mathcal{F}\right] \leq \exp(\lambda\sigma^2), \forall \lambda \in \left[0, \sigma^{-2}\right]$, then for any integer $k \geq 1$ we have*

$$\mathbb{E}\left[\|\xi\|_*^{2k} \mid \mathcal{F}\right] \leq \begin{cases} \sigma^2 & k = 1 \\ e(k!)\sigma^{2k} & k \geq 2 \end{cases}.$$

*Proof.* For the case $k = 1$, given any $\lambda \in \left[0, \sigma^{-2}\right]$, there is

$$\exp\left(\mathbb{E}\left[\lambda\|\xi\|_*^2 \mid \mathcal{F}\right]\right) \leq \mathbb{E}\left[\exp\left(\lambda\|\xi\|_*^2\right) \mid \mathcal{F}\right] \leq \exp(\lambda\sigma^2) \Rightarrow \mathbb{E}\left[\|\xi\|_*^2 \mid \mathcal{F}\right] \leq \sigma^2.$$

For $k \geq 2$, we have

$$\mathbb{E}\left[\|\xi\|_*^{2k} \mid \mathcal{F}\right] = \mathbb{E}\left[\int_0^\infty 2kt^{2k-1}\mathbb{1}\left[\|\xi\|_* \geq t\right] \mathrm{d}t \mid \mathcal{F}\right] = \int_0^\infty 2kt^{2k-1}\mathbb{E}\left[\mathbb{1}\left[\|\xi\|_* \geq t\right] \mid \mathcal{F}\right]\mathrm{d}t$$

$$\leq \int_0^\infty 2kt^{2k-1}\mathbb{E}\left[\frac{\exp(\sigma^{-2}\|\xi\|_*^2)}{\exp(\sigma^{-2}t^2)} \mid \mathcal{F}\right]\mathrm{d}t \overset{(a)}{\leq} \int_0^\infty 2ekt^{2k-1}\exp(-\sigma^{-2}t^2)\mathrm{d}t$$

$$\overset{(b)}{=} \int_0^\infty ek\sigma^{2k}s^{k-1}\exp(-s)\mathrm{d}s = ek\sigma^{2k}\Gamma(k) = e(k!)\sigma^{2k},$$

where $(a)$ is by $\mathbb{E}\left[\exp(\sigma^{-2}\|\xi\|_*^2) \mid \mathcal{F}\right] \leq \exp(\sigma^{-2}\sigma^2) = e$ and $(b)$ is by the change of variable $t = \sigma\sqrt{s}$.  □

Now we are ready to prove Lemma 2.1.

*Proof of Lemma 2.1.* Note that

$$\mathbb{E}\left[\exp\left(\langle\xi, Z\rangle\right) \mid \mathcal{F}\right]$$

$$= \mathbb{E}\left[1 + \langle\xi, Z\rangle + \sum_{k=2}^\infty \frac{(\langle\xi, Z\rangle)^k}{k!} \mid \mathcal{F}\right] \leq \mathbb{E}\left[\langle\xi, Z\rangle \mid \mathcal{F}\right] + \mathbb{E}\left[1 + \sum_{k=2}^\infty \frac{\|\xi\|_*^k\|Z\|^k}{k!} \mid \mathcal{F}\right]$$

$$\overset{(a)}{=} \mathbb{E}\left[1 + \sum_{k=1}^\infty \frac{\|\xi\|_*^{2k}\|Z\|^{2k}}{(2k)!} + \sum_{k=1}^\infty \frac{\|\xi\|_*^{2k+1}\|Z\|^{2k+1}}{(2k+1)!} \mid \mathcal{F}\right]$$

$$\overset{(b)}{\leq} \mathbb{E}\left[1 + \sum_{k=1}^\infty \frac{\|\xi\|_*^{2k}\|Z\|^{2k}}{(2k)!} + \sum_{k=1}^\infty \frac{\|\xi\|_*^{2k}\|Z\|^{2k} + \|\xi\|_*^{2k+2}\|Z\|^{2k+2}/4}{(2k+1)!} \mid \mathcal{F}\right]$$

$$= \mathbb{E}\left[1 + \frac{2\|\xi\|_*^2\|Z\|^2}{3} + \sum_{k=2}^\infty \|\xi\|_*^{2k}\|Z\|^{2k}\left(\frac{1}{4(2k-1)!} + \frac{1}{(2k)!} + \frac{1}{(2k+1)!}\right) \mid \mathcal{F}\right]$$

$$= \mathbb{E}\left[1 + \frac{2\|\xi\|_*^2\|Z\|^2}{3} + \sum_{k=2}^\infty \|\xi\|_*^{2k}\|Z\|^{2k}\frac{1 + k/2 + 1/(2k+1)}{(2k)!} \mid \mathcal{F}\right]$$

$$\overset{(c)}{\leq} 1 + \frac{2\sigma^2\|Z\|^2}{3} + \sum_{k=2}^\infty \frac{\sigma^{2k}\|Z\|^{2k}}{k!} \cdot \frac{e(1 + k/2 + 1/(2k+1))}{\binom{2k}{k}}$$

$$\overset{(d)}{\leq} 1 + \sigma^2\|Z\|^2 + \sum_{k=2}^\infty \frac{\sigma^{2k}\|Z\|^{2k}}{k!} = \exp\left(\sigma^2\|Z\|^2\right),$$

where $(a)$ is by $\mathbb{E}\left[\langle\xi, Z\rangle \mid \mathcal{F}\right] = \langle\mathbb{E}\left[\xi \mid \mathcal{F}\right], Z\rangle = 0$, $(b)$ holds due to AM-GM inequality, $(c)$ is by applying Lemma A.1 to $\mathbb{E}\left[\|\xi\|_*^{2k}\|Z\|^{2k} \mid \mathcal{F}\right] = \|Z\|^{2k}\mathbb{E}\left[\|\xi\|_*^{2k} \mid \mathcal{F}\right]$ and $(d)$ is by $2/3 < 1$ and

$$\max_{k \geq 2, k \in \mathbb{N}} \frac{e(1 + k/2 + 1/(2k+1))}{\binom{2k}{k}} = \frac{e(1 + 1 + 1/5)}{6} < 1.$$

□

# B MISSING PROOFS IN SECTION 4

In this section, we provide the missing proofs of the most important three lemmas.

## B.1 PROOF OF LEMMA 4.1

*Proof of Lemma 4.1.* Inspired by Zamani & Glineur (2023), we first introduce the following auxiliary sequence

$$z^t := \begin{cases} \left(1 - \frac{v_{t-1}}{v_t}\right) x^t + \frac{v_{t-1}}{v_t} z^{t-1} & t \in [T] \\ x^* & t = 0 \end{cases} \Leftrightarrow z^t := \frac{v_0}{v_t} x^* + \sum_{s=1}^{t} \frac{v_s - v_{s-1}}{v_t} x^s, \forall t \in \{0\} \cup [T],$$

where we recall that $v_t = \frac{w_T \gamma_T}{\sum_{s=t}^{T} w_s \gamma_s} \geq 0, \forall t \in [T]$ and $v_0 = v_1$ are non-decreasing. Note that $z^t$ always falls in the domain $\mathcal{X}$ because it is a convex combination of $x^*, x^1, \cdots, x^t$ that are in $\mathcal{X}$.

Now, we start the proof from the $(L, M)$-smoothness of $f$,

$$f(x^{t+1}) - f(x^t) \leq \langle g^t, x^{t+1} - x^t \rangle + \frac{L}{2}\|x^{t+1} - x^t\|^2 + M\|x^{t+1} - x^t\|$$

$$= \langle \xi^t, z^t - x^t \rangle + \underbrace{\langle \xi^t, x^t - x^{t+1} \rangle}_{\text{I}} + \underbrace{\langle \widehat{g}^t, x^{t+1} - z^t \rangle}_{\text{II}}$$

$$+ \underbrace{\langle g^t, z^t - x^t \rangle}_{\text{III}} + \underbrace{\frac{L}{2}\|x^{t+1} - x^t\|^2 + M\|x^{t+1} - x^t\|}_{\text{IV}}, \tag{1}$$

where $g^t := \mathbb{E}\left[\widehat{g}^t | \mathcal{F}^{t-1}\right] \in \partial f(x^t)$ and $\xi^t := \widehat{g}^t - g^t$. Next, we bound these four terms respectively.

- For term I, by applying Cauchy-Schwarz inequality, the 1-strong convexity of $\psi$ and AM-GM inequality, we can get the following upper bound

$$\text{I} \leq \|\xi^t\|_* \|x^t - x^{t+1}\| \leq \|\xi^t\|_* \sqrt{2 D_\psi(x^{t+1}, x^t)} \leq 2\eta_t \|\xi^t\|_*^2 + \frac{D_\psi(x^{t+1}, x^t)}{4\eta_t}. \tag{2}$$

- For term II, we recall that the update rule is $x^{t+1} = \operatorname{argmin}_{x \in \mathcal{X}} h(x) + \langle \widehat{g}^t, x - x^t \rangle + \frac{D_\psi(x, x^t)}{\eta_t}$. Hence, by the optimality condition of $x^{t+1}$, there exists $h^{t+1} \in \partial h(x^{t+1})$ such that for any $y \in \mathcal{X}$

$$\left\langle h^{t+1} + \widehat{g}^t + \frac{\nabla\psi(x^{t+1}) - \nabla\psi(x^t)}{\eta_t}, x^{t+1} - y \right\rangle \leq 0,$$

which implies

$$\langle \widehat{g}^t, x^{t+1} - y \rangle \leq \frac{\langle \nabla\psi(x^t) - \nabla\psi(x^{t+1}), x^{t+1} - y \rangle}{\eta_t} + \langle h^{t+1}, y - x^{t+1} \rangle$$

$$\leq \frac{D_\psi(y, x^t) - D_\psi(y, x^{t+1}) - D_\psi(x^{t+1}, x^t)}{\eta_t} + h(y) - h(x_{t+1}) - \mu_h D_\psi(y, x^{t+1})$$

where the last inequality holds due to $\langle \nabla\psi(x^t) - \nabla\psi(x^{t+1}), x^{t+1} - y \rangle = D_\psi(y, x^t) - D_\psi(y, x^{t+1}) - D_\psi(x^{t+1}, x^t)$ and $\langle h^{t+1}, y - x^{t+1} \rangle \leq h(y) - h(x_{t+1}) - \mu_h D_\psi(y, x^{t+1})$ by the $\mu_h$-strong convexity of $h$. We substitute $y$ with $z^t$ to obtain

$$\text{II} \leq \frac{D_\psi(z^t, x^t) - D_\psi(z^t, x^{t+1}) - D_\psi(x^{t+1}, x^t)}{\eta_t} + h(z_t) - h(x_{t+1}) - \mu_h D_\psi(z^t, x^{t+1}). \tag{3}$$

- For term III, we simply use the $\mu_f$-strong convexity of $f$ to get

$$\text{III} \leq f(z^t) - f(x^t) - \mu_f D_\psi(z^t, x^t). \tag{4}$$

- For term IV, we have

$$\text{IV} \leq LD_\psi(x^{t+1}, x^t) + M\sqrt{2D_\psi(x^{t+1}, x^t)}$$

$$\leq LD_\psi(x^{t+1}, x^t) + 2\eta_t M^2 + \frac{D_\psi(x^{t+1}, x^t)}{4\eta_t}, \tag{5}$$

where the first inequality holds by the 1-strong convexity of $\psi$ again and the second one is due to AM-GM inequality.

By plugging the bounds (2), (3), (4) and (5) into (1), we obtain

$$f(x^{t+1}) - f(x^t)$$
$$\leq \langle \xi^t, z^t - x^t \rangle + 2\eta_t \|\xi^t\|_*^2 + \frac{D_\psi(x^{t+1}, x^t)}{4\eta_t}$$
$$+ \frac{D_\psi(z^t, x^t) - D_\psi(z^t, x^{t+1}) - D_\psi(x^{t+1}, x^t)}{\eta_t} + h(z_t) - h(x_{t+1}) - \mu_h D_\psi(z^t, x^{t+1})$$
$$+ f(z^t) - f(x^t) - \mu_f D_\psi(z^t, x^t) + LD_\psi(x^{t+1}, x^t) + 2\eta_t M^2 + \frac{D_\psi(x^{t+1}, x^t)}{4\eta_t}.$$

Rearranging the terms to get

$$F(x^{t+1}) - F(z^t)$$
$$\leq \langle \xi^t, z^t - x^t \rangle + (\eta_t^{-1} - \mu_f) D_\psi(z^t, x^t) - (\eta_t^{-1} + \mu_h) D_\psi(z^t, x^{t+1})$$
$$+ \left(L - \frac{1}{2\eta_t}\right) D_\psi(x^{t+1}, x^t) + 2\eta_t(M^2 + \|\xi^t\|_*^2)$$
$$\overset{(a)}{\leq} \langle \xi^t, z^t - x^t \rangle + (\eta_t^{-1} - \mu_f) D_\psi(z^t, x^t) - (\eta_t^{-1} + \mu_h) D_\psi(z^t, x^{t+1}) + 2\eta_t(M^2 + \|\xi^t\|_*^2)$$
$$\overset{(b)}{=} \frac{v_{t-1}}{v_t} \langle \xi^t, z^{t-1} - x^t \rangle + (\eta_t^{-1} - \mu_f) D_\psi(z^t, x^t) - (\eta_t^{-1} + \mu_h) D_\psi(z^t, x^{t+1}) + 2\eta_t(M^2 + \|\xi^t\|_*^2)$$
$$\overset{(c)}{\leq} \frac{v_{t-1}}{v_t} \langle \xi^t, z^{t-1} - x^t \rangle + (\eta_t^{-1} - \mu_f) \frac{v_{t-1}}{v_t} D_\psi(z^{t-1}, x^t) - (\eta_t^{-1} + \mu_h) D_\psi(z^t, x^{t+1}) + 2\eta_t(M^2 + \|\xi^t\|_*^2), \tag{6}$$

where $(a)$ is by $\eta_t \leq \frac{1}{2L \vee \mu_f} \leq \frac{1}{2L}, \forall t \in [T] \Rightarrow L - \frac{1}{2\eta_t} \leq 0$, $(b)$ holds due to the definition of $z^t = \left(1 - \frac{v_{t-1}}{v_t}\right) x^t + \frac{v_{t-1}}{v_t} z^{t-1}$ implying $z^t - x^t = \frac{v_{t-1}}{v_t}\left(z^{t-1} - x^t\right)$, $(c)$ is by noticing $\eta_t \leq \frac{1}{2L \vee \mu_f} \leq \frac{1}{\mu_f}, \forall t \in [T] \Rightarrow \eta_t^{-1} - \mu_f \geq 0$ and

$$D_\psi(z^t, x^t) \overset{(d)}{\leq} \left(1 - \frac{v_{t-1}}{v_t}\right) D_\psi(x^t, x^t) + \frac{v_{t-1}}{v_t} D_\psi(z^{t-1}, x^t) = \frac{v_{t-1}}{v_t} D_\psi(z^{t-1}, x^t)$$

where $(d)$ is by the convexity of the first argument in $D_\psi(\cdot, \cdot)$.

Multiplying both sides of (6) by $w_t\gamma_t v_t$ (all of these three terms are non-negative) and summing up from $t = 1$ to $T$, we obtain

$$\sum_{t=1}^{T} w_t\gamma_t v_t \left(F(x^{t+1}) - F(z^t)\right)$$

$$\leq \sum_{t=1}^{T} w_t\gamma_t v_{t-1}\langle\xi^t, z^{t-1} - x^t\rangle + \sum_{t=1}^{T} 2w_t\gamma_t\eta_t v_t(M^2 + \|\xi^t\|_*^2)$$

$$+ \sum_{t=1}^{T} w_t\gamma_t(\eta_t^{-1} - \mu_f)v_{t-1}D_\psi(z^{t-1}, x^t) - w_t\gamma_t(\eta_t^{-1} + \mu_h)v_t D_\psi(z^t, x^{t+1})$$

$$= w_1\gamma_1(\eta_1^{-1} - \mu_f)v_0 D_\psi(z^0, x^1) - w_T\gamma_T(\eta_T^{-1} + \mu_h)v_T D_\psi(z^T, x^{T+1}) + \sum_{t=1}^{T} 2w_t\gamma_t\eta_t v_t(M^2 + \|\xi^t\|_*^2)$$

$$+ \sum_{t=1}^{T} w_t\gamma_t v_{t-1}\langle\xi^t, z^{t-1} - x^t\rangle + \sum_{t=2}^{T} \left(w_t\gamma_t(\eta_t^{-1} - \mu_f) - w_{t-1}\gamma_{t-1}(\eta_{t-1}^{-1} + \mu_h)\right)v_{t-1}D_\psi(z^{t-1}, x^t)$$

$$\overset{(e)}{=} w_1(1 - \mu_f\eta_1)v_0 D_\psi(x^*, x^1) - w_T\gamma_T(\eta_T^{-1} + \mu_h)v_T D_\psi(z^T, x^{T+1}) + \sum_{t=1}^{T} 2w_t\gamma_t\eta_t v_t(M^2 + \|\xi^t\|_*^2)$$

$$+ \sum_{t=1}^{T} w_t\gamma_t v_{t-1}\langle\xi^t, z^{t-1} - x^t\rangle + \sum_{t=2}^{T}(w_t - w_{t-1})\gamma_t(\eta_t^{-1} - \mu_f)v_{t-1}D_\psi(z^{t-1}, x^t), \qquad (7)$$

where $(e)$ holds due to $\gamma_1(\eta_1^{-1} - \mu_f) = \eta_1(\eta_1^{-1} - \mu_f) = 1 - \mu_f\eta_1$, $z^0 = x^*$ and $\gamma_t(\eta_t^{-1} - \mu_f) = \eta_t(\eta_t^{-1} - \mu_f)\prod_{s=2}^{t}\frac{1+\mu_h\eta_{s-1}}{1-\mu_f\eta_s} = (\eta_{t-1}^{-1} + \mu_h)\eta_{t-1}\prod_{s=2}^{t-1}\frac{1+\mu_h\eta_{s-1}}{1-\mu_f\eta_s} = \gamma_{t-1}(\eta_{t-1}^{-1} + \mu_h), \forall t \geq 2$.

By the convexity of $F$ and the definition of $z^t = \frac{v_0}{v_t}x^* + \sum_{s=1}^{t}\frac{v_s - v_{s-1}}{v_t}x^s$ (which means $z^t$ is a convex combination of $x^*, x^1, \cdots, x^t$ by noticing that the weights are summed up to 1 and nonnegative since $v_t, \forall t \in \{0\} \cup [T]$ is non-decreasing), we have

$$F(z^t) \leq \sum_{s=1}^{t}\frac{v_s - v_{s-1}}{v_t}F(x^s) + \frac{v_0}{v_t}F(x^*),$$

which implies

$$\sum_{t=1}^{T} w_t\gamma_t v_t \left(F(x^{t+1}) - F(z^t)\right)$$

$$\geq \sum_{t=1}^{T}\left[w_t\gamma_t v_t F(x^{t+1}) - w_t\gamma_t\left(\sum_{s=1}^{t}(v_s - v_{s-1})F(x^s) + v_0 F(x^*)\right)\right]$$

$$= \sum_{t=1}^{T}\left[w_t\gamma_t v_t\left(F(x^{t+1}) - F(x^*)\right) - w_t\gamma_t\sum_{s=1}^{t}(v_s - v_{s-1})\left(F(x^s) - F(x^*)\right)\right]$$

$$= w_T\gamma_T v_T\left(F(x^{T+1}) - F(x^*)\right) - \left(\sum_{t=1}^{T} w_t\gamma_t\right)(v_1 - v_0)\left(F(x^1) - F(x^*)\right)$$

$$+ \sum_{t=2}^{T}\left[w_{t-1}\gamma_{t-1}v_{t-1} - \left(\sum_{s=t}^{T} w_s\gamma_s\right)(v_t - v_{t-1})\right]\left(F(x^t) - F(x^*)\right).$$

Now by the definition of $v_t = \frac{w_T\gamma_T}{\sum_{s=t}^{T} w_s\gamma_s}, \forall t \in [T]$ and $v_0 = v_1$, we observe that

$$\left(\sum_{t=1}^{T} w_t\gamma_t\right)(v_1 - v_0) = 0,$$

and for $2 \leq t \leq T$,

$$w_{t-1}\gamma_{t-1}v_{t-1} - \left(\sum_{s=t}^{T} w_s\gamma_s\right)(v_t - v_{t-1})$$

$$= \left(\sum_{s=t-1}^{T} w_s\gamma_s\right)v_{t-1} - \left(\sum_{s=t}^{T} w_s\gamma_s\right)v_t$$

$$= \left(\sum_{s=t-1}^{T} w_s\gamma_s\right)\frac{w_T\gamma_T}{\sum_{s=t-1}^{T} w_s\gamma_s} - \left(\sum_{s=t}^{T} w_s\gamma_s\right)\frac{w_T\gamma_T}{\sum_{s=t}^{T} w_s\gamma_s}$$

$$= 0.$$

These two equations immediately imply

$$\sum_{t=1}^{T} w_t\gamma_t v_t \left(F(x^{t+1}) - F(z^t)\right) \geq w_T\gamma_T v_T \left(F(x^{T+1}) - F(x^*)\right). \tag{8}$$

Plugging (8) into (7), we finally get

$$w_T\gamma_T v_T \left(F(x^{T+1}) - F(x^*)\right)$$

$$\leq w_1(1 - \mu_f\eta_1)v_0 D_\psi(x^*, x^1) - w_T\gamma_T(\eta_T^{-1} + \mu_h)v_T D_\psi(z^T, x^{T+1}) + \sum_{t=1}^{T} 2w_t\gamma_t\eta_t v_t(M^2 + \|\xi^t\|_*^2)$$

$$+ \sum_{t=1}^{T} w_t\gamma_t v_{t-1}\langle\xi^t, z^{t-1} - x^t\rangle + \sum_{t=2}^{T}(w_t - w_{t-1})\gamma_t(\eta_t^{-1} - \mu_f)v_{t-1}D_\psi(z^{t-1}, x^t)$$

$$\leq w_1(1 - \mu_f\eta_1)v_0 D_\psi(x^*, x^1) + \sum_{t=1}^{T} 2w_t\gamma_t\eta_t v_t(M^2 + \|\xi^t\|_*^2)$$

$$+ \sum_{t=1}^{T} w_t\gamma_t v_{t-1}\langle\xi^t, z^{t-1} - x^t\rangle + \sum_{t=2}^{T}(w_t - w_{t-1})\gamma_t(\eta_t^{-1} - \mu_f)v_{t-1}D_\psi(z^{t-1}, x^t).$$

$\square$

## B.2 PROOF OF LEMMA 4.2

*Proof of Lemma 4.2.* We invoke Lemma 4.1 with $w_t = 1, \forall t \in [T]$ to get

$$\gamma_T v_T \left(F(x^{T+1}) - F(x^*)\right)$$

$$\leq (1 - \mu_f\eta_1)v_0 D_\psi(x^*, x^1) + \sum_{t=1}^{T} 2\gamma_t\eta_t v_t(M^2 + \|\xi^t\|_*^2) + \sum_{t=1}^{T} \gamma_t v_{t-1}\langle\xi^t, z^{t-1} - x^t\rangle.$$

Taking expectations on both sides to obtain

$$\gamma_T v_T \mathbb{E}\left[F(x^{T+1}) - F(x^*)\right]$$

$$\leq (1 - \mu_f\eta_1)v_0 D_\psi(x^*, x^1) + \sum_{t=1}^{T} 2\gamma_t\eta_t v_t(M^2 + \mathbb{E}\left[\|\xi^t\|_*^2\right]) + \sum_{t=1}^{T} \gamma_t v_{t-1}\mathbb{E}\left[\langle\xi^t, z^{t-1} - x^t\rangle\right]$$

$$\leq (1 - \mu_f\eta_1)v_0 D_\psi(x^*, x^1) + \sum_{t=1}^{T} 2\gamma_t\eta_t v_t(M^2 + \sigma^2),$$

where the last line is due to $\mathbb{E}\left[\|\xi^t\|_*^2\right] = \mathbb{E}\left[\mathbb{E}\left[\|\xi^t\|_*^2 \mid \mathcal{F}^{t-1}\right]\right] \leq \sigma^2$ (Assumption 5A) and $\mathbb{E}\left[\langle\xi^t, z^{t-1} - x^t\rangle\right] = \mathbb{E}\left[\langle\mathbb{E}\left[\xi^t | \mathcal{F}^{t-1}\right], z^{t-1} - x^t\rangle\right] = 0$ ($z^{t-1} - x^t \in \mathcal{F}^{t-1} = \sigma(\widehat{g}^s, s \in [t-1])$ and Assumption 4). Finally, we divide both sides by $\gamma_T v_T$ and plug in $v_t = \frac{w_T\gamma_T}{\sum_{s=t}^{T} w_s\gamma_s} = \frac{\gamma_T}{\sum_{s=t}^{T} \gamma_s}, \forall t \in [T]$ and $v_0 = v_1$ to finish the proof. $\square$

## B.3 Proof of Lemma 4.3

*Proof of Lemma 4.3.* We invoke Lemma 4.1 to get

$$w_T \gamma_T v_T \left( F(x^{T+1}) - F(x^*) \right)$$

$$\leq w_1 (1 - \mu_f \eta_1) v_0 D_\psi(x^*, x^1) + \sum_{t=1}^T 2 w_t \gamma_t \eta_t v_t (M^2 + \|\xi^t\|_*^2)$$

$$+ \sum_{t=1}^T w_t \gamma_t v_{t-1} \langle \xi^t, z^{t-1} - x^t \rangle + \sum_{t=2}^T (w_t - w_{t-1}) \gamma_t (\eta_t^{-1} - \mu_f) v_{t-1} D_\psi(z^{t-1}, x^t). \quad (9)$$

Let $w_t, \forall t \in [T]$ be defined as follows (note that $w_1$ is also well-defined as $w_1 = \frac{1}{\sum_{s=1}^T 2\gamma_s \eta_s \bar{v}_s \sigma^2}$)

$$w_t := \frac{1}{\sum_{s=2}^t \frac{2\gamma_s \eta_s \bar{v}_s \sigma^2}{1 - \mu_f \eta_s} + \sum_{s=1}^T 2\gamma_s \eta_s \bar{v}_s \sigma^2}, \forall t \in [T], \quad (10)$$

where

$$\bar{v}_t := \frac{\gamma_T}{\sum_{s=t}^T \gamma_s}, \forall t \in [T] \text{ and } \bar{v}_1 := \bar{v}_0. \quad (11)$$

Note that $w_t \geq 0, \forall t \in [T]$ is non-increasing, from the definition of $v_t := \frac{w_T \gamma_T}{\sum_{s=t}^T w_s \gamma_s}, \forall t \in [T]$ and $v_0 := v_1$, there are always

$$v_t = \frac{w_T \gamma_T}{\sum_{s=t}^T w_s \gamma_s} \leq \frac{\gamma_T}{\sum_{s=t}^T \gamma_s} = \bar{v}_t, \forall t \in [T] \text{ and } v_0 \leq \bar{v}_0. \quad (12)$$

Now we consider the following non-negative sequence with $U_0 := 1$ and

$$U_s := \exp \left( \sum_{t=1}^s 2 w_t \gamma_t \eta_t v_t \|\xi^t\|_*^2 - 2 w_t \gamma_t \eta_t v_t \sigma^2 \right) \in \mathcal{F}^s, \forall s \in [T].$$

We claim $U_t$ is a supermartingale by observing that

$$\mathbb{E}\left[ U_t \mid \mathcal{F}^{t-1} \right] = U_{t-1} \mathbb{E}\left[ \exp \left( 2 w_t \gamma_t \eta_t v_t \|\xi^t\|_*^2 - 2 w_t \gamma_t \eta_t v_t \sigma^2 \right) \mid \mathcal{F}^{t-1} \right]$$

$$\overset{(a)}{\leq} U_{t-1} \exp \left( 2 w_t \gamma_t \eta_t v_t \sigma^2 - 2 w_t \gamma_t \eta_t v_t \sigma^2 \right) = U_{t-1},$$

where $(a)$ holds due to Assumption 5B by noticing

$$2 w_t \gamma_t \eta_t v_t \overset{(10)}{=} \frac{2 \gamma_t \eta_t v_t}{\sum_{s=2}^t \frac{2\gamma_s \eta_s \bar{v}_s \sigma^2}{1 - \mu_f \eta_s} + \sum_{s=1}^T 2\gamma_s \eta_s \bar{v}_s \sigma^2} \leq \frac{v_t}{\bar{v}_t \sigma^2} \overset{(12)}{\leq} \frac{1}{\sigma^2}.$$

Hence, we know $\mathbb{E}[U_T] \leq U_0 = 1$. Thus, there is

$$\Pr\left[ U_T > \frac{2}{\delta} \right] \overset{(b)}{\leq} \frac{\delta}{2} \mathbb{E}[U_T] \leq \frac{\delta}{2}$$

$$\Rightarrow \Pr\left[ \sum_{t=1}^T 2 w_t \gamma_t \eta_t v_t \|\xi^t\|_*^2 \leq \sum_{t=1}^T 2 w_t \gamma_t \eta_t v_t \sigma^2 + \log \frac{2}{\delta} \right] \geq 1 - \frac{\delta}{2}, \quad (13)$$

where we use Markov's inequality in $(b)$.

Next, we consider another non-negative sequence with $R_0 := 1$ and

$$R_s := \exp \left( \sum_{t=1}^s w_t \gamma_t v_{t-1} \langle \xi^t, z^{t-1} - x^t \rangle - w_t^2 \gamma_t^2 v_{t-1}^2 \sigma^2 \|z^{t-1} - x^t\|^2 \right) \in \mathcal{F}^s, \forall s \in [T].$$

We prove that $R_t$ is also a supermartingale by

$$\mathbb{E}\left[ R_t \mid \mathcal{F}^{t-1} \right] = R_{t-1} \mathbb{E}\left[ \exp \left( w_t \gamma_t v_{t-1} \langle \xi^t, z^{t-1} - x^t \rangle - w_t^2 \gamma_t^2 v_{t-1}^2 \sigma^2 \|z^{t-1} - x^t\|^2 \right) \mid \mathcal{F}^{t-1} \right]$$

$$\overset{(c)}{\leq} R_{t-1} \exp \left( w_t^2 \gamma_t^2 v_{t-1}^2 \sigma^2 \|z^{t-1} - x^t\|^2 - w_t^2 \gamma_t^2 v_{t-1}^2 \sigma^2 \|z^{t-1} - x^t\|^2 \right) = R_{t-1},$$

where $(c)$ is by applying Lemma 2.1 (note that $z^{t-1} - x^t \in \mathcal{F}^{t-1} = \sigma(\widehat{g}^s, s \in [t-1])$). Hence, we have $\mathbb{E}[R_T] \leq R_0 = 1$, which immediately implies

$$\Pr\left[R_T > \frac{2}{\delta}\right] \overset{(d)}{\leq} \frac{\delta}{2}\mathbb{E}[R_T] \leq \frac{\delta}{2}$$

$$\Rightarrow \Pr\left[\sum_{t=1}^T w_t\gamma_t v_{t-1}\langle \xi^t, z^{t-1} - x^t\rangle \leq \sum_{t=1}^T w_t^2\gamma_t^2 v_{t-1}^2\sigma^2\|z^{t-1} - x^t\|^2 + \log\frac{2}{\delta}\right] \geq 1 - \frac{\delta}{2}$$

$$\Rightarrow \Pr\left[\sum_{t=1}^T w_t\gamma_t v_{t-1}\langle \xi^t, z^{t-1} - x^t\rangle \leq \sum_{t=1}^T 2w_t^2\gamma_t^2 v_{t-1}^2\sigma^2 D_\psi(z^{t-1}, x^t) + \log\frac{2}{\delta}\right] \geq 1 - \frac{\delta}{2}, \quad (14)$$

where $(d)$ is by Markov's inequality and the last line is due to $\|z^{t-1} - x^t\|^2 \leq 2D_\psi(z^{t-1}, x^t)$ from the 1-strong convexity of $\psi$.

Combining (9), (13) and (14), with probability at least $1 - \delta$, there is

$$w_T\gamma_T v_T\left(F(x^{T+1}) - F(x^*)\right)$$

$$\leq w_1(1 - \mu_f\eta_1)v_0 D_\psi(x^*, x^1) + 2\log\frac{2}{\delta} + \sum_{t=1}^T 2w_t\gamma_t\eta_t v_t(M^2 + \sigma^2)$$

$$+ \sum_{t=1}^T 2w_t^2\gamma_t^2 v_{t-1}^2\sigma^2 D_\psi(z^{t-1}, x^t) + \sum_{t=2}^T (w_t - w_{t-1})\gamma_t(\eta_t^{-1} - \mu_f)v_{t-1}D_\psi(z^{t-1}, x^t)$$

$$= \left[w_1(1 - \mu_f\eta_1)v_0 + 2w_1^2\gamma_1^2 v_0^2\sigma^2\right] D_\psi(x^*, x^1) + 2\log\frac{2}{\delta} + \sum_{t=1}^T 2w_t\gamma_t\eta_t v_t(M^2 + \sigma^2)$$

$$+ \sum_{t=2}^T \left[(w_t - w_{t-1})(\eta_t^{-1} - \mu_f) + 2w_t^2\gamma_t v_{t-1}\sigma^2\right]\gamma_t v_{t-1}D_\psi(z^{t-1}, x^t).$$

Observing that for $t \geq 2$

$$(w_t - w_{t-1})(\eta_t^{-1} - \mu_f) + 2w_t^2\gamma_t v_{t-1}\sigma^2$$

$$= 2w_t^2\gamma_t v_{t-1}\sigma^2$$

$$- \left(\frac{1}{\sum_{s=2}^{t-1} \frac{2\gamma_s\eta_s\bar{v}_s\sigma^2}{1 - \mu_f\eta_s} + \sum_{s=1}^T 2\gamma_s\eta_s\bar{v}_s\sigma^2} - \frac{1}{\sum_{s=2}^t \frac{2\gamma_s\eta_s\bar{v}_s\sigma^2}{1 - \mu_f\eta_s} + \sum_{s=1}^T 2\gamma_s\eta_s\bar{v}_s\sigma^2}\right)(\eta_t^{-1} - \mu_f)$$

$$= 2w_t^2\gamma_t v_{t-1}\sigma^2 - w_t w_{t-1} \times \frac{2\gamma_t\eta_t\bar{v}_t\sigma^2}{1 - \mu_f\eta_t} \times (\eta_t^{-1} - \mu_f)$$

$$= 2w_t(w_t v_{t-1} - w_{t-1}\bar{v}_t)\gamma_t\sigma^2 \leq 0,$$

where the last line holds due to $w_t \leq w_{t-1}$ and $v_{t-1} \leq v_t \leq \bar{v}_t$. So we know

$$w_T\gamma_T v_T\left(F(x^{T+1}) - F(x^*)\right)$$

$$\leq \left[w_1(1 - \mu_f\eta_1)v_0 + 2w_1^2\gamma_1^2 v_0^2\sigma^2\right] D_\psi(x^*, x^1) + 2\log\frac{2}{\delta} + \sum_{t=1}^T 2w_t\gamma_t\eta_t v_t(M^2 + \sigma^2)$$

$$\overset{(e)}{\leq} w_1\left(1 - \mu_f\eta_1 + 2w_1\gamma_1^2 v_0\sigma^2\right)v_0 D_\psi(x^*, x^1) + 2\log\frac{2}{\delta} + w_1\sum_{t=1}^T 2\gamma_t\eta_t v_t(M^2 + \sigma^2),$$

where $(e)$ is by $w_t \leq w_1, \forall t \in [T]$. Dividing both sides by $w_T \gamma_T v_T$, we get

$$F(x^{T+1}) - F(x^*)$$

$$\leq \frac{w_1}{w_T} \left[ (1 - \mu_f \eta_1 + 2w_1 \gamma_1^2 v_0 \sigma^2) \frac{v_0}{\gamma_T v_T} D_\psi(x^*, x^1) + \frac{2}{w_1 \gamma_T v_T} \log \frac{2}{\delta} + 2 \sum_{t=1}^T \frac{\gamma_t \eta_t v_t}{\gamma_T v_T} (M^2 + \sigma^2) \right]$$

$$\overset{(f)}{\leq} \left( 1 + \max_{2 \leq t \leq T} \frac{1}{1 - \mu_f \eta_t} \right) \left[ \frac{(2 - \mu_f \eta_1) D_\psi(x^*, x^1)}{\sum_{t=1}^T \gamma_t} + 2 \left( M^2 + \sigma^2 \left( 1 + 2 \log \frac{2}{\delta} \right) \right) \sum_{t=1}^T \frac{\gamma_t \eta_t}{\sum_{s=t}^T \gamma_s} \right]$$

$$\leq 2 \left( 1 + \max_{2 \leq t \leq T} \frac{1}{1 - \mu_f \eta_t} \right) \left[ \frac{D_\psi(x^*, x^1)}{\sum_{t=1}^T \gamma_t} + \left( M^2 + \sigma^2 \left( 1 + 2 \log \frac{2}{\delta} \right) \right) \sum_{t=1}^T \frac{\gamma_t \eta_t}{\sum_{s=t}^T \gamma_s} \right],$$

where $(f)$ holds due to the following calculations

$$\frac{w_1}{w_T} = \frac{\sum_{s=1}^T 2\gamma_s \eta_s \bar{v}_s \sigma^2 + \sum_{s=2}^T \frac{2\gamma_s \eta_s \bar{v}_s \sigma^2}{1 - \mu_f \eta_s}}{\sum_{s=1}^T 2\gamma_s \eta_s \bar{v}_s \sigma^2} \leq 1 + \max_{2 \leq t \leq T} \frac{1}{1 - \mu_f \eta_t};$$

$$2w_1 \gamma_1^2 v_0 \sigma^2 = \frac{2\gamma_1^2 v_0 \sigma^2}{\sum_{s=1}^T 2\gamma_s \eta_s \bar{v}_s \sigma^2} = \frac{2\gamma_1 \eta_1 v_1 \sigma^2}{\sum_{s=1}^T 2\gamma_s \eta_s \bar{v}_s \sigma^2} \leq 1;$$

$$\frac{v_0}{\gamma_T v_T} \leq \frac{\bar{v}_0}{\gamma_T v_T} = \frac{1}{\sum_{t=1}^T \gamma_t};$$

$$\frac{2}{w_1 \gamma_T v_T} \log \frac{2}{\delta} = 4\sigma^2 \log \frac{2}{\delta} \sum_{t=1}^T \frac{\gamma_t \eta_t \bar{v}_t}{\gamma_T v_T} = 4\sigma^2 \log \frac{2}{\delta} \sum_{t=1}^T \frac{\gamma_t \eta_t}{\sum_{s=t}^T \gamma_s};$$

$$2 \sum_{t=1}^T \frac{\gamma_t \eta_t v_t}{\gamma_T v_T} (M^2 + \sigma^2) \leq 2(M^2 + \sigma^2) \sum_{t=1}^T \frac{\gamma_t \eta_t \bar{v}_t}{\gamma_T v_T} = 2(M^2 + \sigma^2) \sum_{t=1}^T \frac{\gamma_t \eta_t}{\sum_{s=t}^T \gamma_s}.$$

Hence, the proof is completed. $\qquad \square$

## C GENERAL CONVEX FUNCTIONS

In this section, we present the full version of theorems for general convex functions (i.e., $\mu_f = \mu_h = 0$) with their proofs.

**Theorem C.1.** *Under Assumptions 1-4 and 5A with $\mu_f = \mu_h = 0$:*

*If $T$ is unknown, by taking $\eta_t = \frac{1}{2L} \wedge \frac{\eta}{\sqrt{t}}, \forall t \in [T]$, there is*

$$\mathbb{E} \left[ F(x^{T+1}) - F(x^*) \right] \leq O \left( \frac{L D_\psi(x^*, x^1)}{T} + \frac{1}{\sqrt{T}} \left[ \frac{D_\psi(x^*, x^1)}{\eta} + \eta (M^2 + \sigma^2) \log T \right] \right).$$

*In particular, by choosing $\eta = \Theta \left( \sqrt{\frac{D_\psi(x^*, x^1)}{M^2 + \sigma^2}} \right)$, there is*

$$\mathbb{E} \left[ F(x^{T+1}) - F(x^*) \right] \leq O \left( \frac{L D_\psi(x^*, x^1)}{T} + \frac{(M + \sigma)\sqrt{D_\psi(x^*, x^1)} \log T}{\sqrt{T}} \right).$$

*If $T$ is known, by taking $\eta_t = \frac{1}{2L} \wedge \frac{\eta}{\sqrt{T}}, \forall t \in [T]$, there is*

$$\mathbb{E} \left[ F(x^{T+1}) - F(x^*) \right] \leq O \left( \frac{L D_\psi(x^*, x^1)}{T} + \frac{1}{\sqrt{T}} \left[ \frac{D_\psi(x^*, x^1)}{\eta} + \eta (M^2 + \sigma^2) \log T \right] \right).$$

*In particular, by choosing $\eta = \Theta \left( \sqrt{\frac{D_\psi(x^*, x^1)}{(M^2 + \sigma^2) \log T}} \right)$, there is*

$$\mathbb{E} \left[ F(x^{T+1}) - F(x^*) \right] \leq O \left( \frac{L D_\psi(x^*, x^1)}{T} + \frac{(M + \sigma)\sqrt{D_\psi(x^*, x^1) \log T}}{\sqrt{T}} \right).$$

*Proof.* From Lemma 4.2, if $\eta_t \leq \frac{1}{2L \vee \mu_f}, \forall t \in [T]$, there is

$$\mathbb{E}\left[F(x^{T+1}) - F(x^*)\right] \leq \frac{(1 - \mu_f \eta_1) D_\psi(x^*, x^1)}{\sum_{t=1}^T \gamma_t} + 2(M^2 + \sigma^2) \sum_{t=1}^T \frac{\gamma_t \eta_t}{\sum_{s=t}^T \gamma_s}, \tag{15}$$

where $\gamma_t := \eta_t \prod_{s=2}^t \frac{1 + \mu_h \eta_{s-1}}{1 - \mu_f \eta_s}, \forall t \in [T]$. Note that $\mu_f = \mu_h = 0$ now, so both $\eta_t = \frac{1}{2L} \wedge \frac{\eta}{\sqrt{t}}, \forall t \in [T]$ and $\eta_t = \frac{1}{2L} \wedge \frac{\eta}{\sqrt{T}}, \forall t \in [T]$ satisfy $\eta_t \leq \frac{1}{2L \vee \mu_f} = \frac{1}{2L}, \forall t \in [T]$. Besides, $\gamma_t$ will degenerate to $\eta_t$. Therefore, (15) can be simplified into

$$\mathbb{E}\left[F(x^{T+1}) - F(x^*)\right] \leq \frac{D_\psi(x^*, x^1)}{\sum_{t=1}^T \eta_t} + 2(M^2 + \sigma^2) \sum_{t=1}^T \frac{\eta_t^2}{\sum_{s=t}^T \eta_s}. \tag{16}$$

Before proving convergence rates for these two different step sizes, we first recall some standard results.

$$\sum_{t=1}^T \frac{1}{\sqrt{t}} = \sum_{t=1}^T \sqrt{t} - \frac{t-1}{\sqrt{t}} = \sqrt{T} + \sum_{t=1}^{T-1} \sqrt{t} - \frac{t}{\sqrt{t+1}} \geq \sqrt{T}; \tag{17}$$

$$\sum_{s=t}^T \frac{1}{\sqrt{s}} \geq \int_t^{T+1} \frac{1}{\sqrt{s}} ds = 2(\sqrt{T+1} - \sqrt{t}), \forall t \in [T]; \tag{18}$$

$$\sum_{t=1}^T \frac{1}{t} \leq 1 + \int_1^T \frac{1}{t} dt = 1 + \log T. \tag{19}$$

If $\eta_t = \frac{1}{2L} \wedge \frac{\eta}{\sqrt{t}}, \forall t \in [T]$, we consider the following three cases:

- $\eta < \frac{1}{2L}$: In this case, we have $\eta_t = \frac{\eta}{\sqrt{t}}, \forall t \in [T]$ and

$$
\begin{aligned}
\mathbb{E}\left[F(x^{T+1}) - F(x^*)\right] &\leq \frac{D_\psi(x^*, x^1)}{\eta \sum_{t=1}^T 1/\sqrt{t}} + 2\eta(M^2 + \sigma^2) \sum_{t=1}^T \frac{1}{t \sum_{s=t}^T 1/\sqrt{s}} \\
&\overset{(17),(18)}{\leq} \frac{D_\psi(x^*, x^1)}{\eta \sqrt{T}} + \eta(M^2 + \sigma^2) \sum_{t=1}^T \frac{1}{t(\sqrt{T+1} - \sqrt{t})} \\
&\overset{(a)}{\leq} \frac{D_\psi(x^*, x^1)}{\eta \sqrt{T}} + \frac{4\eta(M^2 + \sigma^2)(1 + \log T)}{\sqrt{T}} \\
&= \frac{1}{\sqrt{T}}\left[\frac{D_\psi(x^*, x^1)}{\eta} + 4\eta(M^2 + \sigma^2)(1 + \log T)\right], \tag{20}
\end{aligned}
$$

where $(a)$ is by

$$
\begin{aligned}
\sum_{t=1}^T \frac{1}{t(\sqrt{T+1} - \sqrt{t})} &= \sum_{t=1}^T \frac{\sqrt{T+1} + \sqrt{t}}{t(T+1-t)} \leq \sum_{t=1}^T \frac{2\sqrt{T+1}}{t(T+1-t)} \\
&= \sum_{t=1}^T \frac{2}{\sqrt{T+1}}\left(\frac{1}{t} + \frac{1}{T+1-t}\right) = \frac{4}{\sqrt{T+1}} \sum_{t=1}^T \frac{1}{t} \\
&\overset{(19)}{\leq} \frac{4(1 + \log T)}{\sqrt{T}}.
\end{aligned}
$$

- $\eta \geq \frac{\sqrt{T}}{2L}$: In this case, we have $\eta_t = \frac{1}{2L}, \forall t \in [T]$ and

$$
\mathbb{E}\left[F(x^{T+1}) - F(x^*)\right] \leq \frac{D_\psi(x^*, x^1)}{T/2L} + \frac{M^2 + \sigma^2}{L} \sum_{t=1}^{T} \frac{1}{T - t + 1}
$$

$$
= \frac{2LD_\psi(x^*, x^1)}{T} + \frac{M^2 + \sigma^2}{L} \sum_{t=1}^{T} \frac{1}{t}
$$

$$
\overset{(19)}{\leq} \frac{2LD_\psi(x^*, x^1)}{T} + \frac{(M^2 + \sigma^2)(1 + \log T)}{L}
$$

$$
\overset{(b)}{\leq} \frac{2LD_\psi(x^*, x^1)}{T} + \frac{2\eta(M^2 + \sigma^2)(1 + \log T)}{\sqrt{T}}, \qquad (21)
$$

where $(b)$ is by $\frac{1}{L} \leq \frac{2\eta}{\sqrt{T}}$.

- $\eta \in [\frac{1}{2L}, \frac{\sqrt{T}}{2L})$: In this case, we define $\tau = \lfloor 4\eta^2 L^2 \rfloor$ where $\lfloor \cdot \rfloor$ is the floor function. Note that

$$
4\eta^2 L^2 \in [1, T) \Rightarrow \tau = \lfloor 4\eta^2 L^2 \rfloor \in [T - 1].
$$

By observing $\frac{\eta}{\sqrt{t}} \geq \frac{1}{2L} \Leftrightarrow t \in [1, \tau]$, we can calculate

$$
\mathbb{E}\left[F(x^{T+1}) - F(x^*)\right] \leq \frac{D_\psi(x^*, x^1)}{\sum_{t=1}^{T} \eta_t} + 2(M^2 + \sigma^2) \sum_{t=1}^{T} \frac{\eta_t^2}{\sum_{s=t}^{T} \eta_s}
$$

$$
\overset{(c)}{\leq} \frac{D_\psi(x^*, x^1)}{T^2} \underbrace{\sum_{t=1}^{T} \frac{1}{\eta_t}}_{\text{I}} + 2(M^2 + \sigma^2) \left( \underbrace{\sum_{t=1}^{\tau} \frac{\eta_t^2}{\sum_{s=t}^{T} \eta_s}}_{\text{II}} + \underbrace{\sum_{t=\tau+1}^{T} \frac{\eta_t^2}{\sum_{s=t}^{T} \eta_s}}_{\text{III}} \right),
$$

where $(c)$ is by $T^2 \leq \left(\sum_{t=1}^{T} \eta_t\right)\left(\sum_{t=1}^{T} \frac{1}{\eta_t}\right)$. Now we bound terms I, II and III as follows

$$
\text{I} = \sum_{t=1}^{T} 2L \vee \frac{\sqrt{t}}{\eta} \leq \sum_{t=1}^{T} 2L + \frac{\sqrt{t}}{\eta} \leq 2LT + \frac{\sqrt{T} + \int_1^T \sqrt{t}\,dt}{\eta}
$$

$$
= 2LT + \frac{\sqrt{T} + \frac{2}{3}(T^{\frac{3}{2}} - 1)}{\eta} \leq 2LT + \frac{5T^{\frac{3}{2}}}{3\eta};
$$

$$
\text{II} = \sum_{t=1}^{\tau} \frac{\eta_t^2}{\sum_{s=t}^{\tau} \eta_s + \sum_{s=\tau+1}^{T} \eta_s} = \sum_{t=1}^{\tau} \frac{1/(4L^2)}{(\tau - t + 1)/2L + \sum_{s=\tau+1}^{T} \eta/\sqrt{s}}
$$

$$
= \frac{1}{2L} \sum_{t=1}^{\tau} \frac{1}{\tau - t + 1 + \sum_{s=\tau+1}^{T} 2\eta L/\sqrt{s}}
$$

$$
\overset{(18)}{\leq} \frac{1}{2L} \sum_{t=1}^{\tau} \frac{1}{\tau - t + 1 + 4\eta L(\sqrt{T+1} - \sqrt{\tau+1})}
$$

$$
\leq \begin{cases} \frac{1}{2L} \sum_{t=1}^{\tau} \frac{1}{\tau - t + 1} \leq \frac{1}{2L}\left(1 + \int_1^\tau \frac{1}{t}\,dt\right) = \frac{1 + \log \tau}{2L} \overset{(d)}{\leq} \frac{\eta(1 + \log T)}{\sqrt{\tau}} \\ \frac{1}{2L} \sum_{t=1}^{\tau} \frac{1}{4\eta L(\sqrt{T+1} - \sqrt{\tau+1})} = \frac{\tau}{8\eta L^2(\sqrt{T+1} - \sqrt{\tau+1})} \overset{(e)}{\leq} \frac{\eta}{2(\sqrt{T+1} - \sqrt{\tau+1})} \end{cases}
$$

$$
\Rightarrow \text{II} \leq \eta(1 + \log T)\left(\frac{1}{\sqrt{\tau}} \wedge \frac{1}{2(\sqrt{T+1} - \sqrt{\tau+1})}\right)
$$

$$
\leq \frac{2\eta(1 + \log T)}{\sqrt{\tau} + 2(\sqrt{T+1} - \sqrt{\tau+1})} \overset{(f)}{\leq} \frac{2\eta(1 + \log T)}{\sqrt{T}},
$$

where $(d)$ is due to $\tau \le T$ and $\sqrt{\tau} \le 2\eta L$, $(e)$ holds by $\tau \le 4\eta^2 L^2$ and $(f)$ is by, for $\tau \in [T-1]$ and $T \ge 2$,

$$\sqrt{\tau} + 2(\sqrt{T+1} - \sqrt{\tau+1}) \ge \sqrt{T-1} + 2\sqrt{T+1} - 2\sqrt{T} \ge \sqrt{T}.$$

$$\begin{aligned}
\text{III} &= \eta \sum_{t=\tau+1}^{T} \frac{1}{t \sum_{s=t}^{T} 1/\sqrt{s}} \overset{(18)}{\le} \eta \sum_{t=\tau+1}^{T} \frac{1}{2t(\sqrt{T+1} - \sqrt{t})} \\
&= \eta \sum_{t=\tau+1}^{T} \frac{\sqrt{T+1} + \sqrt{t}}{2t(T+1-t)} \le \eta \sum_{t=\tau+1}^{T} \frac{\sqrt{T+1}}{t(T+1-t)} \\
&= \eta \sum_{t=\tau+1}^{T} \frac{1}{\sqrt{T+1}} \left( \frac{1}{t} + \frac{1}{T+1-t} \right) \le \frac{2\eta}{\sqrt{T+1}} \sum_{t=1}^{T} \frac{1}{t} \\
&\overset{(19)}{\le} \frac{2\eta(1+\log T)}{\sqrt{T}}.
\end{aligned}$$

Thus, we have

$$\begin{aligned}
&\mathbb{E}\left[ F(x^{T+1}) - F(x^*) \right] \\
&\le \frac{D_\psi(x^*, x^1)}{T^2} \left( 2LT + \frac{5T^{\frac{3}{2}}}{3\eta} \right) + 2(M^2 + \sigma^2) \left[ \frac{2\eta(1+\log T)}{\sqrt{T}} + \frac{2\eta(1+\log T)}{\sqrt{T}} \right] \\
&\le \frac{2LD_\psi(x^*, x^1)}{T} + \frac{1}{\sqrt{T}} \left( \frac{5D_\psi(x^*, x^1)}{3\eta} + 8\eta(M^2 + \sigma^2)(1+\log T) \right). \quad (22)
\end{aligned}$$

Combining (20), (21) and (22), we know

$$\begin{aligned}
\mathbb{E}\left[ F(x^{T+1}) - F(x^*) \right] &\le \frac{1}{\sqrt{T}} \left[ \frac{D_\psi(x^*, x^1)}{\eta} + 4\eta(M^2 + \sigma^2)(1+\log T) \right] \\
&\vee \left[ \frac{2LD_\psi(x^*, x^1)}{T} + \frac{2\eta(M^2 + \sigma^2)(1+\log T)}{\sqrt{T}} \right] \\
&\vee \left[ \frac{2LD_\psi(x^*, x^1)}{T} + \frac{1}{\sqrt{T}} \left( \frac{5D_\psi(x^*, x^1)}{3\eta} + 8\eta(M^2 + \sigma^2)(1+\log T) \right) \right] \\
&\le \frac{2LD_\psi(x^*, x^1)}{T} + \frac{1}{\sqrt{T}} \left( \frac{5D_\psi(x^*, x^1)}{3\eta} + 8\eta(M^2 + \sigma^2)(1+\log T) \right) \\
&= O\left( \frac{LD_\psi(x^*, x^1)}{T} + \frac{1}{\sqrt{T}} \left[ \frac{D_\psi(x^*, x^1)}{\eta} + \eta(M^2 + \sigma^2)\log T \right] \right). \quad (23)
\end{aligned}$$

By plugging in $\eta = \Theta\left( \sqrt{\frac{D_\psi(x^*, x^1)}{M^2 + \sigma^2}} \right)$, we get the desired bound.

If $\eta_t = \frac{1}{2L} \wedge \frac{\eta}{\sqrt{T}}, \forall t \in [T]$, we will obtain

$$\begin{aligned}
\mathbb{E}\left[ F(x^{T+1}) - F(x^*) \right] &\le \frac{D_\psi(x^*, x^1)}{T} \left( 2L \vee \frac{\sqrt{T}}{\eta} \right) + 2\left( \frac{1}{2L} \wedge \frac{\eta}{\sqrt{T}} \right)(M^2 + \sigma^2) \sum_{t=1}^{T} \frac{1}{T-t+1} \\
&= \frac{D_\psi(x^*, x^1)}{T} \left( 2L \vee \frac{\sqrt{T}}{\eta} \right) + 2\left( \frac{1}{2L} \wedge \frac{\eta}{\sqrt{T}} \right)(M^2 + \sigma^2) \sum_{t=1}^{T} \frac{1}{t} \\
&\overset{(19)}{\le} \frac{D_\psi(x^*, x^1)}{T} \left( 2L \vee \frac{\sqrt{T}}{\eta} \right) + 2\left( \frac{1}{2L} \wedge \frac{\eta}{\sqrt{T}} \right)(M^2 + \sigma^2)(1+\log T) \\
&\le \frac{2LD_\psi(x^*, x^1)}{T} + \frac{1}{\sqrt{T}} \left[ \frac{D_\psi(x^*, x^1)}{\eta} + 2\eta(M^2 + \sigma^2)(1+\log T) \right] \\
&= O\left( \frac{LD_\psi(x^*, x^1)}{T} + \frac{1}{\sqrt{T}} \left[ \frac{D_\psi(x^*, x^1)}{\eta} + \eta(M^2 + \sigma^2)\log T \right] \right). \quad (24)
\end{aligned}$$

By plugging in $\eta = \Theta\left(\sqrt{\frac{D_\psi(x^*, x^1)}{(M^2+\sigma^2)\log T}}\right)$, we get the desired bound. $\qquad\square$

**Theorem C.2.** *Under Assumptions 1-4 and 5B with $\mu_f = \mu_h = 0$ and let $\delta \in (0,1)$:*

*If $T$ is unknown, by taking $\eta_t = \frac{1}{2L} \wedge \frac{\eta}{\sqrt{t}}, \forall t \in [T]$, then with probability at least $1 - \delta$, there is*

$$F(x^{T+1}) - F(x^*) \le O\left(\frac{LD_\psi(x^*, x^1)}{T} + \frac{1}{\sqrt{T}}\left[\frac{D_\psi(x^*, x^1)}{\eta} + \eta\left(M^2 + \sigma^2 \log \frac{1}{\delta}\right)\log T\right]\right).$$

*In particular, by choosing $\eta = \Theta\left(\sqrt{\frac{D_\psi(x^*, x^1)}{M^2 + \sigma^2 \log \frac{1}{\delta}}}\right)$, there is*

$$F(x^{T+1}) - F(x^*) \le O\left(\frac{LD_\psi(x^*, x^1)}{T} + \frac{(M + \sigma\sqrt{\log \frac{1}{\delta}})\sqrt{D_\psi(x^*, x^1)}\log T}{\sqrt{T}}\right).$$

*If $T$ is known, by taking $\eta_t = \frac{1}{2L} \wedge \frac{\eta}{\sqrt{T}}, \forall t \in [T]$, then with probability at least $1 - \delta$, there is*

$$F(x^{T+1}) - F(x^*) \le O\left(\frac{LD_\psi(x^*, x^1)}{T} + \frac{1}{\sqrt{T}}\left[\frac{D_\psi(x^*, x^1)}{\eta} + \eta\left(M^2 + \sigma^2 \log \frac{1}{\delta}\right)\log T\right]\right).$$

*In particular, by choosing $\eta = \Theta\left(\sqrt{\frac{D_\psi(x^*, x^1)}{(M^2 + \sigma^2 \log \frac{1}{\delta})\log T}}\right)$, there is*

$$F(x^{T+1}) - F(x^*) \le O\left(\frac{LD_\psi(x^*, x^1)}{T} + \frac{(M + \sigma\sqrt{\log \frac{1}{\delta}})\sqrt{D_\psi(x^*, x^1)\log T}}{\sqrt{T}}\right).$$

*Proof.* From Lemma 4.3, if $\eta_t \le \frac{1}{2L \vee \mu_f}, \forall t \in [T]$, with probability at least $1 - \delta$, there is

$$
\begin{aligned}
F(x^{T+1}) - F(x^*) \le &2\left(1 + \max_{2 \le t \le T} \frac{1}{1 - \mu_f \eta_t}\right) \\
&\times \left[\frac{D_\psi(x^*, x^1)}{\sum_{t=1}^T \gamma_t} + \left(M^2 + \sigma^2\left(1 + 2\log \frac{2}{\delta}\right)\right)\sum_{t=1}^T \frac{\gamma_t \eta_t}{\sum_{s=t}^T \gamma_s}\right],
\end{aligned}
\tag{25}
$$

where $\gamma_t := \eta_t \prod_{s=2}^t \frac{1 + \mu_h \eta_{s-1}}{1 - \mu_f \eta_s}, \forall t \in [T]$. Note that $\mu_f = \mu_h = 0$ now, so both $\eta_t = \frac{1}{2L} \wedge \frac{\eta}{\sqrt{t}}, \forall t \in [T]$ and $\eta_t = \frac{1}{2L} \wedge \frac{\eta}{\sqrt{T}}, \forall t \in [T]$ satisfy $\eta_t \le \frac{1}{2L \vee \mu_f} = \frac{1}{2L}, \forall t \in [T]$. Besides, $\gamma_t$ will degenerate to $\eta_t$. Then we can simplify (25) into

$$F(x^{T+1}) - F(x^*) \le \frac{4D_\psi(x^*, x^1)}{\sum_{t=1}^T \eta_t} + 4\left(M^2 + \sigma^2\left(1 + 2\log \frac{2}{\delta}\right)\right)\sum_{t=1}^T \frac{\eta_t^2}{\sum_{s=t}^T \eta_s}. \tag{26}$$

If $\eta_t = \frac{1}{2L} \wedge \frac{\eta}{\sqrt{t}}, \forall t \in [T]$, similar to (23), we will have

$$F(x^{T+1}) - F(x^*) \le O\left(\frac{LD_\psi(x^*, x^1)}{T} + \frac{1}{\sqrt{T}}\left[\frac{D_\psi(x^*, x^1)}{\eta} + \eta\left(M^2 + \sigma^2 \log \frac{1}{\delta}\right)\log T\right]\right).$$

By plugging in $\eta = \Theta\left(\sqrt{\frac{D_\psi(x^*, x^1)}{M^2 + \sigma^2 \log \frac{1}{\delta}}}\right)$, we get the desired bound.

If $\eta_t = \frac{1}{2L} \wedge \frac{\eta}{\sqrt{T}}, \forall t \in [T]$, similar to (24), we will get

$$F(x^{T+1}) - F(x^*) \le O\left(\frac{LD_\psi(x^*, x^1)}{T} + \frac{1}{\sqrt{T}}\left[\frac{D_\psi(x^*, x^1)}{\eta} + \eta\left(M^2 + \sigma^2 \log \frac{1}{\delta}\right)\log T\right]\right).$$

By plugging in $\eta = \Theta\left(\sqrt{\frac{D_\psi(x^*, x^1)}{(M^2 + \sigma^2 \log \frac{1}{\delta})\log T}}\right)$, we get the desired bound. $\qquad\square$

# D    STRONGLY CONVEX FUNCTIONS

In this section, we present the full version of theorems for strongly convex functions with their proofs.

## D.1    THE CASE OF $\mu_f > 0$

**Theorem D.1.** *Under Assumptions 1-4 and 5A with $\mu_f > 0$ and $\mu_h = 0$, let $\kappa_f := \frac{L}{\mu_f} \geq 0$:*

*If $T$ is unknown, by taking either $\eta_t = \frac{1}{\mu_f(t+2\kappa_f)}, \forall t \in [T]$ or $\eta_t = \frac{2}{\mu_f(t+1+4\kappa_f)}, \forall t \in [T]$, there is*

$$\mathbb{E}\left[F(x^{T+1}) - F(x^*)\right] \leq \begin{cases} O\left(\frac{LD_\psi(x^*,x^1)}{T} + \frac{(M^2+\sigma^2)\log T}{\mu_f(T+\kappa_f)}\right) & \eta_t = \frac{1}{\mu_f(t+2\kappa_f)}, \forall t \in [T] \\ O\left(\frac{L(1+\kappa_f)D_\psi(x^*,x^1)}{T(T+\kappa_f)} + \frac{(M^2+\sigma^2)\log T}{\mu_f(T+\kappa_f)}\right) & \eta_t = \frac{2}{\mu_f(t+1+4\kappa_f)}, \forall t \in [T] \end{cases}.$$

*If $T$ is known, by taking $\eta_t = \begin{cases} \frac{1}{\mu_f(1+2\kappa_f)} & t = 1 \\ \frac{1}{\mu_f(\eta+2\kappa_f)} & 2 \leq t \leq \tau \\ \frac{2}{\mu_f(t-\tau+2+4\kappa_f)} & t \geq \tau+1 \end{cases}, \forall t \in [T]$ where $\eta \geq 0$ can be any*

*number satisfying $\eta + \kappa_f > 1$ and $\tau := \lceil \frac{T}{2} \rceil$, there is*

$$\mathbb{E}\left[F(x^{T+1}) - F(x^*)\right] \leq O\left(\frac{LD_\psi(x^*,x^1)}{\exp\left(\frac{T}{2\eta+4\kappa_f}\right)} + \frac{(M^2+\sigma^2)\log T}{\mu_f(T+\kappa_f)}\right).$$

*Proof.* When $\mu_f > 0$ and $\mu_h = 0$, suppose the condition of $\eta_t \leq \frac{1}{2L \vee \mu_f}, \forall t \in [T]$ in Lemma 4.2 holds, we have

$$\mathbb{E}\left[F(x^{T+1}) - F(x^*)\right] \leq \frac{(1-\mu_f\eta_1)D_\psi(x^*,x^1)}{\sum_{t=1}^{T}\gamma_t} + 2(M^2+\sigma^2)\sum_{t=1}^{T}\frac{\gamma_t\eta_t}{\sum_{s=t}^{T}\gamma_s}. \quad (27)$$

Now observing that for any $t \in [T]$

$$\gamma_t := \eta_t \prod_{s=2}^{t}\frac{1+\mu_h\eta_{s-1}}{1-\mu_f\eta_s} = \eta_t\prod_{s=2}^{t}\frac{1}{1-\mu_f\eta_s} = \eta_t\Gamma_t = \begin{cases} \frac{\Gamma_t-\Gamma_{t-1}}{\mu_f} & t \geq 2 \\ \eta_1 & t = 1 \end{cases},$$

where $\Gamma_t := \prod_{s=2}^{t}\frac{1}{1-\mu_f\eta_s}, \forall t \in [T]$. Hence, (27) can be rewritten as

$$\mathbb{E}\left[F(x^{T+1}) - F(x^*)\right]$$
$$\leq \frac{(1-\mu_f\eta_1)D_\psi(x^*,x^1)}{\eta_1+\frac{\Gamma_T-1}{\mu_f}} + 2(M^2+\sigma^2)\left[\frac{\eta_1^2}{\eta_1+\frac{\Gamma_T-1}{\mu_f}} + \sum_{t=2}^{T}\frac{\mu_f\eta_t^2}{(\Gamma_T/\Gamma_{t-1}-1)(1-\mu_f\eta_t)}\right]. \quad (28)$$

Now let us check the condition of $\eta_t \leq \frac{1}{2L \vee \mu_f}, \forall t \in [T]$ for our three choices respectively:

$$\eta_t = \frac{1}{\mu_f(t+2\kappa_f)} \leq \frac{1}{\mu_f+2L} \leq \frac{1}{2L\vee\mu_f}, \forall t \in [T];$$

$$\eta_t = \frac{2}{\mu_f(t+1+4\kappa_f)} \leq \frac{1}{\mu_f+2L} \leq \frac{1}{2L\vee\mu_f}, \forall t \in [T];$$

$$\eta_t = \begin{cases} \frac{1}{\mu_f(1+2\kappa_f)} = \frac{1}{\mu_f+2L} \leq \frac{1}{2L\vee\mu_f} & t = 1 \\ \frac{1}{\mu_f(\eta+2\kappa_f)} \leq \frac{1}{\mu_f(2\kappa_f\vee(\eta+\kappa_f))} \leq \frac{1}{2L\vee\mu_f} & 2 \leq t \leq \tau \\ \frac{2}{\mu_f(t-\tau+2+4\kappa_f)} \leq \frac{1}{\mu_f+2L} \leq \frac{1}{2L\vee\mu_f} & t \geq \tau+1 \end{cases}.$$

Therefore, (28) holds for all cases.

First, we consider $\eta_t = \frac{1}{\mu_f(t+2\kappa_f)}, \forall t \in [T]$. We can calculate $\eta_1 = \frac{1}{\mu_f(1+2\kappa_f)}$ and

$$\Gamma_t = \prod_{s=2}^{t} \frac{1}{1-\mu_f\eta_s} = \prod_{s=2}^{t} \frac{s+2\kappa_f}{s-1+2\kappa_f} = \frac{t+2\kappa_f}{1+2\kappa_f}, \forall t \in [T].$$

Hence, using (28), we have

$$\mathbb{E}\left[F(x^{T+1}) - F(x^*)\right]$$
$$\leq \frac{(1-\frac{1}{1+2\kappa_f})D_\psi(x^*,x^1)}{\frac{1}{\mu_f(1+2\kappa_f)} + \frac{\Gamma_T-1}{\mu_f}} + 2(M^2+\sigma^2)\left[\frac{\frac{1}{\mu_f^2(1+2\kappa_f)^2}}{\frac{1}{\mu_f(1+2\kappa_f)} + \frac{\Gamma_T-1}{\mu_f}} + \sum_{t=2}^{T} \frac{\mu_f \cdot \frac{1}{\mu_f^2(t+2\kappa_f)^2}}{(\Gamma_T/\Gamma_{t-1}-1)(1-\mu_f \cdot \frac{1}{\mu_f(t+2\kappa_f)})}\right]$$
$$= \frac{2LD_\psi(x^*,x^1)}{(1+2\kappa_f)\Gamma_T - 2\kappa_f} + \frac{2(M^2+\sigma^2)}{\mu_f}\left[\frac{1}{(1+2\kappa_f)((1+2\kappa_f)\Gamma_T - 2\kappa_f)}\right.$$
$$\left.+ \sum_{t=2}^{T} \frac{1}{(\Gamma_T/\Gamma_{t-1}-1)(t-1+2\kappa_f)(t+2\kappa_f)}\right]$$
$$= \frac{2LD_\psi(x^*,x^1)}{T} + \frac{2(M^2+\sigma^2)}{\mu_f}\left[\frac{1}{(1+2\kappa_f)T} + \sum_{t=2}^{T} \frac{1}{(T-t+1)(t+2\kappa_f)}\right]$$
$$= \frac{2LD_\psi(x^*,x^1)}{T} + \frac{2(M^2+\sigma^2)}{\mu_f}\sum_{t=1}^{T} \frac{1}{(T-t+1)(t+2\kappa_f)}$$
$$= \frac{2LD_\psi(x^*,x^1)}{T} + \frac{2(M^2+\sigma^2)}{\mu_f}\sum_{t=1}^{T} \frac{1}{T+1+2\kappa_f}\left(\frac{1}{T-t+1} + \frac{1}{t+2\kappa_f}\right)$$
$$\leq \frac{2LD_\psi(x^*,x^1)}{T} + \frac{2(M^2+\sigma^2)}{\mu_f} \cdot \frac{1+\log T + \frac{1}{1+2\kappa_f} + \log\frac{T+2\kappa_f}{1+2\kappa_f}}{T+1+2\kappa_f}$$
$$\leq \frac{2LD_\psi(x^*,x^1)}{T} + \frac{4(M^2+\sigma^2)(1+\log T)}{\mu_f(T+2\kappa_f)}$$
$$= O\left(\frac{LD_\psi(x^*,x^1)}{T} + \frac{(M^2+\sigma^2)\log T}{\mu_f(T+\kappa_f)}\right). \tag{29}$$

Next, for the case of $\eta_t = \frac{2}{\mu_f(t+1+4\kappa_f)}, \forall t \in [T]$, there are $\eta_1 = \frac{1}{\mu_f(1+2\kappa_f)}$ and

$$\Gamma_t = \prod_{s=2}^{t} \frac{1}{1-\mu_f\eta_s} = \prod_{s=2}^{t} \frac{s+1+4\kappa_f}{s-1+4\kappa_f} = \frac{(t+4\kappa_f)(t+1+4\kappa_f)}{(1+4\kappa_f)(2+4\kappa_f)}, \forall t \in [T].$$

Thus, we have

$$\mathbb{E}\left[F(x^{T+1}) - F(x^*)\right]$$

$$\leq \frac{(1 - \frac{1}{1+2\kappa_f})D_\psi(x^*, x^1)}{\frac{1}{\mu_f(1+2\kappa_f)} + \frac{\Gamma_T - 1}{\mu_f}} + 2(M^2 + \sigma^2)\left[\frac{\frac{1}{\mu_f^2(1+2\kappa_f)^2}}{\frac{1}{\mu_f(1+2\kappa_f)} + \frac{\Gamma_T-1}{\mu_f}} + \sum_{t=2}^{T}\frac{\mu_f \cdot \frac{4}{\mu_f^2(t+1+4\kappa_f)^2}}{(\Gamma_T/\Gamma_{t-1} - 1)(1 - \mu_f \cdot \frac{2}{\mu_f(t+1+4\kappa_f)})}\right]$$

$$= \frac{2LD_\psi(x^*, x^1)}{(1+2\kappa_f)\Gamma_T - 2\kappa_f} + \frac{2(M^2+\sigma^2)}{\mu_f}\left[\frac{1}{(1+2\kappa_f)((1+2\kappa_f)\Gamma_T - 2\kappa_f)}\right.$$

$$\left. + \sum_{t=2}^{T}\frac{4}{(\Gamma_T/\Gamma_{t-1} - 1)(t-1+4\kappa_f)(t+1+4\kappa_f)}\right]$$

$$= \frac{4(1+4\kappa_f)LD_\psi(x^*, x^1)}{T(T+1+8\kappa_f)}$$

$$+ \frac{2(M^2+\sigma^2)}{\mu_f}\left[\frac{2(1+4\kappa_f)}{(1+2\kappa_f)T(T+1+8\kappa_f)} + \sum_{t=2}^{T}\frac{4(t+4\kappa_f)}{(T+1-t)(T+t+8\kappa_f)(t+1+4\kappa_f)}\right]$$

$$= \frac{4(1+4\kappa_f)LD_\psi(x^*, x^1)}{T(T+1+8\kappa_f)} + \frac{2(M^2+\sigma^2)}{\mu_f}\sum_{t=1}^{T}\frac{4(t+4\kappa_f)}{(T+1-t)(T+t+8\kappa_f)(t+1+4\kappa_f)}$$

$$\leq \frac{4(1+4\kappa_f)LD_\psi(x^*, x^1)}{T(T+1+8\kappa_f)} + \frac{2(M^2+\sigma^2)}{\mu_f}\sum_{t=1}^{T}\frac{4}{2T+1+8\kappa_f}\left(\frac{1}{T+1-t} + \frac{1}{T+t+8\kappa_f}\right)$$

$$\leq \frac{4(1+4\kappa_f)LD_\psi(x^*, x^1)}{T(T+1+8\kappa_f)} + \frac{2(M^2+\sigma^2)}{\mu_f} \cdot \frac{4(1 + \log T + \log\frac{2T+8\kappa_f}{T+8\kappa_f})}{2T+1+8\kappa_f}$$

$$\leq \frac{4(1+4\kappa_f)LD_\psi(x^*, x^1)}{T(T+1+8\kappa_f)} + \frac{8(M^2+\sigma^2)(1 + \log 2T)}{\mu_f(2T+1+8\kappa_f)}$$

$$= O\left(\frac{L(1+\kappa_f)D_\psi(x^*, x^1)}{T(T+\kappa_f)} + \frac{(M^2+\sigma^2)\log T}{\mu_f(T+\kappa_f)}\right). \tag{30}$$

Finally, if $T$ is known, recall that we choose for any $t \in [T]$

$$\eta_t = \begin{cases} \frac{1}{\mu_f(1+2\kappa_f)} & t = 1 \\ \frac{1}{\mu_f(\eta+2\kappa_f)} & 2 \leq t \leq \tau \\ \frac{2}{\mu_f(t-\tau+2+4\kappa_f)} & t \geq \tau+1 \end{cases}.$$

Note that we have $\eta_1 = \frac{1}{\mu_f(1+2\kappa_f)}$ and for any $t \in [T]$

$$\Gamma_t = \prod_{s=2}^{t}\frac{1}{1-\mu_f\eta_s} = \begin{cases} \left(\frac{\eta+2\kappa_f}{\eta+2\kappa_f-1}\right)^{t-1} & t \leq \tau \\ \left(\frac{\eta+2\kappa_f}{\eta+2\kappa_f-1}\right)^{\tau-1}\frac{(t-\tau+1+4\kappa_f)(t-\tau+2+4\kappa_f)}{(1+4\kappa_f)(2+4\kappa_f)} & t \geq \tau+1 \end{cases}.$$

So we know

$$\mathbb{E}\left[F(x^{T+1}) - F(x^*)\right]$$

$$\leq \frac{(1 - \mu_f\eta_1)D_\psi(x^*, x^1)}{\eta_1 + \frac{\Gamma_T-1}{\mu_f}} + 2(M^2+\sigma^2)\left[\frac{\eta_1^2}{\eta_1 + \frac{\Gamma_T-1}{\mu_f}} + \sum_{t=2}^{T}\frac{\mu_f\eta_t^2}{(\Gamma_T/\Gamma_{t-1} - 1)(1 - \mu_f\eta_t)}\right]$$

$$= \frac{2LD_\psi(x^*, x^1)}{(1+2\kappa_f)\Gamma_T - 2\kappa_f} + \frac{2(M^2+\sigma^2)}{\mu_f} \cdot \frac{1}{(1+2\kappa_f)((1+2\kappa_f)\Gamma_T - 2\kappa_f)}$$

$$+ 2\mu_f(M^2+\sigma^2)\left[\underbrace{\sum_{t=2}^{\tau}\frac{\eta_t^2}{(\Gamma_T/\Gamma_{t-1} - 1)(1 - \mu_f\eta_t)}}_{\text{I}} + \underbrace{\sum_{t=\tau+1}^{T}\frac{\eta_t^2}{(\Gamma_T/\Gamma_{t-1} - 1)(1 - \mu_f\eta_t)}}_{\text{II}}\right]. \tag{31}$$

Note that we can bound

$$
\frac{1}{(1+2\kappa_f)\Gamma_T - 2\kappa_f}
$$

$$
= \frac{1}{\left(\frac{\eta+2\kappa_f}{\eta+2\kappa_f-1}\right)^{\tau-1} \frac{(T-\tau+1+4\kappa_f)(T-\tau+2+4\kappa_f)}{2(1+4\kappa_f)} - 2\kappa_f}
$$

$$
\overset{(a)}{\leq} \frac{1}{\left(\frac{\eta+2\kappa_f}{\eta+2\kappa_f-1}\right)^{\tau-1}(1+2\kappa_f) - 2\kappa_f} = \frac{1}{2\kappa_f\left[\left(\frac{\eta+2\kappa_f}{\eta+2\kappa_f-1}\right)^{\tau-1} - 1\right] + \left(\frac{\eta+2\kappa_f}{\eta+2\kappa_f-1}\right)^{\tau-1}}
$$

$$
\overset{(b)}{\leq} \left(1 - \frac{1}{\eta+2\kappa_f}\right)^{\tau-1} \leq \exp\left(-\frac{\tau-1}{\eta+2\kappa_f}\right) = \exp\left(\frac{-\tau}{\eta+2\kappa_f} + \frac{1}{\eta+2\kappa_f}\right)
$$

$$
\overset{(c)}{\leq} \exp\left(-\frac{T}{2(\eta+2\kappa_f)} + 1\right), \tag{32}
$$

where $(a)$ holds due to $T - \tau \geq 0$, $(b)$ is by $\kappa_f \geq 0$ and $\left(\frac{\eta+2\kappa_f}{\eta+2\kappa_f-1}\right)^{\tau-1} \geq 1$, $(c)$ is from $\tau \geq \frac{T}{2}$, $\eta + \kappa_f > 1$ and $\kappa_f \geq 0$. We can also bound

$$
\frac{1}{(1+2\kappa_f)((1+2\kappa_f)\Gamma_T - 2\kappa_f)}
$$

$$
= \frac{1}{(1+2\kappa_f)\left[\left(\frac{\eta+2\kappa_f}{\eta+2\kappa_f-1}\right)^{\tau-1} \frac{(T-\tau+1+4\kappa_f)(T-\tau+2+4\kappa_f)}{2(1+4\kappa_f)} - 2\kappa_f\right]}
$$

$$
\leq \frac{1}{(1+2\kappa_f)\left[\frac{(T-\tau+1+4\kappa_f)(T-\tau+2+4\kappa_f)}{2(1+4\kappa_f)} - 2\kappa_f\right]}
$$

$$
= \frac{2(1+4\kappa_f)}{(1+2\kappa_f)(T-\tau+1)(T-\tau+2+8\kappa_f)}
$$

$$
\overset{(d)}{\leq} \frac{2(1+4\kappa_f)}{(1+2\kappa_f)(T-\frac{T+1}{2}+1)(T-\frac{T+1}{2}+2+8\kappa_f)}
$$

$$
= \frac{8(1+4\kappa_f)}{(1+2\kappa_f)(T+1)(T+3+16\kappa_f)} \overset{(e)}{\leq} \frac{8}{T+3+16\kappa_f}, \tag{33}
$$

where $(d)$ is by $\tau \leq \frac{T+1}{2}$ and $(e)$ is by $T \geq 1$. Besides, there is

$$
\mathrm{I} = \frac{1}{\mu_f^2(\eta+2\kappa_f)(\eta+2\kappa_f-1)} \sum_{t=2}^{\tau} \frac{1}{\left(\frac{\eta+2\kappa_f}{\eta+2\kappa_f-1}\right)^{\tau-t+1} \frac{(T-\tau+1+4\kappa_f)(T-\tau+2+4\kappa_f)}{(1+4\kappa_f)(2+4\kappa_f)} - 1}
$$

$$
\overset{(f)}{\leq} \frac{1}{\mu_f^2(\eta+2\kappa_f)(\eta+2\kappa_f-1)\frac{\eta+2\kappa_f}{\eta+2\kappa_f-1}} \sum_{t=2}^{\tau} \frac{1}{\frac{(T-\tau+1+4\kappa_f)(T-\tau+2+4\kappa_f)}{(1+4\kappa_f)(2+4\kappa_f)} - 1}
$$

$$
= \frac{1}{\mu_f^2(\eta+2\kappa_f)^2} \sum_{t=2}^{\tau} \frac{(1+4\kappa_f)(2+4\kappa_f)}{(T-\tau)(T-\tau+3+8\kappa_f)} = \frac{(1+4\kappa_f)(2+4\kappa_f)}{\mu_f^2(\eta+2\kappa_f)^2} \cdot \frac{\tau-1}{(T-\tau)(T-\tau+3+8\kappa_f)}
$$

$$
\overset{(g)}{\leq} \frac{(1+4\kappa_f)(2+4\kappa_f)}{\mu_f^2(\eta+2\kappa_f)^2} \cdot \frac{\frac{T+1}{2}-1}{(T-\frac{T+1}{2})(T-\frac{T+1}{2}+3+8\kappa_f)}
$$

$$
= \frac{(1+4\kappa_f)(2+4\kappa_f)}{\mu_f^2(\eta+2\kappa_f)^2} \cdot \frac{2}{T+5+16\kappa_f} \overset{(h)}{\leq} \frac{32}{\mu_f^2(T+5+16\kappa_f)}, \tag{34}
$$

where $(f)$ is by $\tau - t \geq 0$ (w.l.o.g., we can assume $\tau \geq 2$, otherwise, there is I$= 0 \leq \frac{8}{\mu_f^2(T+5+16\kappa_f)}$) and $\frac{\eta + 2\kappa_f}{\eta + 2\kappa_f - 1} \geq 1$, $(g)$ is due to $\tau \leq \frac{T+1}{2}$ and $(h)$ is by $\eta + \kappa_f > 1$. We also have

$$
\begin{aligned}
\text{II} =& \frac{4}{\mu_f^2} \sum_{t=\tau+1}^{T} \frac{1}{\left[ \frac{(T-\tau+1+4\kappa_f)(T-\tau+2+4\kappa_f)}{(t-\tau+4\kappa_f)(t-\tau+1+4\kappa_f)} - 1 \right] (t-\tau+2+4\kappa_f)(t-\tau+4\kappa_f)} \\
=& \frac{4}{\mu_f^2} \sum_{t=1}^{T-\tau} \frac{1}{\left[ \frac{(T-\tau+1+4\kappa_f)(T-\tau+2+4\kappa_f)}{(t+4\kappa_f)(t+1+4\kappa_f)} - 1 \right] (t+2+4\kappa_f)(t+4\kappa_f)} \\
=& \frac{4}{\mu_f^2} \sum_{t=1}^{T-\tau} \frac{t+1+4\kappa_f}{\left[ (T-\tau+1+4\kappa_f)(T-\tau+2+4\kappa_f) - (t+4\kappa_f)(t+1+4\kappa_f) \right] (t+2+4\kappa_f)} \\
\leq& \frac{4}{\mu_f^2} \sum_{t=1}^{T-\tau} \frac{1}{(T-\tau+1+4\kappa_f)(T-\tau+2+4\kappa_f) - (t+4\kappa_f)(t+1+4\kappa_f)} \\
=& \frac{4}{\mu_f^2} \sum_{t=1}^{T-\tau} \frac{1}{(T-\tau+1-t)(T-\tau+2+8\kappa_f+t)} \\
=& \frac{4}{\mu_f^2} \sum_{t=1}^{T-\tau} \frac{1}{2T-2\tau+3+8\kappa_f} \left( \frac{1}{T-\tau+1-t} + \frac{1}{T-\tau+2+8\kappa_f+t} \right) \\
\leq& \frac{4(1+\log(T-\tau) + \log \frac{2T-2\tau+2+8\kappa_f}{T-\tau+2+8\kappa_f})}{\mu_f^2(2T-2\tau+3+8\kappa_f)} \leq \frac{4(1+\log T)}{\mu_f^2(T+2+8\kappa_f)},
\end{aligned}
\tag{35}
$$

where we use $\frac{T}{2} \leq \tau \leq \frac{T+1}{2}$ in the last inequality.

Plugging (32), (33), (34) and (35) into (31), we have

$$
\begin{aligned}
& \mathbb{E}\left[ F(x^{T+1}) - F(x^*) \right] \\
\leq& 2LD_\psi(x^*, x^1) \exp\left( -\frac{T}{2(\eta+2\kappa_f)} + 1 \right) + \frac{2(M^2+\sigma^2)}{\mu_f} \cdot \frac{8}{T+3+16\kappa_f} \\
& + 2\mu_f(M^2+\sigma^2) \left[ \frac{32}{\mu_f^2(T+5+16\kappa_f)} + \frac{4(1+\log T)}{\mu_f^2(T+2+8\kappa_f)} \right] \\
=& O\left( \frac{LD_\psi(x^*, x^1)}{\exp\left( \frac{T}{2\eta+4\kappa_f} \right)} + \frac{(M^2+\sigma^2)\log T}{\mu_f(T+\kappa_f)} \right).
\end{aligned}
\tag{36}
$$

$\square$

**Theorem D.2.** *Under Assumptions 1-4 and 5B with $\mu_f > 0$ and $\mu_h = 0$, let $\kappa_f := \frac{L}{\mu_f} \geq 0$ and $\delta \in (0,1)$:*

*If $T$ is unknown, by taking either $\eta_t = \frac{1}{\mu_f(t+2\kappa_f)}, \forall t \in [T]$ or $\eta_t = \frac{2}{\mu_f(t+1+4\kappa_f)}, \forall t \in [T]$, then with probability at least $1 - \delta$, there is*

$$
F(x^{T+1}) - F(x^*) \leq \begin{cases} O\left( \frac{\mu_f(1+\kappa_f)D_\psi(x^*,x^1)}{T} + \frac{(M^2+\sigma^2 \log \frac{1}{\delta})\log T}{\mu_f(T+\kappa_f)} \right) & \eta_t = \frac{1}{\mu_f(t+2\kappa_f)}, \forall t \in [T] \\ O\left( \frac{\mu_f(1+\kappa_f)^2 D_\psi(x^*,x^1)}{T(T+\kappa_f)} + \frac{(M^2+\sigma^2 \log \frac{1}{\delta})\log T}{\mu_f(T+\kappa_f)} \right) & \eta_t = \frac{2}{\mu_f(t+1+4\kappa_f)}, \forall t \in [T] \end{cases}.
$$

*If $T$ is known, by taking $\eta_t = \begin{cases} \frac{1}{\mu_f(1+2\kappa_f)} & t = 1 \\ \frac{1}{\mu_f(\eta+2\kappa_f)} & 2 \leq t \leq \tau \\ \frac{2}{\mu_f(t-\tau+2+4\kappa_f)} & t \geq \tau+1 \end{cases}, \forall t \in [T]$ where $\eta \geq 0$ can be any number satisfying $\eta + \kappa_f > 1$ and $\tau := \lceil \frac{T}{2} \rceil$, then with probability at least $1 - \delta$, there is*

$$
F(x^{T+1}) - F(x^*) \leq O\left( \left( 1 \vee \frac{1}{\eta+2\kappa_f-1} \right) \left[ \frac{\mu_f(1+\kappa_f)D_\psi(x^*,x^1)}{\exp\left( \frac{T}{2\eta+4\kappa_f} \right)} + \frac{(M^2+\sigma^2 \log \frac{1}{\delta})\log T}{\mu_f(T+\kappa_f)} \right] \right).
$$

*Proof.* When $\mu_f > 0$ and $\mu_h = 0$, suppose the condition of $\eta_t \leq \frac{1}{2L \vee \mu_f}, \forall t \in [T]$ in Lemma 4.3 holds, we will have with probability at least $1 - \delta$

$$
\begin{aligned}
F(x^{T+1}) - F(x^*) \leq & 2\left(1 + \max_{2 \leq t \leq T} \frac{1}{1 - \mu_f \eta_t}\right) \\
& \times \left[\frac{D_\psi(x^*, x^1)}{\sum_{t=1}^T \gamma_t} + \left(M^2 + \sigma^2\left(1 + 2\log\frac{2}{\delta}\right)\right) \sum_{t=1}^T \frac{\gamma_t \eta_t}{\sum_{s=t}^T \gamma_s}\right].
\end{aligned}
\tag{37}
$$

Now observing that for any $t \in [T]$

$$
\gamma_t := \eta_t \prod_{s=2}^t \frac{1 + \mu_h \eta_{s-1}}{1 - \mu_f \eta_s} = \eta_t \prod_{s=2}^t \frac{1}{1 - \mu_f \eta_s} = \eta_t \Gamma_t = \begin{cases} \frac{\Gamma_t - \Gamma_{t-1}}{\mu_f} & t \geq 2 \\ \eta_1 & t = 1 \end{cases},
$$

where $\Gamma_t := \prod_{s=2}^t \frac{1}{1 - \mu_f \eta_s}, \forall t \in [T]$. Hence, (37) can be rewritten as

$$
\begin{aligned}
F(x^{T+1}) - F(x^*) \leq & 2\left(1 + \max_{2 \leq t \leq T} \frac{1}{1 - \mu_f \eta_t}\right) \\
& \times \left(\frac{D_\psi(x^*, x^1)}{\eta_1 + \frac{\Gamma_T - 1}{\mu_f}} + \left(M^2 + \sigma^2\left(1 + 2\log\frac{2}{\delta}\right)\right) \sum_{t=1}^T \frac{\gamma_t \eta_t}{\sum_{s=t}^T \gamma_s}\right).
\end{aligned}
\tag{38}
$$

Now let us check the condition of $\eta_t \leq \frac{1}{2L \vee \mu_f}, \forall t \in [T]$ for our three choices respectively:

$$
\begin{aligned}
\eta_t &= \frac{1}{\mu_f(t + 2\kappa_f)} \leq \frac{1}{\mu_f + 2L} \leq \frac{1}{2L \vee \mu_f}, \forall t \in [T]; \\
\eta_t &= \frac{2}{\mu_f(t + 1 + 4\kappa_f)} \leq \frac{1}{\mu_f + 2L} \leq \frac{1}{2L \vee \mu_f}, \forall t \in [T]; \\
\eta_t &= \begin{cases} \frac{1}{\mu_f(1 + 2\kappa_f)} = \frac{1}{\mu_f + 2L} \leq \frac{1}{2L \vee \mu_f} & t = 1 \\ \frac{1}{\mu_f(\eta + 2\kappa_f)} \leq \frac{1}{\mu_f(2\kappa_f \vee (\eta + \kappa_f))} \leq \frac{1}{2L \vee \mu_f} & 2 \leq t \leq \tau \\ \frac{2}{\mu_f(t - \tau + 2 + 4\kappa_f)} \leq \frac{1}{\mu_f + 2L} \leq \frac{1}{2L \vee \mu_f} & t \geq \tau + 1 \end{cases}.
\end{aligned}
$$

Therefore, (38) holds for all cases.

First, we consider $\eta_t = \frac{1}{\mu_f(t + 2\kappa_f)}, \forall t \in [T]$. We can find $1 + \max_{2 \leq t \leq T} \frac{1}{1 - \mu_f \eta_t} = 1 + \frac{1}{1 - \mu_f \eta_2} \leq 3$, $\eta_1 = \frac{1}{\mu_f(1 + 2\kappa_f)}$ and

$$
\Gamma_t = \prod_{s=2}^t \frac{1}{1 - \mu_f \eta_s} = \prod_{s=2}^t \frac{s + 2\kappa_f}{s - 1 + 2\kappa_f} = \frac{t + 2\kappa_f}{1 + 2\kappa_f}, \forall t \in [T].
$$

Hence, using (38), we have

$$
F(x^{T+1}) - F(x^*) \leq 6\left(\frac{D_\psi(x^*, x^1)}{\eta_1 + \frac{\Gamma_T - 1}{\mu_f}} + \left(M^2 + \sigma^2\left(1 + 2\log\frac{2}{\delta}\right)\right) \sum_{t=1}^T \frac{\gamma_t \eta_t}{\sum_{s=t}^T \gamma_s}\right).
$$

Following similar steps in the proof of (29), we can get

$$
F(x^{T+1}) - F(x^*) \leq O\left(\frac{\mu_f(1 + \kappa_f)D_\psi(x^*, x^1)}{T} + \frac{(M^2 + \sigma^2 \log\frac{1}{\delta})\log T}{\mu_f(T + \kappa_f)}\right).
$$

Next, for the case of $\eta_t = \frac{2}{\mu_f(t + 1 + 4\kappa_f)}, \forall t \in [T]$, there are $1 + \max_{2 \leq t \leq T} \frac{1}{1 - \mu_f \eta_t} = 1 + \frac{1}{1 - \mu_f \eta_2} \leq 4$, $\eta_1 = \frac{1}{\mu_f(1 + 2\kappa_f)}$ and

$$
\Gamma_t = \prod_{s=2}^t \frac{1}{1 - \mu_f \eta_s} = \prod_{s=2}^t \frac{s + 1 + 4\kappa_f}{s - 1 + 4\kappa_f} = \frac{(t + 4\kappa_f)(t + 1 + 4\kappa_f)}{2(1 + 4\kappa_f)(1 + 2\kappa_f)}, \forall t \in [T].
$$

Thus, we have

$$F(x^{T+1}) - F(x^*) \le 8 \left( \frac{D_\psi(x^*, x^1)}{\eta_1 + \frac{\Gamma_T - 1}{\mu_f}} + \left( M^2 + \sigma^2 \left( 1 + 2\log\frac{2}{\delta} \right) \right) \sum_{t=1}^{T} \frac{\gamma_t \eta_t}{\sum_{s=t}^{T} \gamma_s} \right).$$

Following similar steps in the proof of (30), we can get

$$F(x^{T+1}) - F(x^*) \le O \left( \frac{\mu_f (1 + \kappa_f)^2 D_\psi(x^*, x^1)}{T(T + \kappa_f)} + \frac{(M^2 + \sigma^2 \log\frac{1}{\delta}) \log T}{\mu_f(T + \kappa_f)} \right).$$

Finally, if $T$ is known, we recall the current choice is for any $t \in [T]$

$$\eta_t = \begin{cases} \frac{1}{\mu_f(1 + 2\kappa_f)} & t = 1 \\ \frac{1}{\mu_f(\eta + 2\kappa_f)} & 2 \le t \le \tau \\ \frac{2}{\mu_f(t - \tau + 2 + 4\kappa_f)} & t \ge \tau + 1 \end{cases},$$

where $\eta > 1$ and $\tau = \lceil \frac{T}{2} \rceil$. Note that we have

$$1 + \max_{2 \le t \le T} \frac{1}{1 - \mu_f \eta_t} = 1 + \frac{1}{1 - \mu_f(\eta_2 \vee \eta_{\tau+1})} = 2 + \frac{1}{\eta \wedge 1.5 + 2\kappa_f - 1}$$

$$= 2 + \frac{1}{\eta + 2\kappa_f - 1} \vee \frac{1}{0.5 + 2\kappa_f}$$

$$\le 4 + \frac{1}{\eta + 2\kappa_f - 1},$$

$\eta_1 = \frac{1}{\mu_f(1 + 2\kappa_f)}$ and for any $t \in [T]$

$$\Gamma_t = \prod_{s=2}^{t} \frac{1}{1 - \mu_f \eta_s} = \begin{cases} \left( \frac{\eta + 2\kappa_f}{\eta + 2\kappa_f - 1} \right)^{t-1} & t \le \tau \\ \left( \frac{\eta + 2\kappa_f}{\eta + 2\kappa_f - 1} \right)^{\tau - 1} \frac{(t - \tau + 1 + 4\kappa_f)(t - \tau + 2 + 4\kappa_f)}{(1 + 4\kappa_f)(2 + 4\kappa_f)} & t \ge \tau + 1 \end{cases}.$$

Thus, we have

$$F(x^{T+1}) - F(x^*)$$

$$\le \left( 8 + \frac{2}{\eta + 2\kappa_f - 1} \right) \left( \frac{D_\psi(x^*, x^1)}{\eta_1 + \frac{\Gamma_T - 1}{\mu_f}} + \left( M^2 + \sigma^2 \left( 1 + 2\log\frac{2}{\delta} \right) \right) \sum_{t=1}^{T} \frac{\gamma_t \eta_t}{\sum_{s=t}^{T} \gamma_s} \right).$$

Following similar steps in the proof of (36), we can get

$$F(x^{T+1}) - F(x^*)$$

$$\le O \left( \left( 1 \vee \frac{1}{\eta + 2\kappa_f - 1} \right) \left[ \frac{\mu_f (1 + \kappa_f) D_\psi(x^*, x^1)}{\exp\left( \frac{T}{2\eta + 4\kappa_f} \right)} + \frac{(M^2 + \sigma^2 \log\frac{1}{\delta}) \log T}{\mu_f(T + \kappa_f)} \right] \right).$$

$\square$

## D.2 THE CASE OF $\mu_h > 0$

**Theorem D.3.** *Under Assumptions 1-4 and 5A with $\mu_f = 0$ and $\mu_h > 0$, let $\kappa_h := \frac{L}{\mu_h} \ge 0$:*

*If $T$ is unknown, by taking $\eta_t = \frac{2}{\mu_h(t + 4\kappa_h)}, \forall t \in [T]$, there is*

$$\mathbb{E}\left[ F(x^{T+1}) - F(x^*) \right] \le O \left( \frac{\mu_h (1 + \kappa_h)^2 D_\psi(x^*, x^1)}{T(T + \kappa_h)} + \frac{(M^2 + \sigma^2) \log T}{\mu_h(T + \kappa_h)} \right).$$

*If $T$ is known, by taking $\eta_t = \begin{cases} \frac{1}{\mu_h(\eta + 2\kappa_h)} & t \le \tau \\ \frac{2}{\mu_h(t - \tau + 4\kappa_h)} & t \ge \tau + 1 \end{cases}, \forall t \in [T]$ where $\eta \ge 0$ can be any number satisfying $\eta + \kappa_h > 0$ and $\tau := \lceil \frac{T}{2} \rceil$, there is*

$$\mathbb{E}\left[ F(x^{T+1}) - F(x^*) \right] \le O \left( \frac{\mu_h D_\psi(x^*, x^1)}{\exp\left( \frac{T}{2(1 + \eta + 2\kappa_h)} \right) - 1} + \left( 1 \vee \frac{1}{\eta + 2\kappa_h} \right) \frac{(M^2 + \sigma^2) \log T}{\mu_h(T + \kappa_h)} \right).$$

*Proof.* When $\mu_f = 0$ and $\mu_h > 0$, suppose the condition of $\eta_t \leq \frac{1}{2L \vee \mu_f} = \frac{1}{2L}, \forall t \in [T]$ in Lemma 4.2 holds, we will have

$$
\mathbb{E}\left[F(x^{T+1}) - F(x^*)\right] \leq \frac{(1 - \mu_f \eta_1)D_\psi(x^*, x^1)}{\sum_{t=1}^T \gamma_t} + 2(M^2 + \sigma^2) \sum_{t=1}^T \frac{\gamma_t \eta_t}{\sum_{s=t}^T \gamma_s}
$$

$$
= \frac{D_\psi(x^*, x^1)}{\sum_{t=1}^T \gamma_t} + 2(M^2 + \sigma^2) \sum_{t=1}^T \frac{\gamma_t \eta_t}{\sum_{s=t}^T \gamma_s}. \tag{39}
$$

Observing that

$$
\gamma_t := \eta_t \prod_{s=2}^t \frac{1 + \mu_h \eta_{s-1}}{1 - \mu_f \eta_s} = \eta_t \prod_{s=2}^t (1 + \mu_h \eta_{s-1}) = \eta_t \Gamma_{t-1} = \frac{\Gamma_t - \Gamma_{t-1}}{\mu_h}, \forall t \in [T],
$$

where $\Gamma_t := \prod_{s=1}^t (1 + \mu_h \eta_s), \forall t \in \{0\} \cup [T]$. So (39) can be rewritten as

$$
\mathbb{E}\left[F(x^{T+1}) - F(x^*)\right] \leq \frac{\mu_h D_\psi(x^*, x^1)}{\Gamma_T - 1} + 2\mu_h(M^2 + \sigma^2) \sum_{t=1}^T \frac{\eta_t^2}{\Gamma_T/\Gamma_{t-1} - 1}. \tag{40}
$$

We can check that

$$
\eta_t = \frac{2}{\mu_h(t + 4\kappa_h)} \leq \frac{1}{2\kappa_h \mu_h} = \frac{1}{2L}, \forall t \in [T];
$$

$$
\eta_t = \begin{cases} \frac{1}{\mu_h(\eta + 2\kappa_h)} \leq \frac{1}{2\kappa_h \mu_h} = \frac{1}{2L} & t \leq \tau \\ \frac{2}{\mu_h(t - \tau + 4\kappa_h)} \leq \frac{1}{2\kappa_h \mu_h} = \frac{1}{2L} & t \geq \tau + 1 \end{cases}.
$$

Therefore, (40) is true for all cases.

If $\eta_t = \frac{2}{\mu_h(t + 4\kappa_h)}, \forall t \in [T]$, we have

$$
\Gamma_t = \prod_{s=1}^t (1 + \mu_h \eta_s) = \frac{(t + 1 + 4\kappa_h)(t + 2 + 4\kappa_h)}{(1 + 4\kappa_h)(2 + 4\kappa_h)}, \forall t \in \{0\} \cup [T].
$$

Hence, by (40),

$$
\mathbb{E}\left[F(x^{T+1}) - F(x^*)\right]
$$

$$
\leq \frac{\mu_h D_\psi(x^*, x^1)}{\frac{(T+1+4\kappa_h)(T+2+4\kappa_h)}{(1+4\kappa_h)(2+4\kappa_h)} - 1} + \frac{8(M^2 + \sigma^2)}{\mu_h} \sum_{t=1}^T \frac{\frac{1}{(t+4\kappa_h)^2}}{\frac{(T+1+4\kappa_h)(T+2+4\kappa_h)}{(t+4\kappa_h)(t+1+4\kappa_h)} - 1}
$$

$$
= \frac{(1 + 4\kappa_h)(2 + 4\kappa_h)\mu_h D_\psi(x^*, x^1)}{T(T + 3 + 8\kappa_h)} + \frac{8(M^2 + \sigma^2)}{\mu_h} \sum_{t=1}^T \frac{t + 1 + 4\kappa_h}{t + 4\kappa_h} \cdot \frac{1}{(T + 1 - t)(T + 2 + 8\kappa_h + t)}
$$

$$
\leq \frac{(1 + 4\kappa_h)(2 + 4\kappa_h)\mu_h D_\psi(x^*, x^1)}{T(T + 3 + 8\kappa_h)} + \frac{16(M^2 + \sigma^2)}{\mu_h} \sum_{t=1}^T \frac{1}{2T + 3 + 8\kappa_h} \left(\frac{1}{T + 1 - t} + \frac{1}{T + 2 + 8\kappa_h + t}\right)
$$

$$
\leq \frac{(1 + 4\kappa_h)(2 + 4\kappa_h)\mu_h D_\psi(x^*, x^1)}{T(T + 3 + 8\kappa_h)} + \frac{16(M^2 + \sigma^2)}{\mu_h} \cdot \frac{1 + \log T + \log \frac{2T + 2 + 8\kappa_h}{T + 2 + 8\kappa_h}}{2T + 3 + 8\kappa_h}
$$

$$
\leq \frac{(1 + 4\kappa_h)(2 + 4\kappa_h)\mu_h D_\psi(x^*, x^1)}{T(T + 3 + 8\kappa_h)} + \frac{16(M^2 + \sigma^2)(1 + \log 2T)}{\mu_h(2T + 3 + 8\kappa_h)}
$$

$$
= O\left(\frac{\mu_h(1 + \kappa_h)^2 D_\psi(x^*, x^1)}{T(T + \kappa_h)} + \frac{(M^2 + \sigma^2)\log T}{\mu_h(T + \kappa_h)}\right). \tag{41}
$$

If $\eta_t = \begin{cases} \frac{1}{\mu_h(\eta + 2\kappa_h)} & t \leq \tau \\ \frac{2}{\mu_h(t - \tau + 4\kappa_h)} & t \geq \tau + 1 \end{cases}, \forall t \in [T]$, we know for any $t \in \{0\} \cup [T]$

$$
\Gamma_t = \prod_{s=1}^t (1 + \mu_h \eta_s) = \begin{cases} \left(1 + \frac{1}{\eta + 2\kappa_h}\right)^t & t \leq \tau \\ \left(1 + \frac{1}{\eta + 2\kappa_h}\right)^\tau \frac{(t - \tau + 1 + 4\kappa_h)(t - \tau + 2 + 4\kappa_h)}{(1 + 4\kappa_h)(2 + 4\kappa_h)} & t \geq \tau + 1 \end{cases}.
$$

So we can obtain

$$
\mathbb{E}\left[F(x^{T+1}) - F(x^*)\right]
$$

$$
\leq \frac{\mu_h D_\psi(x^*, x^1)}{\Gamma_T - 1} + 2\mu_h(M^2 + \sigma^2) \sum_{t=1}^{T} \frac{\eta_t^2}{\Gamma_T/\Gamma_{t-1} - 1}
$$

$$
= \frac{\mu_h D_\psi(x^*, x^1)}{\left(1 + \frac{1}{\eta + 2\kappa_h}\right)^\tau \frac{(T - \tau + 1 + 4\kappa_h)(T - \tau + 2 + 4\kappa_h)}{(1 + 4\kappa_h)(2 + 4\kappa_h)} - 1} + 2\mu_h(M^2 + \sigma^2) \sum_{t=1}^{T} \frac{\eta_t^2}{\Gamma_T/\Gamma_{t-1} - 1}
$$

$$
\overset{(a)}{\leq} \frac{\mu_h D_\psi(x^*, x^1)}{\left(1 + \frac{1}{\eta + 2\kappa_h}\right)^{T/2} - 1} + 2\mu_h(M^2 + \sigma^2) \sum_{t=1}^{T} \frac{\eta_t^2}{\Gamma_T/\Gamma_{t-1} - 1}
$$

$$
\overset{(b)}{\leq} \frac{\mu_h D_\psi(x^*, x^1)}{\exp\left(\frac{T}{2(1 + \eta + 2\kappa_h)}\right) - 1} + 2\mu_h(M^2 + \sigma^2) \sum_{t=1}^{T} \frac{\eta_t^2}{\Gamma_T/\Gamma_{t-1} - 1}
$$

$$
= \frac{\mu_h D_\psi(x^*, x^1)}{\exp\left(\frac{T}{2(1 + \eta + 2\kappa_h)}\right) - 1} + 2\mu_h(M^2 + \sigma^2) \left( \underbrace{\sum_{t=1}^{\tau} \frac{\eta_t^2}{\Gamma_T/\Gamma_{t-1} - 1}}_{\text{I}} + \underbrace{\sum_{t=\tau+1}^{T} \frac{\eta_t^2}{\Gamma_T/\Gamma_{t-1} - 1}}_{\text{II}} \right), \quad (42)
$$

where $(a)$ is by $T - \tau \geq 0$ and $\tau \geq \frac{T}{2}$, $(b)$ is due to

$$
\left(1 + \frac{1}{\eta + 2\kappa_h}\right)^{T/2} = \exp\left(\frac{T}{2}\log\left(1 + \frac{1}{\eta + 2\kappa_h}\right)\right) \geq \exp\left(\frac{T}{2(1 + \eta + 2\kappa_h)}\right).
$$

Now we bound

$$
\text{I} = \frac{1}{\mu_h^2(\eta + 2\kappa_h)^2} \sum_{t=1}^{\tau} \frac{1}{\left(1 + \frac{1}{\eta + 2\kappa_h}\right)^{\tau - t + 1} \frac{(T - \tau + 1 + 4\kappa_h)(T - \tau + 2 + 4\kappa_h)}{(1 + 4\kappa_h)(2 + 4\kappa_h)} - 1}
$$

$$
\overset{(c)}{\leq} \frac{1}{\mu_h^2(\eta + 2\kappa_h)^2 \left(1 + \frac{1}{\eta + 2\kappa_h}\right)} \sum_{t=1}^{\tau} \frac{1}{\frac{(T - \tau + 1 + 4\kappa_h)(T - \tau + 2 + 4\kappa_h)}{(1 + 4\kappa_h)(2 + 4\kappa_h)} - 1}
$$

$$
= \frac{1}{\mu_h^2(\eta + 2\kappa_h)(\eta + 1 + 2\kappa_h)} \sum_{t=1}^{\tau} \frac{(1 + 4\kappa_h)(2 + 4\kappa_h)}{(T - \tau)(T - \tau + 8\kappa_h + 3)}
$$

$$
= \frac{(1 + 4\kappa_h)(2 + 4\kappa_h)}{\mu_h^2(\eta + 2\kappa_h)(\eta + 1 + 2\kappa_h)} \cdot \frac{\tau}{(T - \tau)(T - \tau + 8\kappa_h + 3)}
$$

$$
\overset{(d)}{\leq} \frac{2(1 + 4\kappa_h)}{\mu_h^2(\eta + 2\kappa_h)} \cdot \frac{2(T + 1)}{(T - 1)(T + 5 + 16\kappa_h)}
$$

$$
\overset{(e)}{\leq} \frac{2\left(2 + \frac{1}{\eta + 2\kappa_h}\right)}{\mu_h^2} \cdot \frac{6}{T + 5 + 16\kappa_h} = \frac{12\left(2 + \frac{1}{\eta + 2\kappa_h}\right)}{\mu_h^2(T + 5 + 16\kappa_h)}, \quad (43)
$$

where $(c)$ is by $\tau - t \geq 0$ and $1 + \frac{1}{\eta + 2\kappa_h} \geq 1$. We use $\eta \geq 0$, $\tau \leq \frac{T+1}{2}$ in $(d)$ and $T \geq 2$ in $(e)$. Next, there is

$$
\begin{aligned}
\text{II} &= \frac{4}{\mu_h^2} \sum_{t=\tau+1}^{T} \frac{1}{(t - \tau + 4\kappa_h)^2 \left( \frac{(T-\tau+1+4\kappa_h)(T-\tau+2+4\kappa_h)}{(t-\tau+4\kappa_h)(t-\tau+1+4\kappa_h)} - 1 \right)} \\
&= \frac{4}{\mu_h^2} \sum_{t=1}^{T-\tau} \frac{1}{(t + 4\kappa_h)^2 \left( \frac{(T-\tau+1+4\kappa_h)(T-\tau+2+4\kappa_h)}{(t+4\kappa_h)(t+1+4\kappa_h)} - 1 \right)} \\
&= \frac{4}{\mu_h^2} \sum_{t=1}^{T-\tau} \frac{t + 1 + 4\kappa_h}{(t + 4\kappa_h) \left( (T - \tau + 1 + 4\kappa_h)(T - \tau + 2 + 4\kappa_h) - (t + 4\kappa_h)(t + 1 + 4\kappa_h) \right)} \\
&\leq \frac{8}{\mu_h^2} \sum_{t=1}^{T-\tau} \frac{1}{(T - \tau + 1 + 4\kappa_h)(T - \tau + 2 + 4\kappa_h) - (t + 4\kappa_h)(t + 1 + 4\kappa_h)} \\
&= \frac{8}{\mu_h^2} \sum_{t=1}^{T-\tau} \frac{1}{(T - \tau + 1 - t)(T - \tau + 2 + 8\kappa_h + t)} \\
&= \frac{8}{\mu_h^2} \sum_{t=1}^{T-\tau} \frac{1}{2T - 2\tau + 3 + 8\kappa_h} \left( \frac{1}{T - \tau + 1 - t} + \frac{1}{T - \tau + 2 + 8\kappa_h + t} \right) \\
&\leq \frac{8}{\mu_h^2 (2T - 2\tau + 3 + 8\kappa_h)} \left( 1 + \log(T - \tau) + \log \frac{2T - 2\tau + 2 + 8\kappa_h}{T - \tau + 2 + 8\kappa_h} \right) \\
&\leq \frac{8(1 + \log T)}{\mu_h^2 (T + 2 + 8\kappa_h)}, \tag{44}
\end{aligned}
$$

where we use $\frac{T}{2} \leq \tau \leq \frac{T+1}{2}$ in the last inequality.

Plugging (43) and (44) into (42) to get

$$
\begin{aligned}
&\mathbb{E}\left[ F(x^{T+1}) - F(x^*) \right] \\
&\leq \frac{\mu_h D_\psi(x^*, x^1)}{\exp\left( \frac{T}{2(1+\eta+2\kappa_h)} \right) - 1} + \frac{2(M^2 + \sigma^2)}{\mu_h} \left( \frac{12 \left( 2 + \frac{1}{\eta+2\kappa_h} \right)}{T + 5 + 16\kappa_h} + \frac{8(1 + \log T)}{T + 2 + 8\kappa_h} \right) \\
&= O\left( \frac{\mu_h D_\psi(x^*, x^1)}{\exp\left( \frac{T}{2(1+\eta+2\kappa_h)} \right) - 1} + \left( 1 \vee \frac{1}{\eta + 2\kappa_h} \right) \frac{(M^2 + \sigma^2) \log T}{\mu_h(T + \kappa_h)} \right). \tag{45}
\end{aligned}
$$

$\square$

**Theorem D.4.** *Under Assumptions 1-4 and 5B with $\mu_f = 0$ and $\mu_h > 0$, let $\kappa_h := \frac{L}{\mu_h} \geq 0$ and $\delta \in (0, 1)$:*

*If $T$ is unknown, by taking $\eta_t = \frac{2}{\mu_h(t + 4\kappa_h)}, \forall t \in [T]$, then with probability at least $1 - \delta$, there is*

$$
F(x^{T+1}) - F(x^*) \leq O\left( \frac{\mu_h(1 + \kappa_h)^2 D_\psi(x^*, x^1)}{T(T + \kappa_h)} + \frac{(M^2 + \sigma^2 \log \frac{1}{\delta}) \log T}{\mu_h(T + \kappa_h)} \right).
$$

*If $T$ is known, by taking $\eta_t = \begin{cases} \frac{1}{\mu_h(\eta+2\kappa_h)} & t \leq \tau \\ \frac{2}{\mu_h(t-\tau+4\kappa_h)} & t \geq \tau + 1 \end{cases}, \forall t \in [T]$ where $\eta \geq 0$ can be any number satisfying $\eta + \kappa_h > 0$ and $\tau := \lceil \frac{T}{2} \rceil$, then with probability at least $1 - \delta$, there is*

$$
F(x^{T+1}) - F(x^*) \leq O\left( \frac{\mu_h D_\psi(x^*, x^1)}{\exp\left( \frac{T}{2(1+\eta+2\kappa_h)} \right) - 1} + \left( 1 \vee \frac{1}{\eta + 2\kappa_h} \right) \frac{(M^2 + \sigma^2 \log \frac{1}{\delta}) \log T}{\mu_h(T + \kappa_h)} \right).
$$

*Proof.* When $\mu_f = 0$ and $\mu_h > 0$, suppose the condition of $\eta_t \leq \frac{1}{2L \vee \mu_f} = \frac{1}{2L}, \forall t \in [T]$ in Lemma 4.3 holds, we will have with probability at least $1 - \delta$

$$F(x^{T+1}) - F(x^*)$$

$$\leq 2\left(1 + \max_{2 \leq t \leq T} \frac{1}{1 - \mu_f \eta_t}\right)\left[\frac{D_\psi(x^*, x^1)}{\sum_{t=1}^T \gamma_t} + \left(M^2 + \sigma^2\left(1 + 2\log\frac{2}{\delta}\right)\right)\sum_{t=1}^T \frac{\gamma_t \eta_t}{\sum_{s=t}^T \gamma_s}\right]$$

$$= \frac{4D_\psi(x^*, x^1)}{\sum_{t=1}^T \gamma_t} + 4\left(M^2 + \sigma^2\left(1 + 2\log\frac{2}{\delta}\right)\right)\sum_{t=1}^T \frac{\gamma_t \eta_t}{\sum_{s=t}^T \gamma_s}. \tag{46}$$

Observing that

$$\gamma_t := \eta_t \prod_{s=2}^t \frac{1 + \mu_h \eta_{s-1}}{1 - \mu_f \eta_s} = \eta_t \prod_{s=2}^t (1 + \mu_h \eta_{s-1}) = \eta_t \Gamma_{t-1} = \frac{\Gamma_t - \Gamma_{t-1}}{\mu_h}, \forall t \in [T],$$

where $\Gamma_t := \prod_{s=1}^t (1 + \mu_h \eta_s), \forall t \in \{0\} \cup [T]$. So (46) can be rewritten as

$$F(x^{T+1}) - F(x^*) \leq \frac{4\mu_h D_\psi(x^*, x^1)}{\Gamma_T - 1} + 4\mu_h\left(M^2 + \sigma^2\left(1 + 2\log\frac{2}{\delta}\right)\right)\sum_{t=1}^T \frac{\eta_t^2}{\Gamma_T/\Gamma_{t-1} - 1}. \tag{47}$$

We can check that

$$\eta_t = \frac{2}{\mu_h(t + 4\kappa_h)} \leq \frac{1}{2\kappa_h \mu_h} = \frac{1}{2L}, \forall t \in [T];$$

$$\eta_t = \begin{cases} \frac{1}{\mu_h(\eta + 2\kappa_h)} \leq \frac{1}{2\kappa_h \mu_h} = \frac{1}{2L} & t \leq \tau \\ \frac{2}{\mu_h(t - \tau + 4\kappa_h)} \leq \frac{1}{2\kappa_h \mu_h} = \frac{1}{2L} & t \geq \tau + 1 \end{cases}.$$

Therefore, (47) is true for all cases.

If $\eta_t = \frac{2}{\mu_h(t + 4\kappa_h)}, \forall t \in [T]$, following the similar steps in the proof of (41), we can finally get

$$F(x^{T+1}) - F(x^*) \leq O\left(\frac{\mu_h(1 + \kappa_h)^2 D_\psi(x^*, x^1)}{T(T + \kappa_h)} + \frac{(M^2 + \sigma^2 \log\frac{1}{\delta})\log T}{\mu_h(T + \kappa_h)}\right).$$

If $\eta_t = \begin{cases} \frac{1}{\mu_h(\eta + 2\kappa_h)} & t \leq \tau \\ \frac{2}{\mu_h(t - \tau + 4\kappa_h)} & t \geq \tau + 1 \end{cases}, \forall t \in [T]$, following the similar steps in the proof of (45), we can finally get

$$F(x^{T+1}) - F(x^*) \leq O\left(\frac{\mu_h D_\psi(x^*, x^1)}{\exp\left(\frac{T}{2(1 + \eta + 2\kappa_h)}\right) - 1} + \left(1 \vee \frac{1}{\eta + 2\kappa_h}\right)\frac{(M^2 + \sigma^2 \log\frac{1}{\delta})\log T}{\mu_h(T + \kappa_h)}\right).$$

$\square$

