# OpenReview forum: "Revisiting the Last-Iterate Convergence of Stochastic Gradient Methods"
_ICLR.cc/2024/Conference — ICLR 2024 poster_

### Official Review · Reviewer_yV1Y · 2023-10-16

**Soundness:** 3 good
**Presentation:** 3 good
**Contribution:** 3 good
**Rating:** 8
**Confidence:** 3

**Summary:**

This work studies the last-iterate convergence of stochastic gradient methods for convex optimization. The considered algorithm is general enough to include common algorithms as special cases, and the considered setting is general enough to cover general domain/unbounded noise/high probability bound/wide range of losses. This work achieves near-optimal bounds under these general settings, by a simple yet powerful observation made in a previous work Zamani and Glineur 23. The key technique is to leverage the convexity of loss for proving the last-iterate convergence, by carefully designing a sequence $z_t$ to compare the loss values with $x_t$. As a result, the obtained technical lemma yields a unified analysis for the general settings.

**Strengths:**

This work makes a solid contribution to the understanding of the last-iterate convergence of stochastic gradient methods, which is an important problem is convex optimization and particularly gains interest from the ML community, since in practice the theoretically sub-optimal choice of the last iterate is cheaper and thus more popular.

The technical results are general and cover a wide range of settings, bypassing a few constraints of previous works including assumptions on compact domain and bounded noise.

The idea of constructing the sequence $z_t$ to make use of convexity is intuitive yet powerful.

Besides recovering the near-optimal convergence results under a more general setting, for smooth functions, this work provides an $\tilde{O}(1/\sqrt{T})$ bound which improves upon the previously known best result $O(1/T^{1/3})$.

**Weaknesses:**

I do not find any substantial weakness. One constraint of this work is that it lacks a proof sketch or discussion on the main idea in the main-text. I believe the paper can benefit from adding a simplest example of $z_t$, explaining how it's used to utilize convexity. Doing so can improve the readability by giving the readers more intuitions on the design of $z_t$ and how it works.

**Questions:**

It looks to me the presentation of results currently takes up a lot of space. Is it possible to make it more compact and add more discussions on the main idea of using $z_t$?

---

> ### Author Response · Authors · 2023-11-13
> **Official Response**
>
> We thank the reviewer for the positive comments and endorsement of our results. We would like to answer your questions and the weaknesses below.
>
> **Weaknesses&Questions**: Thanks for your valuable suggestions. We are currently working on the revision to improve the organization and readability. As you advised, we are trying to add more discussions on the sequence $z_{t}$. It may take some time to coordinate the changes suggested by you and other reviewers. Once the revision is done, we will update the response to let you know.
>
> **Update**: Following the reviewer's advice, we added some sentences in the second revision to describe the idea of using $z^t$ at the end of the second paragraph in Section 4 on page 8. We are planning to add more explanations if more space is permitted in the final version.

---

> > ### Comment · Reviewer_yV1Y · 2023-11-22
> >
> > Thanks for the reply. Although I kinda agree with other reviewers' opinion that the technical novelty is a bit oversell, I still think it's a nice result and will keep my score (with a lower confidence).

---

> > > ### Author Response · Authors · 2023-11-22
> > > **Thanks for the feedback**
> > >
> > > We sincerely thank the reviewer for the supportive comment. We will continue to polish the paper and improve the quality of our work.

---

### Official Review · Reviewer_4SRm · 2023-10-31

**Soundness:** 2 fair
**Presentation:** 3 good
**Contribution:** 2 fair
**Rating:** 6
**Confidence:** 2

**Summary:**

The work presents the last-iterative convergence of stochastic gradient methods  in expectation and in high probability on general domains not necessarly bounded.

**Strengths:**

The work presents high-probability and in expectation convergence result for the last iterate of SGD in general domains for convex or strongly convex objectives

**Weaknesses:**

-In find the work incremental compared to the litterature. In fact, the work generelises the convergence results of the last iterate SGD for convex or strongly convex objectives to the general domains not necessarly compact. I find the content of the paper more adapted to be publisehd in a math/optimisation journal than ICLR.

**Questions:**

-It will be good to add discussion about your techniques to prove the last iterate SGD convergence for non convex objectives.
-Bounded variance: This assumption contradicts strong convexity. I understand it is in several works in the literature. Take for instance F(x,y,z) = zx^2 + (1-z)y^2, z follows Bernoulli (1/2), then E_z(norm(\nabla F(x,y,z) - \nabla E_z(F(x,y,z)))^2) = (x-y)^2 and this is not bounded for all x and y in R.
-You mentioned in your criticism to the SOTA in the abstract that " the existing results are all limited to a single objective..", as far as I can check you are also considering only a single objective, not multi objectives. Composite optimization problem still a single objective?
-I did not check the proofs carefully.

---

> ### Author Response · Authors · 2023-11-13
> **Official Response**
>
> We thank the reviewer for the valuable feedback. We would like to address the reviewer's concerns below.
>
> **Weaknesses**: We want to highlight some points of our study to respond to the reviewer's concerns.
>
> 1. First, understanding the convergence of stochastic gradient methods serves a fundamental position in the ML community. In particular, proving the convergence of the last iterate is an important problem as also pointed out by the other two reviewers. We believe a good understanding of the plain stochastic gradient methods (e.g., SGD) will be beneficial to studying other more advanced algorithms like momentum SGD and adaptive algorithms including AdaGrad, Adam, etc.
>
> 2. Second, our paper aims at solving the convergence of the last iterate for much broader function classes and general domains, which are more challenging compared with the previous studies based on Lipschitz functions and bounded domains. Especially when considering the high-probability convergence, proving convergence in general domains is generally harder than in bounded domains. Even for the averaged output, proving the high-probability rate without compactness has only recently been resolved by [1] as far as we know. Hence, we would like to argue that our work is an important step in filling in the missing pieces in the related literature.
>
> 3. Additionally, our analysis framework, which could be of independent interest, is simple yet useful. As mentioned in the paper, it can help us bypass several restrictive conditions assumed in prior works. We hope it can shed further light on the convergence of the last iterate of other algorithms used for convex optimization.
>
> 4. Last but not least, our results also have an important practical implication that is to provide theoretical justification for using the last iterate as the choice (which is cheaper and popular though sub-optimal as mentioned by Reviewer yV1Y).
>
> Therefore, we believe the results in the paper are interesting enough to build upon for future work.
>
> **Q1**: We remark that the key in our proof is to utilize the convexity of the objective as described at the beginning of Section 4 (before Lemma 4.1), which means this technique cannot be generalized to non-convex objectives. Additionally, as far as we know, the stochastic gradient methods generally cannot guarantee convergence in the function value gap without convexity unless some special assumptions (e.g., the Polyak-Lojasiewicz condition) hold. Hence, we think proving the convergence of the last iterate for the non-convex functions is beyond the scope of this paper.
>
> **Q2**: (We sincerely apologize for the following mistake in the first version of our response) ~~We do not understand your example very well due to the following two reasons. Could you explain more about your point?~~
>
> ~~1. In your example, we can find $\nabla^{2}F(x,y,z)=\mathrm{diag}(2z,2(1-z))$, which is only PSD for any realization of $z\in\left\\{ 0,1\right\\}$. Hence $F(x,y,z)$ itself is only convex but not strongly convex.~~
>
> ~~2. Moreover, it only requires the expectation of $F(x,y,z)$ to be strongly convex in our setting. However, there is $f(x,y)\coloneqq\mathbb{E}_{z}\left[F(x,y,z)\right]=0$, which is not strongly convex.~~
>
> We remark that it is always possible to construct unbiased stochastic gradients with unbounded variance regardless of what conditions are required to hold for the objective function $F(x)$. For example, one can always consider the stochastic gradient as $\widehat{\nabla}F(x)\coloneqq\nabla F(x)+\xi$ where $F(x)$ can be any objective and $\xi$ is a centered r.v. only admitting finite $\alpha$-th moment where $\alpha\in(1,2)$ (e.g., $\alpha$-stable distribution). However, this counterexample does not mean the bounded variance condition should not be assumed. Arguably, it is the most standard and common condition when proving the expected convergence and appeared in lots of research related to stochastic optimization. Hence, we think the finite variance assumption is reasonable.
>
> **Q3**: Thanks for pointing it out. The phrase "a single objective" has been changed to "a non-composite objective" in the first revision.
>
> **Q4**: We believe our proofs are correct and are happy to answer the related questions if the reviewer has any in the future.
>
> **References**
>
> [1] Liu et al. High probability convergence of stochastic gradient methods. ICML 2023.

---

> > ### Comment · Reviewer_4SRm · 2023-11-20
> > **Bounded variance**
> >
> > f(x,y) is not equal to zero! $f(x,y) = 1/2 x^2 + 1/2 y^2$ is strongly convex. My question is about the contradiction between the bounded variance assumption and the strong convexity assumption. The simple example I gave you is to show this!

---

> > > ### Author Response · Authors · 2023-11-20
> > > **Thanks for the feedback**
> > >
> > > We highly appreciate the reviewer for the feedback!
> > >
> > > - Thanks for pointing it out. Your example is correct and we sincerely apologize for our mistake. We agree that your construction gives a counterexample when considering the bounded variance and strong convexity assumptions. We will delete the corresponding part in the response accordingly.
> > >
> > > - However, as mentioned in our response. It is indeed possible to construct stochastic gradients without bounded variance for any function. But this does not mean that bounded variance should not be assumed. Arguably, it is the most standard assumption.
> > >
> > > - To summarize, we think both your example and bounded variance assumption are meaningful. In this work, our goal is to show the last-iterate convergence under the bounded variance assumption. For future work, we will try to relax it and prove the last-iterate convergence under more general assumptions (e.g., your example).

---

> > > > ### Comment · Reviewer_4SRm · 2023-11-20
> > > >
> > > > I already mentioned in my first comment that I know that this hypothesis is found in several works of literature. This is not enough to say that the hypothesis is too restrictive or not. For stochastic optimization, except special cases where you cannot have access to the exact gradient, all you can do is sample as in my example so no guarantee on the variance. Even if we can calculate the exact gradient, I don't see the need to add noise to this to have an SG, just use the gradient directly. So can you give some practical examples where we can have guarantees on this hypothesis.

---

> > > > > ### Author Response · Authors · 2023-11-20
> > > > > **Reply to Official Comment by Reviewer 4SRm**
> > > > >
> > > > > Thanks for the question. Here we give some practical examples where the finite variance assumption holds.
> > > > >
> > > > > Let's consider the general linear classification problem in the form of $f(x)\coloneqq\mathbb{E}_{(Z, Y)\sim\mathcal{D}}[\ell(\langle x, Z \rangle, Y)]$ where $\ell$ is some loss function, $Z\in\mathbb{R}^d$ denotes the feature, $Y\in\\{\pm1\\}$ is the label, and $\mathcal{D}$ is the distribution of the data. We additionally assume the feature $Z$ has bounded support, i.e., $\\|Z\\|_2\leq r$ almost surely for some $r>0$. Now we consider the following two popular loss functions:
> > > > >
> > > > > - Logistic Loss: In this case, we have $f(x)=\mathbb{E}_{(Z,Y)\sim\mathcal{D}}[\log(1+\exp(-Y\langle x, Z \rangle))]$. For any given $(Z_i,Y_i)$, we can find the stochastic gradients as $\widehat{\nabla}f(x)=-Y_iZ_i/(1+\exp(Y_i\langle x,Z_i\rangle))$. Note that $\\|\widehat{\nabla}f(x)\\|_2\leq \\|Z_i\\|_2\leq r$, hence there is $\mathbb{E}[\\|\widehat{\nabla}f(x)-\nabla f(x)\\|_2^2]\leq 4r^2,\forall x\in\mathbb{R}^d$, which implies a bounded variance.
> > > > >
> > > > > - Hinge Loss: In this example, we have $f(x)=\mathbb{E}_{(Z,Y)\sim\mathcal{D}}[\max(0,1-Y\langle x,Z\rangle)]$. For any given $(Z_i,Y_i)$, we can find the stochastic gradients as $\widehat{\nabla}f(x)=-Y_iZ_i\mathbb{I}[Y_i\langle x,Z_i\rangle\leq 1]$. Note that $\\|\widehat{\nabla}f(x)\\|_2\leq \\|Z_i\\|_2\leq r$, hence there is $\mathbb{E}[\\|\widehat{\nabla}f(x)-\nabla f(x)\\|_2^2]\leq 4r^2,\forall x\in\mathbb{R}^d$, which implies a bounded variance.

---

### Official Review · Reviewer_V1N7 · 2023-11-07

**Soundness:** 3 good
**Presentation:** 2 fair
**Contribution:** 3 good
**Rating:** 6
**Confidence:** 4

**Summary:**

The paper focuses on a fundamental problem on the convergence of SGD. Specifically, the main novelty is high probability, last iterate rates for SGD without bounded domain assumptions for solving convex problems. While doing so, the paper also presents unified results for composite problems / Bregman distances / convex or strongly convex problems / smooth or Lipschitz problems and expectation or high probability guarantees.

**Strengths:**

As implicitly stated in my "Summary", I do think that the goal of the paper is interesting. A result involving high probability guarantees for the last iterate of SGD for convex problems is interesting in my opinion. The proofs are comprehensive and mostly carefully written (even though there are readability issues I will expand on in the later parts of my review). Having a unified result is also nice to cover different important settings.

**Weaknesses:**

Even though I think the main result of high probability rates for last iterate of SGD without bounded domains is interesting and worthy of acceptance (once the correctness is verified) there are many issues with the writing of the paper and proofs that should also be addressed which prevented me to be able to verify the correctness. Right now, the repeating theme in the paper is for the authors to spend way too much time and effort to show their improvements in marginal cases, which confuses reader to miss their main contribution (and also at times misleading comparisons with related works). In addition, the generality and unifications that the authors are trying to achieve (unification is a good thing as I said before but it should be presented in a correct way), makes the proofs unnecessarily complicated and difficult to check. I have suggestions to improve this.

The main reason for my rating is because I could not verify the proof for the main result of the paper in a reasonable time-span with many questions about technical aspects (given that as reviewers we need to review 4-6 papers in less than 20 days in addition to our regular work load). As a result, I cannot be convinced yet about the correctness. Hence, with my suggestions below, I am giving the authors the opportunity to clarify their main contribution, present their main contribution in a readable way, answer my questions about the proofs and help me verify the correctness which is required for acceptance. Once and if this is done, I will be willing to raise my score.

In particular, the issue is that the authors are trying too hard in the paper to "sell the result" which causes them to write the proofs in such a generality that it makes it hard to verify the result without investing days. The results of the paper should speak for themselves since high probability guarantees for last iterate without bounded domains for such an important algorithm SGD is important. I already invested days to clarify the exact contribution of the paper because the authors are trying to claim novelty in all the presented results, for this they just generalize things so much so that they can claim the novelty. But in some cases, some of these generalizations are straightforward and just degrade readability. Even though the paper definitely cites and states that they use the techniques from Zamani and Glineur, 2023 and Liu et al., 2023, they spent too much effort trying to justify their analysis is still novel.

For example, before Lemma 4.1, the authors give an effort to describe a high level message of Zamani and Glineur, 2023, but they say "This brand new analysis is where we different from all the previous works". It is unclear what are the "previous work" or what is "brand new". If there is a difference on this part of the analysis from Zamani and Glineur, 2023, please specify what it is, if not, this sentence should be clarified. It is okay to use a tool from another paper, Zamani and Glineur, 2023 in this case, but one should describe clearly what is used directly from that paper and what is modified and what the precise modification is.

Because of the unclarity in the writing of the paper, I had to spend a considerable amount of time checking the results of Zamani and Glineur, 2023, Liu et al., 2023, Orabona 2020 and Shamir, Zhang 2013 to identify precisely what the contribution is. At the end, writing of the paper felt like an oversell to me and misleading at times even though I still find the main contribution interesting (high probability, last iterate without bounded domains).

After checking these 3 papers, I saw that we know (1) expectation rates for last iterate of SGD without bounded domains since Orabona, 2020, (2) we know high probability rates for average iterate of SGD without bounded domains since Liu et al., 2023. The contribution of the paper is to unify these two. However, the paper tries to also claim credit for doing (1) i.e., expectation guarantees for composite problems or Bregman distances or smooth problems: this is not good.

Because looking at the blog post Orabona, 2020, it is clearly written that Bregman extension and using smoothness instead of Lipschitzness are left as "exercises" to the reader. Also, looking at Orabona, 2020 analysis which builds on Shamir, Zhang, 2013, I agree with the blog post that these extensions (composite, Bregman, Lipschitz gradient) should be straightforward. If the authors disagree, they should clearly write why Orabona, 2020's approach does not generalize. Otherwise, the contribution of the paper is still enough without trying to justify every little thing. The authors can still add a remark saying that these were not explicit in Orabona, 2020 and they have these generalizations, but they should not emphasize this and focus on the main contribution of the paper: high probability guarantees for last iterate of SGD for convex problems without bounded domains.

*** As such, what I recommend the authors to do is the following: as the main result of the paper, identify the simplest result that still captures the novelty by forgoing generality. In my view, the authors can just focus on a constrained problem with Euclidean distances, Lipschitz continuous (or smooth) convex function and provide a self-contained proof for high probability convergence of last iterate without bounded domains in this simplified setting. This way, both a reviewer such as myself and the readers of the paper can quickly verify the main result of the paper and appreciate the main contribution. The authors can then provide statements for the general cases with strong convexity/smoothness/composite cases/Bregman distances etc. and show the "unification" and provide their generalized proofs in the appendix (they can state the generalized result in the main text if they have the space). But what is needed is to distill the main contribution and give a digestible statement and proof for the reader so I can verify, right now there are many things I could not verify (written below). Right now, I have to go through so many generalized results to verify the correctness of the result which is tedious and time-consuming with the extensive review load.

**Questions:**

+ Are the results of Zamani, Glineur 2023 and Liu et al., 2023 work well together to get the high probability rate on the last iterate without bounded domains, or do additional difficulties arise while combining these results? If so, what are these difficulties? To clarify, I think the paper is worthy of acceptance even if the combination of these tools do not create additional difficulties. Identifying 2 interesting techniques and deriving an interesting result (as stated before I think the result of the paper is interesting) is enough for acceptance in my opinion. But this should be clearly written.

+ Before Sec. 4 the authors say "to the best knowledge, our results are entirely new and the first to prove ....", there is no need for such repetition or such exaggerated words as "entirely new", the sentence meaning is the same if you just say, "to our knowledge our results are the first to prove ....". Also the beginning of the sentence should be "to our best knowledge". Similarly, before Lemma 4.1 the authors use "brand new analysis" which is unnecessary and also unclear if it refers to the analysis of Zamani and Glineur, 2023 or this paper.

+ In my understanding, the analysis of Zamani, Glineur 2023 seems like an alternative to Orabona, 2020 who built on Shamir, Zhang, 2013. Why is the analysis in Orabona, 2020 not sufficient to extend to high probability guarantees? Or do the authors think similar results can be shown with Orabona, 2020's analysis too? Explaining this will help the reader understand what is going on. Put another way, what is the reason that the paper builds on Zamani, Glineur 2023 instead of Orabona, 2020 for the last iterate of SGD - expectation guarantees without bounded domains?

+ Throughout, the authors use notation such as (this one is for example from Lemma 4.1 but it is like this all around the paper and the proofs) $\eta_{t\in [T]} \leq \frac{1}{2L \vee \mu_f}$ which is convoluted, difficult to read and non-standard, whereas they can simply write in a standard way as $\eta_t \leq \frac{1}{2L \vee \mu_f}$ for $t \in [T]$. Things become much more cluttered when the authors start writing things like (again Lemma 4.1) $v_{t\in \{ 0 \}\cup [T]} > 0$, this is unnecessarily complicated and not very readable, please consider rewriting like $v_t > 0$ for $t \in \{ 0, 1, \dots, T\}$, which is much easier to read.

+ In Lemma 4.1 and other places, author also use definitions such as $v_t = \frac{w_T \gamma_T}{\sum_{s=1\vee t}^T w_s\gamma_s}$, why not just define $v_0$ separately and just write the sum starting from $s=t$ like normal? This is much more readable than seeing $t\vee 1$ every time for such things.

+ Please describe clearly in which ways Lemma 4.2 improves over the result of Orabona, 2020 since Bregman case or smooth case is just left to readers as exercises in Orabona, 2020.

+ Many places, such as the footnote in page 6 says that "Harvey et al. (2019a) claims their proof can extend do sub-Gaussian noises ...." Or in the beginning of page 2: "whether these two assumptions (referring to bounded noise) remains unclear" This is rather strange as it gives the impression that the authors think the claim of Harvey et al. (2019a) is not correct. If so, please tell the readers why you believe their proof does not extend to sub-Gaussian noises. Also, it is so easy to write to the authors of Harvey et al. (2019a) an email to ask them the extension for sub-Gaussian case, if you are not convinced. It is not very nice to write it like this in a paper without checking with the authors of the existing paper to clarify their claims. Looking at Harvey et al. (2019a) I also think that their result is extendable to sub-Gaussian as they also described clearly what would change and how the changes would be handled. Again, the current submission does not need to focus on such minor issues, even if Harvey et al. (2019a)'s claim is true (which is in my opinion), the submission's contribution still stands since they get rid of the bounded domain assumption.

+ What are the main technical novelties in addition to Liu et al., 2023? Please describe this in Sec. 1.2. The two works are quite related and this section does not clearly state this.

+ Page 3 mentions "smoothness with respect to optimum", what do the authors refer to by this phrase?

+ page 4, you can just use $\mathcal{X}=\mathbb{R}^d$ and define the domains of $f,g$ in standard ways to get rid of all the discussion about int(dom) etc. These are standard problem setups, it is better not to complicate things unnecessarily.

+ page 4 and many places of the paper refer to the book Lan, 2020 for some results. Please clearly specify what you refer to in this book so that the readers know where to look at, we cannot expect readers to browse a whole book to understand what the authors are using from this book.

+ Sec 2.1 is unnecessary and should be moved to the appendix. Moreover, the sentence starting with "We also would like to emphasize that ...." Reads strange, please consider rewriting.

+ Theorem 3.1 has a step size choice depending on $x^*$, this is not good and should be avoided. The authors say after the statement that things work with any $\eta$ independent of $x^*$, if so, write things that way in the main text. Of course, a result needing distance $x^*$ to set step size is definitely not desirable.

+ The discussion after Theorem 3.1 says also that "Orabona 2020 cannot be applied to composite optimization", why? What breaks in their argument? It is not clear to me why Orabona, 2020 would not extend to composite setting in a straightforward way. If the authors make such claims such "cannot be applied" for previous results, they should explain why. Again, whether or not Orabona, 2020 can be extended to composite is not essential for this submission. The main contribution of the submission is independent of this. By putting phrases like this, the authors only confuse their reader. They do not have to claim novelty for every single result contained in the paper (in some cases they can just recover existing ones, this is okay), the authors should clarify their main contribution, which is high probability guarantees for last iterate of SGD for convex problems.

+ Page 5 also claims that the results are the first improvement over Moulines, Bach 2011 for smooth problems, whereas Orabona, 2020 just left it as an exercise to extend their results for smooth problems. If the authors think that Orabona, 2020's analysis does not work with smoothness, they should write why they think so. Otherwise, this is misleading for a reader and only distracts from their main contribution of the submission.

+ Before Theorem 3.2, the authors mention recovering $L/T$ rate in the noiseless case. It seems that this is not true with any $\eta$ but only the $\eta$ depending on $x^*$, can the authors clarify?

+ The case of strongly convex is strange since in this case the bounded gradient assumption (that is, Lipschitzness of the convex function) and strongly convex assumption contradict each other. This is well-known, see for example Section 1 of "SGD and Hogwild! Convergence Without the Bounded Gradients Assumption" by Lam M. Nguyen, Phuong Ha Nguyen, Marten van Dijk, Peter Richtarik, Katya Scheinberg, Martin Takac. In this case, what is the significance of the strongly convex and bounded gradient case? Of course, the unified result also contains cases where we do not have bounded gradient and strong convexity together, but this should be explained well and be separate from the main contribution of the paper.

+ Paper uses the phrase at many places (including the title) "last-iterative convergence", this is not a correct terminology, also probably not a correct grammar, it should be "last-iterate convergence". It is used many times and disrupts the flow, please change it throughout.

+ Please explain the 5th line in the chain of equations in the beginning of the proof of Lemma 2.1.

+ Proof of Lemma 4.1. Explain the requirements of $v_k$ in the beginning of the proof. Add the reason why $z_t \in \mathcal{X}$, that is: because it is a convex combination of points $x^*, x^1, ..., x^t$ that are in $\mathcal{X}$.

+ beginning of page 15, fourth line is difficult to follow, please write clearly.

+ After eq. (7), when authors use convexity of $F$ with $z^t$, $z^t$ is not written in a proper way for the reader to understand it is a convex combination of points. Please write the correct representation to improve readability. That is, you need to recall that $v_s$ is monotone and weights sum up to $1$. Zamani, Glineur, 2023 writes this clearly for example.

+ Last inequality chain in page 15, please explain the last step.

+ Page 17, the estimation of $2w_t\gamma_t \eta_t v_t$, please explain the first inequality. Also refer here to the definition of $\bar v$ to improve readability.

+ Beginning of page 18 while using Lemma 2.1, clarify what refers to $\lambda$, $Z$ in the lemma and why the requirement in Lemma 2.1 on $\lambda$ is satisfied here.

+ Beginning of page 19: $w_1$ is undefined since $w_t$ contains a sum starting from $2$. Of course it is implicitly clear what it is, please define $w_1$ properly.

+ Page 19, bound of $w_1/w_T$, please explain the inequality.

+ After Lemma 4.2, the authors again say "first unified inequality for the in-expectation" and after Lemma 4.3, they say the same for high probability, I understand that the authors want to emphasize the novelties but after a point it gets tedious for the reader and gives the feeling of "oversell". Hence, please try to be more concise, state the novelty once or twice in the paper and then tell the readers main contributions clearly.

---

> ### Author Response · Authors · 2023-11-13
> **Official Response**
>
> We highly appreciate the reviewer's detailed feedback and thank you for being interested in our unified analysis. To make our response more clearly correspond to your questions, we will label your questions from 1 to 28 in the order. Additionally, our comments will be separated into the following three parts:
>
> - In part I, we will try our best to help the reviewer verify the proof's correctness and focus mainly on the questions related to assumptions/theorems/steps in the proof (i.e., the technical side).
>
> - In part II, we would like to address the questions that are more about writing and readability.
>
> - In part III, we give a summary to clarify the difference compared with prior works mentioned by the reviewer in the weakness part/the remaining questions.

---

> ### Author Response · Authors · 2023-11-13
> **Official Response Part I (1/3)**
>
> We first provide some comments on the importance of different lemmas/theorems here to help the reviewer better understand the work. The most important three results are Lemmas 4.1, 4.2 and 4.3, which are keys to proving every main theorem. Lemma 2.1 is less important since it is only a generalization of the standard property of sub-Gaussian random variables to sub-Gaussian random vectors. It also appeared in prior works as mentioned in the paper. The other left proofs are less important. What we do is to plug in different step sizes and find out the final convergence rate with some careful calculations. Therefore, to verify the correctness of proofs and understand our results with the smallest efforts, it is enough to go through the proofs of Lemma 4.1, 4.2 and 4.3 (of course, the proof of Lemma 2.1 is also worth reading if one wants to check it).
>
> Next, we address the specific questions asked by the reviewer below. However, since the manuscript will be updated again later, the following answers to the questions that include the number of pages or lines may be only valid for the version updated on Nov 12 (one can use the Revisions button to find the corresponding version).
>
> ---
>
> >**Q9**: Page 3 mentions "smoothness with respect to optimum", what do the authors refer to by this phrase?
>
> **A9**: It means $f(x)-f(x^{\*})\leq\frac{L}{2}\\|x-x^{\*}\\|^2, \forall x\in\mathcal{X}$ for some $L>0$ (the original definition can be found in Eq. (2) in [1]. Here we state it in our notations). With this extra assumption, [1] transferred the well-known expected rate $\mathbb{E}\left[\\|x^{T}-x^{*}\\|\right]=O(\frac{1}{T})$  for the Lipschitz and strongly convex functions into the convergence w.r.t. the function value gap. However, this smoothness with respect to optimum condition may not hold in general. We have added a footnote to clearly state the definition in the first revision.
>
> ---
> >**Q10**: page 4, you can just use $\mathcal{X}=\mathbb{R}^{d}$ and define the domains of $f,g$ in standard ways $\cdots$.
>
> **A10**: We refer the reviewer to the answer **A17** for why we consider $\mathcal{X}$ to be a general closed convex set instead of setting it to be $\mathbb{R}^{d}$ directly.
>
> ---
> >**Q13**: Theorem 3.1 has a step size choice depending on $x^*$, this is not good and should be avoided $\cdots$.
>
> **A13**: Sorry for the confusion. Here we mean that the algorithm can still guarantee the convergence of the last iterate for any $\eta>0$ rather than any $\eta>0$ leads to the same convergence rate as in Theorem 3.1. In other words, even if we don't know the Bregman divergence w.r.t. $x^*$, the last iterate still provably converges. But with $\eta_*$ (which depends on the knowledge of $M$,$\sigma$ and $D_{\psi}(x^*,x^1))$, we will get better dependence on the parameters in the final rate. We refer the reviewer to Theorem C.1 for a detailed result. The discussion here has been clarified in the first revision.
>
> Moreover, we would like to keep the current form of Theorem 3.1 since the full version (Theorem C.1) is rather long but we only have limited space for the main text.
>
> ---
> >**Q16**: Before Theorem 3.2, the authors mention recovering rate $L/T$ in the noiseless case $\cdots$.
>
> **A16**: From Theorem C.1, we have the following convergence rate for smooth functions (taking $M=0$) with the step size $\eta_t=\frac{1}{2L}\land\frac{\eta}{\sqrt{t}}$ where $\eta>0$ can be any number, $\mathbb{E}\left[F(x^{T+1})-F(x^*)\right]\leq O\left(\frac{LD_{\psi}(x^*,x^1)}{T}+\frac{1}{\sqrt{T}}\left(\frac{D_{\psi}(x^*,x^1)}{\eta}+\eta\sigma^2\log T\right)\right)$. With this result, we would like to clarify two points:
>
> - As you pointed out, it is not true to recover the rate $O(L/T)$ in the noiseless case for any $\eta>0$.
>
> - But if $\eta\coloneqq\frac{\bar{\eta}}{\sigma}$ where $\bar{\eta}>0$ can be any number, we can obtain the rate $\mathbb{E}\left[F(x^{T+1})-F(x^*)\right]\leq O\left(\frac{LD_{\psi}(x^*,x^1)}{T}+\frac{\sigma}{\sqrt{T}}\left(\frac{D_{\psi}(x^*,x^1)}{\bar{\eta}}+\bar{\eta}\log T\right)\right)$. Hence, even if the step size does not depend on $x^*$, the rate can still be $O(L/T)$ in the noiseless case. Of course, the choice $\eta=\frac{\bar{\eta}}{\sigma}$ is highly impossible in practice since it requires to know $\sigma$. But from the theoretical side, a step size in this form is commonly used to obtain the noise adaptive rate (not limited to convex optimization). For example, see the step size on Page 121 (under the equation (4.1.32)) in [2] or Theorem 3.6 and Theorem 4.1 in [3].

---

> ### Author Response · Authors · 2023-11-13
> **Official Response Part I (2/3)**
>
> >**Q17**: The case of strongly convex is strange since in this case the bounded gradient assumption $\cdots$.
>
> **A17**: We do not think the strong convexity contradicts the bounded gradient assumption due to the following reasons. In our opinion, what [4] conveyed is that the strong convexity and the bounded gradient assumption contradict each other when $\mathcal{X}=\mathbb{R}^d$ meaning that the objective $f$ can not enjoy these two properties at the same time on $\mathbb{R}^{d}$. However, this does not imply these two conditions can not be assumed locally on the constraint set $\mathcal{X}$. Taking a simple non-composite example on $\mathbb{R}$, let $f(x)=\frac{1}{2}x^2$ and $\mathcal{X}=[-1,1]$. Then $f(x)$ is both strongly convex on $\mathcal{X}$ (of course, it is also strongly convex on $\mathbb{R}$) and Lipschitz on $\mathcal{X}$ with Lipschitz parameter $1$. This is also why our assumptions are described to be held only on $\mathcal{X}$ instead of $\mathbb{R}^d$.
>
> Moreover, we would like to emphasize that not requiring compactness on $\mathcal{X}$ does not mean $\mathcal{X}$ must be unbounded. If the problem indeed requires a bounded set (for example, even if we require $f$ to be locally strongly convex and Lipschitz on $\mathcal{X}$. Then it is well-known $\mathcal{X}$ should be bounded, e.g., see Lemma 2 in [1]), then it can be bounded. Our contributions regarding the general domains are more about saying that:
>
> 1. The constraint set $\mathcal{X}$ can be arbitrarily set once it does not trigger any contradictions to other assumptions rather than requiring that $\mathcal{X}$ must be unbounded.
>
> 2. We do not require to know whether $\mathcal{X}$ is compact or not in the analysis.
>
> This is also the reason we do not let $\mathcal{X}=\mathbb{R}^d$ directly.
>
> ---
> >**Q19**: Please explain the 5th line in the chain of equations in the beginning of the proof of Lemma 2.1.
>
> **A19**: From the 4th line to the 5th line, what we did was relabelling the index $k$ and aggregating the same $k$ in the summation. That is:
> - $\sum_{k=1}^{\infty}\frac{\\|\xi\\|_*^{2k}\\|Z\\|^{2k}}{(2k)!}=\frac{\\|\xi\\|\_*^2\\|Z\\|^{2}}{2}+\sum\_{k=2}^{\infty}\frac{\\|\xi\\|\_*^{2k}\\|Z\\|^{2k}}{(2k)!};$
>
> - $\sum_{k=1}^{\infty}\frac{\\|\xi\\|_*^{2k}\\|Z\\|^{2k}}{(2k+1)!}=\frac{\\|\xi\\|\_*^{2}\\|Z\\|^{2}}{6}+\sum\_{k=2}^{\infty}\frac{\\|\xi\\|\_*^{2k}\\|Z\\|^{2k}}{(2k+1)!};$
>
> - $\sum_{k=1}^{\infty}\frac{\\|\xi\\|_*^{2k+2}\\|Z\\|^{2k+2}/4}{(2k+1)!}=\sum\_{k=2}^{\infty}\frac{\\|\xi\\|\_*^{2k}\\|Z\\|^{2k}/4}{(2k-1)!}.$
>
> By summing up the above three results, the equation between the 4th line and the 5th line holds.
>
> ---
> > **Q21**: beginning of page 15, fourth line is difficult to follow, please write clearly.
>
> **A21**: What we did was relabelling the index $t$ and aggregating the same $t$ in the summation
> \begin{aligned}
> 	&\sum_{t=1}^T w_t \gamma_t (\eta_t^{-1}-\mu_f)v_{t-1}D_{\psi}(z^{t-1},x^t)-w_t \gamma_t (\eta_t^{-1}+\mu_h)v_t D_{\psi}(z^t,x^{t+1})\\\\
> =&\sum_{t=1}^T w_t \gamma_t (\eta_t^{-1}-\mu_f)v_{t-1}D_{\psi}(z^{t-1},x^t)-\sum_{t=2}^{T+1}w_{t-1}\gamma_{t-1}(\eta_{t-1}^{-1}+\mu_h)v_{t-1}D_{\psi}(z^{t-1},x^t)\\\\
> =&w_1\gamma_1(\eta_1^{-1}-\mu_f)v_0D_{\psi}(z^0,x^1)-w_T\gamma_T(\eta_T^{-1}+\mu_h)v_T D_{\psi}(z^T,x^{T+1})\\\\
> &+\sum_{t=2}^T\left(w_t \gamma_t(\eta_t^{-1}-\mu_f)-w_{t-1}\gamma_{t-1}(\eta_{t-1}^{-1}+\mu_h)\right)v_{t-1}D_{\psi}(z^{t-1},x^t).
> \end{aligned}
>
> ---
> >**Q23**: Last inequality chain in page 15, please explain the last step.
>
> **A23**: What we did was relabelling the index t, changing the order of the summation and aggregating the same t in the summation. That is
> \begin{aligned}
> &\sum_{t=1}^T\left[w_t\gamma_tv_t(F(x^{t+1})-F(x^*))-w_t\gamma_t\sum_{s=1}^t(v_s-v_{s-1})(F(x^s)-F(x^*))\right]\\\\
> =&\underbrace{\sum_{t=1}^Tw_t\gamma_tv_t(F(x^{t+1})-F(x^*))}\_A-\underbrace{\sum_{t=1}^T\sum_{s=1}^{t}w_t\gamma_t(v_s-v_{s-1})(F(x^s)-F(x^*))}\_B.
> \end{aligned}
> We know $$A=\sum_{t=2}^{T+1}w_{t-1}\gamma_{t-1}v_{t-1}(F(x^t)-F(x^*))=w_T\gamma_Tv_T(F(x^{T+1})-F(x^*))+\sum_{t=2}^Tw_{t-1}\gamma_{t-1}v_{t-1}(F(x^t)-F(x^*)),$$ and
> \begin{aligned}
> B&=\sum_{s=1}^T\left(\sum_{t=s}^Tw_t\gamma_t\right)(v_s-v_{s-1})(F(x^s)-F(x^*))=\sum_{t=1}^T\left(\sum_{s=t}^Tw_s\gamma_s\right)(v_t-v_{t-1})(F(x^t)-F(x^*))\\\\
> &=\left(\sum_{t=1}^{T}w_t\gamma_t\right)(v_1-v_0)(F(x^1)-F(x^*))+\sum_{t=2}^T\left(\sum_{s=t}^Tw_s\gamma_s\right)(v_t-v_{t-1})(F(x^t)-F(x^*)).
> \end{aligned}
> Putting all together to obtain the desired result.
>
> ---
> >**Q24**: Page 17, the estimation of $2w_t\gamma_t\eta_tv_t$, please explain the first inequality $\cdots$.
>
> **A24**: The inequality holds due to $\sum_{s=2}^t\frac{2\gamma_s\eta_s\bar{v}_s\sigma^2}{1-\mu_f\eta_s}\geq0,\forall t\in[T]$ and $\sum\_{s=1}^T2\gamma_s\eta_s\bar{v}_s\sigma^2\geq2\gamma_t\eta_t\bar{v}_t\sigma^2$. The reference to the definition has been added to the inequality chain as you suggested in the first revision.

---

> ### Author Response · Authors · 2023-11-13
> **Official Response Part I (3/3)**
>
> >**Q25**: Beginning of page 18 while using Lemma 2.1, clarify what refers to $\lambda, Z$ in the lemma and why the requirement in Lemma 2.1 on $\lambda$ is satisfied here.
>
> **A25**: We do not understand your question very well. In Lemma 2.1, there does not exist $\lambda$. When employing Lemma 2.1 at the beginning of page 18, we simply apply it to $\xi=\xi^t, Z=w_t\gamma_tv_{t-1}(z^{t-1}-x^t)$ and $\mathcal{F}=\mathcal{F}^{t-1}$ to obtain $\mathbb{E}\left[\exp\left(w_t\gamma_tv_{t-1}\langle\xi^t,z^{t-1}-x^t\rangle\right)\mid\mathcal{F}^{t-1}\right]\leq\exp\left(w_t^{2}\gamma_t^2v_{t-1}^2\sigma^2\\|z^{t-1}-x^t\\|^{2}\right)$. If this is not what you want, could you please clarify your question further?
>
> ---
> >**Q26**: Beginning of page 19: $w_1$ is undefined since contains a sum starting from $2$ $\cdots$.
>
> **A26**: We believe $w_1$ is explicitly well-defined since there is $w_1=\left(\sum\_{s=2}^1\frac{2\gamma\_s\eta\_s\bar{v}\_s\sigma^2}{1-\mu\_f\eta\_s}+\sum\_{s=1}^T2\gamma\_s\eta\_s\bar{v}\_s\sigma^2\right)^{-1}=\left(\sum\_{s=1}^T2\gamma\_s\eta\_s\bar{v}\_s\sigma^2\right)^{-1}$ from the definition where $\sum_{s=2}^1\frac{2\gamma_s\eta_s\bar{v}_s\sigma^2}{1-\mu_f\eta_s}=0$ is by mathematical convention. However, to reduce the confusion, we added a sentence to describe what $w\_1$ is on Page 17 above in Eq. (10) in the first revision.
>
> ---
> >**Q27**: Page 19, bound of $w_1/w_T$, please explain the inequality.
>
> **A27**: Let $\rho\coloneqq\max_{2\leq t\leq T}\frac{1}{1-\mu_{f}\eta_{t}}$. Clearly, there is $\rho\geq\frac{1}{1-\mu_f\eta_s},\forall2\leq s\leq T$. Then the inequality holds due to $$\frac{\sum_{s=1}^T2\gamma_s\eta_s\bar{v}_s\sigma^2+\sum\_{s=2}^T\frac{2\gamma_s\eta_s\bar{v}_s\sigma^2}{1-\mu_f\eta_s}}{\sum\_{s=1}^T2\gamma_s\eta_s\bar{v}_s\sigma^2}=1+\frac{\sum\_{s=2}^T\frac{2\gamma_s\eta_s\bar{v}_s\sigma^2}{1-\mu_f\eta_s}}{\sum\_{s=1}^T2\gamma_s\eta_s\bar{v}_s\sigma^2}\leq1+\frac{\rho\sum\_{s=2}^T2\gamma_s\eta_s\bar{v}_s\sigma^2}{\sum\_{s=1}^T2\gamma_s\eta_s\bar{v}_s\sigma^2}\leq 1+\rho.$$
>
> ---
> We hope our answers can address the reviewer's concerns and help the reviewer verify the proofs. The other two parts of the response will be posted later. We are also preparing the next revision mainly focusing on the improvements in the writing and readability. Please stay tuned.
>
> ---
> **References**
>
> [1] Rakhlin et al. Making gradient descent optimal for strongly convex stochastic optimization. arXiv:1109.5647 (2011).
>
> [2] Lan, Guanghui. First-order and stochastic optimization methods for machine learning. Springer, 2020.
>
> [3] Liu et al. High probability convergence of stochastic gradient methods. ICML 2023.
>
> [4] Nguyen et al. SGD and Hogwild! convergence without the bounded gradients assumption. ICML 2018.

---

> ### Author Response · Authors · 2023-11-18
> **Official Response Part II**
>
> In this part, we are focusing on the questions about writing and readability. The changes mentioned in the following answers are added in the second revision updated on Nov 17.
>
> ----
> >**Q2**: Before Sec. 4 the authors say "to the best knowledge, our results are entirely new and the first to prove ...." $\cdots$.
>
> **A2**: These two sentences have been modified as the reviewer suggested.
>
> ---
> >**Q4**: Throughout, the authors use notation such as (this one is for example from Lemma 4.1 but it is like this all around the paper and the proofs) $\eta_{t\in[T]}\leq\frac{1}{2L\lor\mu_f}$ $\cdots$.
>
> **A4**: The notations have been changed throughout the paper accordingly.
>
> ---
> >**Q5**: In Lemma 4.1 and other places, author also use definitions such as $v_{t}=\frac{w_T\gamma_T}{\sum_{s=1\lor t}^Tw_s\gamma_s}$ $\cdots$.
>
> **A5**: The notations have been changed according to the reviewer's suggestion.
>
> ---
> >**Q7**: Many places, such as the footnote in page 6 says that "Harvey et al. (2019a) claims their proof can extend do sub-Gaussian noises ...." $\cdots$.
>
> **A7**: Thanks for the suggestion! In the second revision, we have further clarified the related discussion when talking about the assumptions on bounded noises and compact domains. Instead of saying the existing works rely on both these assumptions, we restate it as the existing works rely on one of these two assumptions. Consequently, we remove the previous footnote/sentence about [1].
>
> ---
> >**Q11**: page 4 and many places of the paper refer to the book Lan, 2020 for some results $\cdots$.
>
> **A11**: We have clarified them following the reviewer's advice.
>
> ---
> >**Q12**: Sec 2.1 is unnecessary and should be moved to the appendix $\cdots$.
>
> **A12**: We put it there mainly to connect and contrast the two common convergence criteria. For optimization experts, this is certainly redundant. However, since ICLR has a broad audience, some may not be very familiar with the concepts. We will make the paragraph as concise as possible so that it doesn't take much space. Besides, we have rephrased the corresponding sentence following the reviewer's advice.
>
> ---
> >**Q18**: Paper uses the phrase at many places (including the title) "last-iterative convergence" $\cdots$.
>
> **A18**: Thanks for pointing it out. They have been all changed according to the suggestion.
>
> ---
> >**Q20**: Proof of Lemma 4.1. Explain the requirements of $v_k$ in the beginning of the proof $\cdots$.
>
> **A20**: The explanation is added following the reviewer's suggestion.
>
> ---
> >**Q22**: After eq. (7), when authors use convexity of $F$ with $z^t$, $z^t$ is not written in a proper way $\cdots$.
>
> **A22**: The explanation is added following the reviewer's suggestion.
>
> ---
> >**Q28**: After Lemma 4.2, the authors again say "first unified inequality for the in-expectation" and after Lemma 4.3 $\cdots$.
>
> **A28**: We have rephrased these two sentences to make them more concise. Please let us know if the reviewer still does not think they are proper.
>
> ---
> **References**
>
> [1] Harvey et al. Tight analyses for nonsmooth stochastic gradient descent. COLT 2019.

---

> ### Author Response · Authors · 2023-11-18
> **Official Response Part III (1/3)**
>
> In this part, we clarify the differences compared with prior works mentioned by the reviewer in the Weakness/Question part:
>
> ---
> The following two questions are about [1] and [2]
> >**Q1**: Are the results of Zamani, Glineur 2023 and Liu et al., 2023 work well together to get the high probability rate on the last iterate $\cdots$.
>
> >**Q8**: What are the main technical novelties in addition to Liu et al., 2023 $\cdots$.
>
> **A1&8**: We list the key differences from [1] and [2] here:
>
> **Compared with [1]**: The starting point of the proof is very different. The proof of [1] can be viewed as beginning from the convex inequality $f(x^t)-f(z^t)\leq\langle g^t,x^t-z^t\rangle$ where $g^t\in\partial f(x^t)$. However, if we still start from convexity when considering composite optimization (for simplicity, we assume there are no noises in the gradient, $\mu_f=\mu_h=0$, $\\|\cdot\\|=\\|\cdot\\|_2$ and $\psi(x)=\frac{1}{2}\\|x\\|_2^2)$, there will be
> \begin{align*}
> F(x^t)-F(z^t)&=f(x^t)-f(z^t)+h(x^t)-h(z^t)\leq\langle g^t,x^t-z^t\rangle+h(x^t)-h(z^t)\\\\
> &=\langle g^t,x^{t+1}-z^{t}\rangle+\langle g^t,x^t-x^{t+1}\rangle+h(x^t)-h(z^t)\\\\
> &\overset{(a)}{\leq}\frac{\\|z^t-x^t\\|^2-\\|z^t-x^{t+1}\\|^2-\\|x^{t+1}-x^t\\|^2}{2\eta_t}+h(z^t)-h(x^{t+1})+\langle g^t,x^t-x^{t+1}\rangle+h(x^t)-h(z^t)\\\\
> &=\frac{\\|z^t-x^t\\|^2-\\|z^t-x^{t+1}\\|^2}{2\eta_t}+\langle g^t,x^t-x^{t+1}\rangle-\frac{\\|x^{t+1}-x^t\\|^2}{2\eta_t}+h(x^t)-h(x^{t+1})\\\\
> &\overset{(b)}{\leq}\frac{\\|z^t-x^t\\|^2-\\|z^t-x^{t+1}\\|^2}{2\eta_t}+\frac{\eta_t\\|g^t\\|^2}{2}+h(x^t)-h(x^{t+1})\\\\
> \Rightarrow\eta_t(F(x^t)-F(z^t))&\leq\frac{\\|z^t-x^t\\|^2-\\|z^t-x^{t+1}\\|^2}{2}+\frac{\eta^2_t\\|g^t\\|^2}{2}+\eta_t(h(x^t)-h(x^{t+1})),
> \end{align*}
>
> where $(a)$ is by the optimality condition for $x^{t+1}=\mathrm{argmin}_{x\in\mathcal{X}}h(x)+\langle g^t,x-x^t\rangle+\frac{\\|x-x^t\\|^2}{2\eta_t}$ (for a detailed explanation of this step, please refer to how to bound term II on page 13 of the paper) and $(b)$ is due to Cauchy-Schwarz inequality and AM-GM inequality to get $\langle g^t,x^t-x^{t+1}\rangle\leq \\|g^t\\|\\|x^t-x^{t+1}\\|\leq \frac{\\|x^{t+1}-x^t\\|^2}{2\eta_t}+\frac{\eta_t\\|g^t\\|^2}{2}$.
>
> Note that when $h=0$ and $\\|g^t\\|\leq G$, one can follow [1] to finish the following proof. However, in our setting, there will be two problems.
>
> 1)  The term of $\\|g^t\\|^2$ can not be bounded since we are considering the general $(L,M)$-smoothness assumption. Even for $M=0$ (i.e., the standard smooth case), $\\|g^t\\|^2$ does not admit a uniform bound.
>
> 2) The term $\eta_t(h(x^t)-h(x^{t+1}))$ can not be bounded. Note that it cannot be telescoping summed either since we need to multiply both sides by $v_t$ (or $v_t^2$ if we follow [1] rigorously) before summing from $t=1$ to $T$.
>
> Due to these two reasons, we start our proof from the $(L,M)$-smoothness as mentioned at the beginning of the proof of Lemma 4.1 on page 13 instead of employing convexity directly. This difference from [1] is very important to make our proof be able to work.
>
> **Compared with [2]**: The main difference is in how to find $w_t$. That is saying that if we follow the way in [2], then one can find the following definition: $$w_t\coloneqq\left(\sum_{s=2}^{t}\frac{2\gamma_s\eta_sv_s\sigma^2}{1-\mu_f\eta_s}+\sum_{s=1}^T2\gamma_s\eta_sv_s\sigma^2\right)^{-1},\forall t\in[T].$$ However, recalling that $v_t$ is defined as $v_t\coloneqq\frac{w_T\gamma_T}{\sum_{s=t}^Tw_s\gamma_s},\forall t\in[T]$ and $v_0\coloneqq v_1$, this means that we need $w_t$ to define $v_t$ and also need $v_t$ to define $w_t$, which is like the problem of "Chicken or the egg".
>
> A potential way to deal with this issue is to plug $v_t$ into the above definition of $w_t$ (or do it reversely) to obtain a set of equations w.r.t. $w_t,\forall t\in[T]$ and prove that there exists a sequence of $w_t,\forall t\in[T]$ satisfying the corresponding equations. However, this approach is rather complicated.
>
> Instead, what we did in the paper is to use a new proxy sequence $\bar{v}_t\coloneqq\frac{\gamma_T}{\sum\_{s=t}^T\gamma_s},\forall t\in[T]$ and $\bar{v}_0\coloneqq\bar{v}_1$ to define $$w_t\coloneqq\left(\sum\_{s=2}^t\frac{2\gamma_s\eta_s\bar{v}_s\sigma^2}{1-\mu_f\eta_s}+\sum\_{s=1}^T2\gamma_s\eta_s\bar{v}_s\sigma^2\right)^{-1},\forall t\in[T].$$ As the reviewer can see, this difference helps us get rid of the above problem and makes the proof relatively simple.
>
> Moreover, a paragraph for [2] has been added in Section 2.1 following the reviewer's advice in the second revision.
>
> ---
> **References**
>
> [1] Zamani et al. Exact convergence rate of the last iterate in subgradient methods. arXiv preprint arXiv:2307.11134.
>
> [2] Liu et al. High probability convergence of stochastic gradient methods. ICML 2023.

---

> > ### Comment · Reviewer_V1N7 · 2023-11-21
> > **Follow-up**
> >
> > Thank you for the detailed rebuttal. I am still trying to go over and verify the provided feedback. So, I will write more later but I want to follow-up as quickly as possible as I am reading so that the authors will have chance to reply. Thank you for writing things out explicitly.
> >
> > About this aspect, I said it before, but let me repeat: Even if two tools, in this case, Zamani, Glineur and Liu et al., 2023 work well together (without major difficulties while combining two results), this is **not** a reason for rejection in my opinion, as long as the provided result is interesting. And as I said before, I think the result in this paper **is** interesting. What I am trying to prevent is not having misleading statements in the paper or "oversell" of the result so that readers of the paper will understand well what is happening. This is what I am attempting, I am definitely not trying to be adversarial. As long as I can verify the result of the paper (I am still working on this, thanks for clarifications in your rebuttal), I will be voting for acceptance.
> >
> > So, first, I will go over the reasoning the authors give for why starting the proof like Zamani, Glineur does not work. I do not quite agree with this. To start with, it would be better if the authors would refer where their estimations above fits in Zamani, Glineur or their own proof. Simply saying "start of the proof is different" is not very descriptive since there are so many different results in both papers. So, in the future, please show clearly the estimations you are referring to. In this case, I think you are referring to the proof of Lemma 2.1 in Zamani, Glineur, just by judging the inequalities.
> >
> > > "Compared with [1]: The starting point of the proof is very different"
> >
> > So here in the inequality, you are bounding $\langle g^t, x^t - x^{t-1}\rangle$ with AM-GM and then arguing the resulting term $\|g^t\|^2$ is difficult to bound, and also the additional $h(x^t)  - h(x^{t+1})$ that you argue to be hard to bound. But I think AM-GM here is too lose and not the correct tool to use.Instead of AM-GM, why not use smoothness here? When we have smoothness of $f$ (or essentially Assumption 3 in the submission), we have (for simplicity I omit $M\|x-y\|$ in your assumption)
> > $$
> > f(x^{t+1}) \leq f(x^t) + \langle g^t, x^{t+1} - x^t \rangle + \frac{L}{2} \| x^{t+1} - x^t\|^2.
> > $$
> > This gives
> > $$
> > \langle g^t, x^{t} - x^{t+1} \rangle \leq f(x^t) - f(x^{t+1})  + \frac{L}{2} \| x^{t+1} - x^t\|^2.
> > $$
> > When you plug this in your inequality chain (essentially labeled as (b) where you bound $\langle g^t, x^t - x^{t+1} \rangle$), you can also join $f(x^t) - f(x^{t+1})$ to $h(x^t)  - h(x^{t+1})$ on the right hand side to make up $F(x^t) - F(x^{t+1})$ and get
> > $$
> > \eta(F(x^{t+1}) - F(z^t) ) \leq \frac{1}{2}\left( \| x^t - x^t \|^2 - \| z^t - x^{t+1}\|^2 \right) + \left(\frac{L\eta_t}{2} - \frac{1}{2} \right) \| x^{t+1} - x^t\|^2
> > $$
> > Note that compared to the inequality you have in the rebuttal, I get $F(x^{t+1})$ instead of $F(x^t)$ on the left side.
> >
> > Then, this, to me seems a very similar inequality to the proof of Lemma 4.1 in the submission (eq. (6) in the current version). So, my argument is when you use the correct tools (smoothness of $f$ instead of AM-GM), even if you start as Zamani, Glineur, you will get the same thing (this solves both of the problems you pointed out). Therefore, it is a difficult argument to make to say that starting from a different point will make things not work, it always depends on the tools you are using. I think what I describe here also applies below where you point out Orabona's approach does not work. You also have AM-GM there for the inner product whereas the correct tool to use is smoothness, so also take this as my answer to Part III(2/3) where you argue Orabona's proof does not work out. Am I missing something here?

---

> ### Author Response · Authors · 2023-11-18
> **Official Response Part III (2/3)**
>
> The following questions are all about [1].
> >**Q3**: In my understanding, the analysis of Zamani, Glineur 2023 seems like an alternative to Orabona, 2020 $\cdots$.
>
> >**Q6**: Please describe clearly in which ways Lemma 4.2 improves over the result of Orabona, 2020 $\cdots$.
>
> >**Q14**: The discussion after Theorem 3.1 says also that "Orabona 2020 cannot be applied to composite optimization" $\cdots$.
>
> >**Q15**: Page 5 also claims that the results are the first improvement over Moulines, Bach 2011 for smooth problems $\cdots$.
>
> **A3&6&14&15**: We give a detailed comparison with [1] in this answer. Let us first recall the key lemma in [1]:
>
> >**Lemma 1  in [1]**:  Let $\eta_t>0,\forall t\in[T]$ be a non-increasing sequence and $q_t\geq0,\forall t\in[T]$. Then $\eta_Tq_T\leq\frac{1}{T}\sum_{t=1}^T\eta_tq_t+\sum_{k=1}^{T-1}\frac{1}{k(k+1)}\sum_{t=T-k+1}^{T}\eta_t(q_t-q_{T-k})$.
>
> Now, we list the differences:
>
> **Can [1] be extended to smooth or/and composite case straightforwardly?** Our answer is **no** (for simplicity, the following discussion is for $\\|\cdot\\|=\\|\cdot\\|_2$ and $\psi=\frac{1}{2}\\|x\\|^2$ with $\mu_f=\mu_h=0$). Let us check what will happen if we follow [1]. Let $q_t=F(x^t)-F(x^*)$ be the same as [1] to get $$\eta_T(F(x^t)-F(x^*))\leq\frac{1}{T}\underbrace{\sum\_{t=1}^T\eta_t(F(x^t)-F(x^*))}_A+\sum\_{k=1}^{T-1}\frac{1}{k(k+1)}\underbrace{\sum\_{t=T-k+1}^T\eta_t(F(x^t)-F(x^{T-k}))}_B.$$ Now let us bound $A$ and $B$ by mimicking what [1] did. Specifically, we follow the similar step that [1] used to prove Eq. (1) in the blog. Let $g^t=\nabla f(x^t)$ and $\widehat{g}^t$ be the stochastic gradient, for any $u\in\mathcal{X}$ (we need the bound for any $u$ since it will also be used to bound $B$ as in [1]).
>
> \begin{align*}
> F(x^t)-F(u)=&f(x^t)-f(u)+h(x^t)-h(u)\overset{(a)}{\leq}\langle g^t,x^t-u\rangle+h(x^t)-h(u)\\\\
> =&\langle g^t-\widehat{g}^t,x^t-u\rangle+\langle\widehat{g}^t,x^{t+1}-u\rangle+\langle\widehat{g}^t,x^t-x^{t+1}\rangle+h(x^t)-h(u)\\\\
> \overset{(b)}{\leq}&\langle g^t-\widehat{g}^t,x^t-u\rangle+\frac{\\|u-x^t\\|^2-\\|u-x^{t+1}\\|^2-\\|x^t-x^{t+1}\\|^2}{2\eta_t}+h(u)-h(x^{t+1})\\\\
> &+\langle\widehat{g}^t,x^t-x^{t+1}\rangle+h(x^t)-h(u)\\\\
> \overset{(c)}{\leq}&\langle g^t-\widehat{g}^t,x^t-u\rangle+\frac{\\|u-x^t\\|^2-\\|u-x^{t+1}\\|^2}{2\eta_t}+\frac{\eta_t\\|\widehat{g}^t\\|^2}{2}+h(x^t)-h(x^{t+1})\\\\
> \Rightarrow\mathbb{E}[\eta_t(F(x^t)-F(u))]\leq&\frac{\mathbb{E}[\\|u-x^t\\|^2-\\|u-x^{t+1}\\|^2]}{2}+\frac{\mathbb{E}[\eta_t^2\\|\widehat{g}^t\\|^2]}{2}+\eta_t\mathbb{E}[h(x^t)-h(x^{t+1})]\\\\
> \Rightarrow A\leq&\frac{\\|u-x^1\\|^2-\mathbb{E}[\\|u-x^{t+1}\\|^2]}{2}+\frac{\mathbb{E}[\sum_{t=1}^T\eta_t^2\\|\widehat{g}^t\\|^2]}{2}\\\\
> &+\underbrace{\eta_1h(x^1)-\eta_T\mathbb{E}[h(x^{T+1})]+\sum_{t=1}^T(\eta_t-\eta_{t-1})\mathbb{E}[h(x^t)]}\_C,
> \end{align*} where $(a)$ is by the convexity of $f$, $(b)$ is by the optimality condition of $x^{t+1}=\mathrm{argmin}_{x\in\mathcal{X}}h(x)+\langle\widehat{g}^t,x-x^t\rangle+\frac{\\|x-x^t\\|^2}{2\eta_t}$ (for a detailed explanation of this step, please refer to how to bound term II on page 13 of the paper) and $(c)$ is by Cauchy-Schwarz inequality and AM-GM inequality to get $\langle\widehat{g}^t,x^t-x^{t+1}\rangle\leq\\|\widehat{g}^t\\|\\|x^t-x^{t+1}\\|\leq\frac{\eta_t\\|\widehat{g}^t\\|^2}{2}+\frac{\\|x^t-x^{t+1}\\|^2}{2\eta_t}$.
>
> Now two problems show up:
> 1) The first problem is how to bound $\mathbb{E}[\\|\widehat{g}^t\\|^2]$. Of course, we can do $\mathbb{E}[\\|\widehat{g}^t\\|^2]=\mathbb{E}[\\|\xi^t+g^t\\|^2]=\mathbb{E}[\\|\xi^t\\|^2+\\|g^t\\|^2]\leq\sigma^2+\mathbb{E}[\\|g^t\\|^2]$. However, unlike the Lipschitz case in [1], $\mathbb{E}[\\|g^t\\|^2]$ does not have a uniform bound since we are in the smooth case.
>
> 2) The second one is the term $C$. Though $\eta_t\leq\eta_{t-1}$ as required by Lemma 1 in [1], we don't know whether $\mathbb{E}[h(x^t)]$ is positive or not. Hence, a possible way is to rewrite it as (where $h^*\coloneqq\inf_{x\in\mathcal{X}}h(x)$)
> $$C=\eta_1(h(x^1)-h^*)-\eta_T\mathbb{E}[(h(x^{T+1})-h^*)]+\sum_{t=1}^T(\eta_t-\eta_{t-1})\mathbb{E}[(h(x^t)-h^*)]\leq\eta_{1}(h(x^1)-h^*).$$
> However, this approach brings us an extra undesired factor at least (we use "at least" since other extra factors will appear when bounding $B$) in the order of $\eta_1(h(x^1)-h^*)/(\eta_TT)=O((h(x^1)-h^*)/\sqrt{T})$ (for dynamic step size) in the final bound for $F(x^T)-F(x^*)$. In contrast, our Lemma 4.2 does not have this term.
>
> The above two issues also appear when bounding $B$. For example, for the second problem, there will be a residual term $\eta_{T-k}(h(x^{T-k})-h^*)$, which finally gives us an extra term $(\eta_T)^{-1}\sum_{k=1}^{T-1}\frac{\eta_{T-k}(h(x^{T-k})-h^*))}{k(k+1)}$ when bounding $F(x^T)-F(x^*)$. Even for the special case, assuming $h$ is bounded on $\mathcal{X}$ and considering the constant step size $\eta_t=\eta$, this extra term is $O(1)$!
>
> Thus, one can see that [1] cannot be extended to either smooth or composite cases directly.

---

> ### Author Response · Authors · 2023-11-18
> **Official Response Part III (3/3)**
>
> **A3&6&14&15** (Cont'd):
>
> **Can [1] be extended to the case of Bregman divergence straightforwardly?** Our answer is **yes**. **But** as pointed out in Official Response Part III (2/3), it can only deal with the non-smooth non-composite case. In contrast, the setting in our paper is much more general.
>
> **Another important difference**. The proof system proposed by [1] can only bound $F(x^t)-F(x)$ where $x\in\mathcal{X}$ satisfies $F(x^t)-F(x)\geq0,\forall t\in[T]$. This is because Lemma 1 in [1] requires $q_t\geq0$. However, our proof can indeed bound $F(x^t)-F(x)$ for any $x\in\mathcal{X}$ (for both in-expectation and high-probability bounds). This is because we follow the idea introduced by [2].
>
> *Remark*: We didn't include this point in the second revision since we want to keep the paper simple. However, this point can be verified very easily. One just needs to define $z^0\coloneqq x$ for any $x\in\mathcal{X}$ instead of  $z^0\coloneqq x^*$. We are also happy to revise the paper further if the reviewer thinks including this point can improve the quality of our work.
>
> Here we use an example to explain why the bound for any $x\in\mathcal{X}$ is an important property.
>
> *Example (Excess Risk)*: Suppose we are interested in the problem $\min_{x\in\mathcal{X}}L(x)\coloneqq\mathbb{E}_{z\sim\mathcal{D}}[\ell(x,z)]$ where $\ell(x,z)$ is convex for any given $z$, a standard way is considering ERM method to minimize $\widehat{L}(x)\coloneqq\frac{1}{n}\sum\_{i=1}^n\ell(x,z\_i)$ by SGD where $z\_i,\forall i\in[n]$ are i.i.d. sampled from the distribution $\mathcal{D}$. Let $x^*\coloneqq\mathrm{argmin}\_{x\in\mathcal{X}}L(x)$, $\widehat{x}^*\coloneqq\mathrm{argmin}\_{x\in\mathcal{X}}\widehat{L}(x)$ and $\widehat{x}^T$ be the last iterate output by SGD (suppose the trajectory is $\widehat{x}^t,t\in[T]$ and $\widehat{x}^1$ is a deterministic starting point, e.g., $\mathbf{0})$, the excess risk can be decomposed as $$\mathbb{E}[L(\widehat{x}^T)-L(x^*)]=\mathbb{E}[\underbrace{L(\widehat{x}^T)-\widehat{L}(\widehat{x}^T)}_A+\underbrace{\widehat{L}(\widehat{x}^T)-\widehat{L}(x^*)}_B],$$ where the expectation is taken over the randomness in SGD and samples $\left\\{ z\_{i}\right\\}$.
>
> How to bound $A$ is not related to optimization, hence, we don't discuss it here. Let us focus on $B$. By our any point result, we can bound $B=O(\\|\widehat{x}^1-x^*\\|^2)$ (the dependence on $T$ is not important here, so we omit it). Note that both $\widehat{x}^1$ and $x^*$ are not random, so we have $\mathbb{E}[B]\leq O(\\|\widehat{x}^1-x^*\\|^2)$.
>
> Instead, if we follow [1], we have to decompose $B$ further to get $B=\widehat{L}(\widehat{x}^T)-\widehat{L}(x)+\widehat{L}(x)-\widehat{L}(x^*)$. The reason is that [1] can only bound $\widehat{L}(\widehat{x}^T)-\widehat{L}(x)$ for $x\in\mathcal{X}$ satisfying $\widehat{L}(\widehat{x}^t)-\widehat{L}(x)\geq0,\forall t\in[T]$ as mentioned. The most standard choice in statistical learning is $x=\widehat{x}^*$. In this case, we have $$B=\widehat{L}(\widehat{x}^T)-\widehat{L}(\widehat{x}^*)+\widehat{L}(\widehat{x}^*)-\widehat{L}(x^*)\leq\widehat{L}(\widehat{x}^T)-\widehat{L}(\widehat{x}^*)=O(\\|\widehat{x}^1-\widehat{x}^*\\|^2)\Rightarrow\mathbb{E}[B]\leq\mathbb{E}[O(\\|\widehat{x}^1-\widehat{x}^*\\|^2)].$$ Whereas, note that $\widehat{x}^*$ is random due to its definition. Hence, we need to pay extra efforts to find a proper bound on $\mathbb{E}[O(\\|\widehat{x}^1-\widehat{x}^*\\|^2)]$.
>
> From this example, we can see that any point property is important.
>
> ---
> We have answered all 28 questions asked by the reviewer. Please feel free to let us know if you have more concerns about our work.
>
> ---
> **References**
>
> [1] Francesco Orabona. Last iterate of sgd converges (even in unbounded domains). 2020.
>
> [2] Zamani et al. Exact convergence rate of the last iterate in subgradient methods. arXiv preprint arXiv:2307.11134.

---

> ### Author Response · Authors · 2023-11-22
> **Thanks for the detailed feedback**
>
> We would like to express sincere thanks for the reviewer's detailed feedback. We believe such a discussion can improve the quality of our paper further.
>
> ---
> > So, in the future, please show clearly the estimations you are referring to. In this case, I think you are referring to the proof of Lemma 2.1 in Zamani, Glineur, just by judging the inequalities.
>
> Thanks for the advice. We will make our response clearer in the future. Your statement is correct. We are referring to Lemma 2.1 in [1].
>
> ---
> For the proof proposed by the reviewer:
>
> - We fully agree that the reviewer's argument works well to obtain the last-iterate convergence and thus can solve the issues we mentioned above for [1] and [2].
>
> - It would be worth noting that this proof idea is essentially the same as ours in the paper (but only in a different order when employing assumptions and inequalities). Consequently, Eq. (6) (for the smooth and general convex case) can be obtained following this idea as pointed out by the reviewer.
>
> - We think this is exactly the difference compared with [1] and [2]. In other words, the difference is how to use smoothness. Of course, this difference may not be significant for the experts in the optimization area. But for one who does not focus on optimization, it may not be easy to realize how to incorporate the smoothness in the proof system in [1] or [2]. Instead, meeting the problems mentioned in our previous response is highly possible.
>
> - To further incorporate the reviewer's view and improve the writing. We plan to make the following changes later (**update: the following changes have been done in the third revision**):
>
> 1) When compared to [2] at the end of Page 5 for the smooth case, following the reviewer's suggestion, we will say that [2] does not explicitly give the proof for the smooth case.
>
> 2) At the beginning of the third paragraph in Section 4 on Page 8, instead of saying ''Though the proof in Zamani & Glineur (2023) only works for deterministic non-composite...'', we will change it to ''Though Zamani & Glineur (2023) only shows how to prove the last-iterate convergence for deterministic non-composite...'',
>
> Besides, Compared with [2], we believe the last difference still holds (i.e., the **Another important difference** in Official Response Part III (3/3))
>
> ---
> **References**
>
> [1] Zamani et al. Exact convergence rate of the last iterate in subgradient methods. arXiv preprint arXiv:2307.11134.
>
> [2] Francesco Orabona. Last iterate of sgd converges (even in unbounded domains). 2020.

---

> > ### Comment · Reviewer_V1N7 · 2023-11-23
> > **Follow-up and increasing of score - Part 1/2**
> >
> > 1. I am increasing to a "6" and vote for acceptance. This is because the result is interesting and worthy of publication in the venue in my opinion. The reason I am not increasing further is because I need to reserve the higher scores for papers with more significant contributions in my view. Result of our discussion and my current understanding is that even though the combination of two different techniques in Zamani, Glineur, 2023 and Liu et al., 2023 is certainly non-trivial, in my reading, it also does not present significant challenges that requires many additional techniques. As the authors say, a judgement like this (whether a difference is significant or not) depends on the expertise of the reviewer, but this is always the case for peer-review and my criteria reflects my view.
> >
> > 2. The authors say:
> >
> > > "Of course, this difference may not be significant for the experts in the optimization area. But for one who does not focus on optimization, it may not be easy to realize how to incorporate the smoothness in the proof system in [1] or [2]. Instead, meeting the problems mentioned in our previous response is highly possible."
> >
> > This is fair and thank you for your honesty. But please make sure that writing of your paper reflects this. The writing of the paper should not be tailored to impressing the unknowledgeable readers, instead it should be tailored also for experts who will try to understand your techniques. Of course, I understand that sometimes for unknowledgeable reviewers (who are unfortunately the majority), it may hurt the authors to clearly explain the connections with the existing works/techniques because the unknowledgeable reviewers then may use this to claim that the paper is not novel enough. However, knowledgeable reviewers will know to judge the value of combining two existing results (in this case both techniques are highly technical and the combination is nontrivial in my opinion) and the significance of the result. We cannot adapt when writing to unknowledgeable reviewers. Papers serve a greater purpose of disseminating knowledge than getting acceptance decisions from unknowledgeable reviewers. Please keep the "overselling" in the paper as small as possible and try to have parts for the expert readers and describe to experts your idea clearly.
> >
> > 3. Please add in your paper the explanation that you provided to me for the estimations I had trouble following. Likely, your readers will get stuck in similar places, please help them understand your paper easier.
> >
> > 4. Please add the examples for bounded variance (and subGaussian noise) to your main text.
> >
> > 5. Please include your comments about the alternative approaches such as Orabona's in the revision of the paper. For example, you say around Theorem 3.3 that Orabona's approach is circuitous, why? This looks subjective, please provide objective reasons and a fair comparison. Also please add your comment about "Another important difference" in Part III (3/3) in your revision text.
> >
> > 6. I recommend that you do not keep the step size depending on $x^*$ in Theorem 3.1. This will confuse many of your readers. Please have realistic parameter choices.
> >
> > 7. Your argument with the choice of set $\mathcal{X}$ is not very accurate. In particular, if your set $\mathcal{X}$ is not $\mathbb{R}^d$ and if $h$ is not nonzero, how are you going to compute the prox step of the algorithm? Because now the prox step will involve the prox of $h+i_{\mathcal{X}}$ where $i$ is the indicator function. Of course, computing prox of the sum generally is not computationally efficient even when the prox of individual terms is efficient. So, you either restrict $\mathcal{X}$ but then you have to set $h$ to be zero, or efficiency of prox will go away. This is why I said why not set $\mathcal{X}=\mathbb{R}^d$. As such, the discussion with compact $\mathcal{X}$ is not very useful since it is the main contribution of this paper to remove compactness assumption. Otherwise, there is already Harvey et al's result. Since there is almost no time left for the discussion window, authors should not feel pressure to respond to this, it will not change my score. I am only putting this here for improving the presentation of your paper.
> >
> > 8. try to keep exaggerating words such as "great" or "very" to a minimal (both abstract and the text).  Also, as we established during the discussion period, some of the extensions such as [composite case, smooth case, Bregman case] are not necessarily that non-trivial. Emphasizing these points too much in your abstract actually distracts away the reader from your main contribution, which is the high probability guarantees for last iterate without bounded domains.

---

> > > ### Comment · Reviewer_V1N7 · 2023-11-23
> > > **Part 2/2**
> > >
> > > 9. Related to point 8, for now, I cannot go over the text (unfortunately the changes are not colored in the revision and I cannot compare versions in the system) to make sure the "oversell" is reduced in the text, because I already spent a significant amount of time working on this paper. Please do these changes. If I can go through the text later, I might add more comments and revisions to fix.
> > >
> > > typos: Lipschtiz in the abstract and page 1, please go through your paper to correct typos.

---

> > > > ### Author Response · Authors · 2023-11-23
> > > > **Thanks for the comments and increasing the score**
> > > >
> > > > We greatly thank the reviewer for the follow-up comments and for increasing the score. We will polish our paper more based on your suggestions.

---

### Author Response · Authors · 2023-11-13
**Summary of revision**

We thank the reviewers for their valuable comments on our work. We will summarize the changes in the revision below:

- **First Revision (Nov 12)**: Some typos were fixed. We also changed/added some expressions according to the reviews.

- **Second Revision (Nov 17)**: Following the suggestions from reviewer V1N7 and reviewer yV1Y, we have further polished the writings and improved the readability. One can refer to our responses to the reviewers to check what has been changed.

- **Third Revision (Nov 22)**: Following the suggestions from reviewer V1N7, we have further modified some sentences related to prior works. One can refer to our response to the reviewer V1N7 to see the changes.

---

### Meta-Review · Area_Chair_wMum · 2023-12-06

**Metareview:**

This paper treats the last-iterate – not "last-iterative" - convergence rate of SGD in convex and strongly convex problems, and it lifts the "bounded domain" and "bounded noise" assumptions of previous works on the topic.

This paper was reviewed and discussed meticulously and extensively - Reviewer V1N7 in particular provided a real service to the authors - and the conclusion was that the paper's merits outweigh its flaws.

A key concern that came up in the discussion was that the original version of the paper was overselling certain claims instead of providing a clearer positioning that would allow experts to correctly and concisely pinpoint the new tools and techniques that the paper is contributing. This issue was largely fixed, and the current version is much clearer in this regard. [At the same time, the authors should remove all non-factual adjectives and adverbs like "vastly" and the like, as these are at the readers' discretion to opine upon, not the authors']

Another technical concern that came up during the discussion phase had to do with the fact that certain of the paper's technical assumptions do not line up well with each other, especially regarding the Bregman case. For example, the Bregman strong convexity part of Assumption 2 is, in general, incompatible with Assumption 3 ("general smoothness"): if the Bregman function is Legendre (i.e., its gradient blows up at the boundary of its domain), the objective cannot be Lipschitz smooth (or even Lipschitz continuous), as it has to be lower-bounded by a non-Lipschitz minorant. It is not clear if this limitation can be lifted: the paper is stating right after the assumptions that ordinary Lipschitz smoothness can be replaced by "relative" Lipschitz smoothness in the sense of Bauschke et al. and Lu et al. (which would solve the blow-up issue), but the committee was not convinced that the analysis would go through as smoothly as the authors suggest (and, at any rate, the burden of proof here lies with the authors).

Some additional incompatibilities that were pointed out by the reviewers have to do with the variance assumption versus strong convexity, and with Lipschitzness versus strong convexity when the domain is unbounded.

Regardless, after the extensive discussion, the committee felt that these issues can be left to the authors to fix without a fresh round of reviews, so a decision was reached to make an "accept" recommendation.

**Justification For Why Not Higher Score:**

Likely concerns a relatively small subset of the ICLR audience.

**Justification For Why Not Lower Score:**

The contribution is technically and conceptually valid.

---

### Decision · Program_Chairs · 2024-01-16

Accept (poster)